# Decoupling the Class Label and the Target Concept in Machine Unlearning

**Jianing Zhu**[1,2*]   **Bo Han**[1†]   **Jiangchao Yao**[3]   **Jianliang Xu**[1]
**Gang Niu**[2]   **Masashi Sugiyama**[2,4]
[1]Hong Kong Baptist University [2]RIKEN Center for Advanced Intelligence Project
[3]CMIC, Shanghai Jiao Tong University [4]The University of Tokyo
{csjnzhu, xujl, bhanml}@comp.hkbu.edu.hk Sunarker@sjtu.edu.cn
 gang.niu.ml@gmail.com sugi@k.u-tokyo.ac.jp

## Abstract

Machine unlearning as an emerging research topic for data regulations, aims to adjust a trained model to approximate a retrained one that excludes a portion of training data. Previous studies showed that class-wise unlearning is effective in forgetting the knowledge of a training class, either through gradient ascent on the forgetting data or fine-tuning with the remaining data. However, while these methods are useful, they are insufficient as the class label and the target concept are often considered to coincide. In this work, we expand the scope by considering the label domain mismatch and investigate three problems beyond the conventional *all matched* forgetting, e.g., *target mismatch*, *model mismatch*, and *data mismatch* forgetting. We systematically analyze the new challenges in restrictively forgetting the target concept and also reveal crucial forgetting dynamics in the representation level to realize these tasks. Based on that, we propose a general framework, namely, *TARget-aware Forgetting* (TARF). It enables the additional tasks to actively forget the target concept while maintaining the rest part, by simultaneously conducting annealed gradient ascent on the forgetting data and selected gradient descent on the hard-to-affect remaining data. Empirically, various experiments under our newly introduced settings are conducted to demonstrate the effectiveness of our TARF. Our code is publicly available at https://github.com/tmlr-group/TARF.

## 1 Introduction

In response to data regulations (Hoofnagle et al., 2019), machine unlearning (Cao & Yang, 2015; Shaik et al., 2023; Xu et al., 2023) has emerged to eliminate the influence of training data from a trained model (Xu et al., 2023). The intuitive goal is to forget the specific data as if the model had never used it during training (Bourtoule et al., 2021). To achieve that, a direct way (Shaik et al., 2023) is to retrain the model from scratch by excluding the data to be unlearned, termed *exact unlearning*. Considering the intensive computational cost, much attention has been paid to *approximate unlearning* (Izzo et al., 2021; Chen et al., 2023; Fan et al., 2023), which adjusts the trained model for approximating the behaviors of the retrained one. Focusing target granularity as semantic clusters, recent studies (Kurmanji et al., 2023; Jia et al., 2023; Chen et al., 2023; Fan et al., 2023) showed *class-wise* unlearning is effective in forgetting the knowledge of a training class, either through reverse optimization (Thudi et al., 2022a; Izzo et al., 2021) on the class data or fine-tuning on the remaining data (Golatkar et al., 2020) to realize catastrophic forgetting (Kirkpatrick et al., 2017).

Despite the promising achievements, the previously studied scenario (Warnecke et al., 2023; Golatkar et al., 2020; Chen et al., 2023; Jia et al., 2023; Fan et al., 2023) mainly assumed the target concept to coincide with the class label, overlooking that the practical unlearning request (Bommasani et al., 2021; Hashimoto et al., 2018; Kovashka et al., 2016) may violate the taxonomy of the pre-training tasks. Raised by the model users, the reported cases to be unlearned can involve different concerns from original tasks, spanning from privacy, fairness, copyright, or the hazardous capabilities (Li et al.,

---

*Work done during an internship at RIKEN Center for Advanced Intelligence Project.

†Correspondence to Bo Han (bhanml@comp.hkbu.edu.hk).

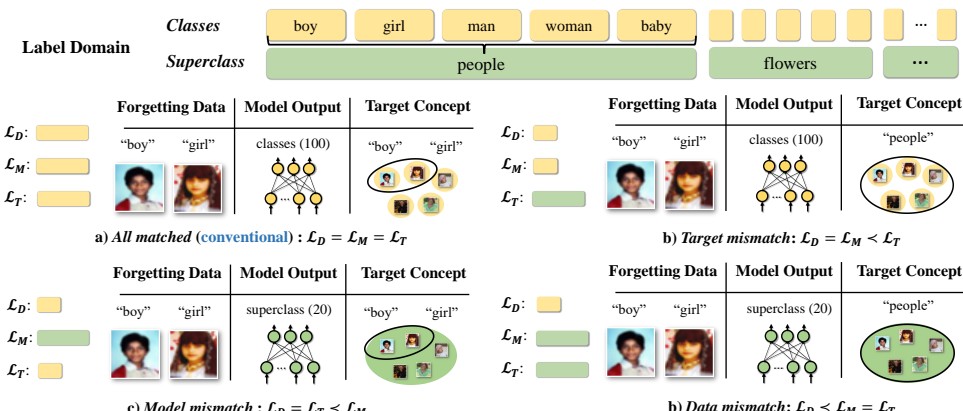

Taking the *CIFAR-100* (Krizhevsky, 2009) dataset with its classes and superclass (two different label domains for modeling different taxonomy from pre-training tasks) as an example, we instantiate four tasks given the same forgetting data with the class labels of "boy" and "girl": a) *all matched forgetting* (conventional scenario): unlearn "boy" and "girl" with the model trained on the classes; b) *target mismatch forgetting*: unlearn "people" with the model trained on the classes; c) *model mismatch forgetting*: unlearn "boy" and "girl" with the model trained on the superclass; d) *data mismatch forgetting*: unlearn "people" with the model trained on the superclass. More discussion is provided in Appendix D.

Figure 1: Illustrations of decoupling the class label and the target concept.

2024), which can not always be the conventional matched scenario where all the identified correspond to one pre-training class. In contrast, those cases may be only a semantic subset within a class, for which the model developer needs to unlearn the small set considering reserving model utility on the other parts. In addition, sometimes the user would identify limited cases of the target concept. With a conservative attitude for protecting the reputation of serving (Bommasani et al., 2021; Liu et al., 2024; Li et al., 2024) (e.g., IP conflicts), the developer tends to unlearn a larger semantic cluster.

In this work, we decouple the target concept with the class label, to model the unlearning scenarios for research explorations. To be specific, we consider the different label domains of the forgetting data $\mathcal{L}_D$, the model output $\mathcal{L}_M$, and the target concept $\mathcal{L}_T$ in unlearning. We introduce two relations between two label domains, i.e., $\mathcal{L}_1$ matches $\mathcal{L}_2$ ($\mathcal{L}_1 = \mathcal{L}_2$) and $\mathcal{L}_1$ is the subclass domain of $\mathcal{L}_2$ ($\mathcal{L}_1 \prec \mathcal{L}_2$)[1], then modeling scenarios corresponding to the target concept being larger or smaller than the class unit. As the reported forgetting data are included in the target concept, e.g., $\mathcal{L}_D \preceq \mathcal{L}_T$, we have *all matched* $\mathcal{L}_D = \mathcal{L}_T = \mathcal{L}_M$; *target mismatch* $\mathcal{L}_D = \mathcal{L}_M \prec \mathcal{L}_T$; *model mismatch* $\mathcal{L}_D = \mathcal{L}_T \prec \mathcal{L}_M$; and *data mismatch* $\mathcal{L}_D \prec \mathcal{L}_T = \mathcal{L}_M$ settings (task instances refer to Figure 1).

Given the aforementioned tasks, we identify new challenges with the mismatched label domains (refer to Figure 2). Unlike the accurate unlearning approximation in the conventional all matched task (Golatkar et al., 2020; Jia et al., 2023; Chen et al., 2023), the representative unlearning methods (Warnecke et al., 2023; Thudi et al., 2022a) exhibit different performance gap with the retrained reference in the other tasks. Specifically, the under-entangled feature representation (when $\mathcal{L}_M \prec \mathcal{L}_T$) or the under-representative forgetting data (when $\mathcal{L}_D \prec \mathcal{L}_T$) results in insufficient forgetting, while the entangled feature representation (when $\mathcal{L}_T \preceq \mathcal{L}_M$) prevents the decomposition of target concept with the retaining part. The former requires target identification in the remaining dataset, while the latter requires explicit target separation over the entangled feature representation.

Based on the above analysis, we propose a novel framework, namely, *TARget-aware Forgetting* (TARF), for unlearning. In general, we consider two parts (refer to Eq. 3), i.e., annealed forgetting and target-aware retaining, which collaboratively enable the target identification and separation for these forgetting tasks. Specifically, the algorithmic framework (refer to Figure 4) incorporates an annealed gradient ascent and target-aware gradient descent in a dynamical manner. First, it actively unlearns the identified forgetting data, and constructs the contrast information to filter out the remaining data which is hard to be affected. Then, simultaneously learning the selected retaining data with gradient descent deconstructs the entangled feature representation. Ultimately, the learning objective can progressively approach standard retraining using the aligned retaining data (refer to Figure 5). We present comprehensive experiments on different setups of benchmarks and also real-world applications to verify the effectiveness. Our main contributions can be summarized as: Conceptually, we introduce new settings that decouple the class label and the target concept, which

---

[1] $\mathcal{L}_1 \prec \mathcal{L}_2$: For any label $y \in \mathcal{L}_1$, there exists a label $y' \in \mathcal{L}_2$ that an instance labeled with $y$ can also be labeled with $y'$, but not all instances labeled with $y'$ can be labeled with $y$.

investigate the label domain mismatch in class-wise unlearning (in Section 3.1); Empirically, we systematically reveal the challenges of restrictive unlearning with the mismatched label domains, and demonstrate that the representation gravity in forgetting dynamics is critical for achieving the forgetting target in the new tasks (in Section 3.2); Technically, we propose a general framework, namely, *TARF*, to realize the target identification and separation in unlearning. It consists of annealed forgetting and target-aware retaining which collaboratively approximate retraining on the retaining data (in Section 3.3); Experimentally, we conduct extensive explorations to validate the effectiveness of our framework and perform various ablations to characterize algorithm properties (in Section 4).

## 2 PRELIMINARIES

**Problem setup.** Following the literature Shaik et al. (2023); Xu et al. (2023), we mainly consider the multi-class classification as the original training task for class-wise unlearning. Let $\mathcal{X} \subset \mathbb{R}^d$ denote the input space and $\mathcal{Y} = \{1, \ldots, C\}$ denote the label space, where $C$ is the number of classes, the training dataset $\mathcal{D} = \{(x_i, y_i)\}_{i=1}^N$ generally consists of two subsets in machine unlearning, e.g., the forgetting dataset $\mathcal{D}_f$ and the retaining dataset $\mathcal{D}_r = \mathcal{D} \backslash \mathcal{D}_f$. Building upon the model $f_{\theta^*} : \mathcal{X} \to \mathcal{Y}$ trained on $\mathcal{D}$ with the loss function $\ell$, the general goal of this problem is to find an unlearned model $\theta_{un}^*$, which approximates the behaviors of the model $\theta^r$ that retrained on $\mathcal{D}_r$ from scratch,

$$\theta_{un}^* = \arg\min_\theta \frac{1}{|\mathcal{D}|} \sum_{(x,y)\sim\mathcal{D}} \mathcal{R}(\theta, \theta^r, x, y) \quad \text{s.t.} \ \theta^r = \arg\min_\theta \underbrace{\frac{1}{|\mathcal{D}_r|} \sum_{(x,y)\sim\mathcal{D}_r} \ell(f_\theta(x), y)}_{L_{retrain}}, \quad (1)$$

where $\mathcal{R}$ indicates a general risk measure for model behavior consistency (Golatkar et al., 2020; Shaik et al., 2023), which can be instantiated by an averaged gap with various evaluation metrics (Jia et al., 2023; Fan et al., 2023) (e.g., unlearning accuracy (UA), retaining accuracy (RA), and others related to privacy) in experiments to pursue the unlearning efficacy and the model utility (Xu et al., 2023). The specific metrics are introduced in Section 4.1, more discussion on related work is in Appendix A.

**Dataset partition in mismatched setting.** As the target concept is decoupled from the class label, we adopt $\mathcal{D}_t$ to indicate the dataset of the target concept, $\mathcal{D}_f$ to indicate the *given forgetting* dataset, and summarize the notations in Table 1. We can find that the previous assumptions of $\mathcal{D}_f = \mathcal{D}_t$ and

Table 1: Notation summary and training set data partition corresponding to four major forgetting tasks.

| Notation Explanation | Scenario | Data Partition | |
|---|---|---|---|
| $\mathcal{D}_f$ : given forgetting dataset | All matched | $\mathcal{D}_f = \mathcal{D}_t$ | $\mathcal{D}_{un} = \mathcal{D}_r$ |
| $\mathcal{D}_t$ : dataset of target concept | Target mismatch | $\mathcal{D}_f \subset \mathcal{D}_t$ | $\mathcal{D}_{un} = \mathcal{D}_{fr} \cup \mathcal{D}_r$ |
| $\mathcal{D}_{un} := \mathcal{D}\backslash\mathcal{D}_f$, remaining data | Model mismatch | $\mathcal{D}_f = \mathcal{D}_t$ | $\mathcal{D}_{un} = \mathcal{D}_r$ |
| $\mathcal{D}_r, \mathcal{D}_{fr}$ : true/false retaining dataset | Data mismatch | $\mathcal{D}_f \subset \mathcal{D}_t$ | $\mathcal{D}_{un} = \mathcal{D}_{fr} \cup \mathcal{D}_r$ |

$\mathcal{D}_r = \mathcal{D}\backslash\mathcal{D}_f$ only hold in all matched setting. In model mismatch forgetting, the former is still held while we notice that there exists *affected retaining* data in $\mathcal{D}_{ar}$ having the same class label with that in $\mathcal{D}_f$; in target mismatch forgetting and data mismatch forgetting, $\mathcal{D}_f \subseteq \mathcal{D}_t$ and the remaining dataset $\mathcal{D}_{un} = \mathcal{D}\backslash\mathcal{D}_f$ include both true retaining dataset $\mathcal{D}_r \subseteq \mathcal{D}_{un}$ and the *false retaining* dataset $\mathcal{D}_{fr} = \mathcal{D}_t\backslash\mathcal{D}_f$, where the data belong to the target concept but included in the remaining dataset. Considering specific task feasibility, we assume that the number of classes in $\mathcal{D}_{un}$ belonging to the target concept is known in target mismatch forgetting, and the retrained model for every task is trained using $\mathcal{D}_r = \mathcal{D}\backslash\mathcal{D}_t$. More details about unlearning request construction are provided in Appendix D.4.

**Different focus from prior methods.** Assuming that $\mathcal{D}_f = \mathcal{D}_t$ and $\mathcal{D}_r = \mathcal{D}\backslash\mathcal{D}_f$, the common approximation unlearning methods either focus on retaining or forgetting objectives. The former, represented by Fine-tuning (FT) (Warnecke et al., 2023), fine-tunes the model $\theta^o$ on $\mathcal{D}_r$ to induce catastrophic forgetting over $\mathcal{D}_f$. Later advances assign random labels (Golatkar et al., 2020) on $\mathcal{D}_f$ to enforce forgetting or adopt $L_1$-norm (Jia et al., 2023) to infuse weight sparsity in approximation. The latter, represented by gradient ascent (GA), reverse gradient updates on $\mathcal{D}_f$. And another line of works (Izzo et al., 2021) utilizes the influence function (Koh & Liang, 2017) to erase the influence.

## 3 TARF: *TARget-aware Forgetting*

### 3.1 EXPLORING MISMATCHED TAXONOMY IN UNLEARNING

Given its technical nature of mitigating the data influence from a trained model, unlearning is given a broader significance in the context of trustworthiness (Bommasani et al., 2021), where the requests

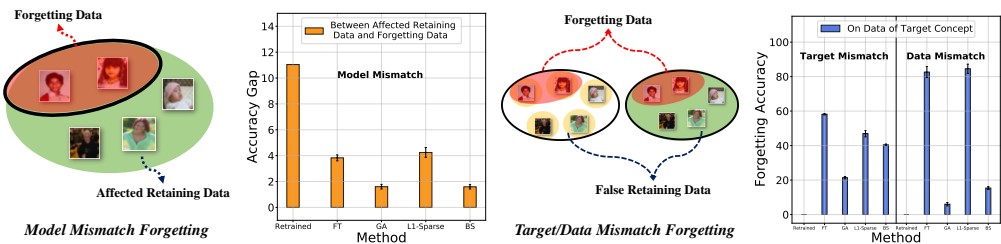

We conduct various unlearning methods for the four tasks. In conventional all-matched forgetting, all the methods can perform similarly to Retrained. In contrast, we can find that model mismatch forgetting can be affected by the trained model, coupling the behaviors on the forgetting and affected retaining data, leaving less accuracy gap between them. In target or data mismatch forgetting, the class labels cannot fully represent the target concepts, leaving false retaining data (belongs to the target concept) not completely forgotten. Full results can refer to Figure 8.

Figure 2: The challenges of restrictive unlearning with the mismatched label domains.

can be varied beyond the withdrawal from data owner (Shah et al., 2023), and may be applied in mitigating bias (Yao et al., 2023) to improve fairness, erasing harmful content (Li et al., 2024) to ensure safety usage, or removing inappropriate content (Gandikota et al., 2023) for social good. Recently, a series of studies (Golatkar et al., 2020; Warnecke et al., 2023; Jia et al., 2023; Fan et al., 2023; Chen et al., 2023) have several proposals on forgetting a training class of the models, and demonstrated it can be successfully achieved by partially scrubbing the class data or fine-tuning on the retaining data to realize catastrophic forgetting (French, 1999; Goodfellow et al., 2015). However, a general scenario considered in previous works is that the target concept is aligned with the taxonomy of the pre-training tasks, which may not always hold in practical scenarios with the previous meanings (du e to the space, we leave more discussion in Appendix D.5). This naturally motivates the question,

*What if the class labels and target concept do not coincide in unlearning?*

In Figure 2, we conduct the unlearning on the four forgetting tasks as instantiated in Figure 1 (full results refer to Figure 8). As a result, those unlearning methods, e.g., the representative FT, GA, and the recent $L_1$-sparse (Jia et al., 2023) and BS (Chen et al., 2023) show different performance gaps compared with the retrained models except in the conventional all matched setting. It can be found that the *affected retaining* data (which is under the same superclass as the model trained on) are entangled with the forgetting part when $\mathcal{L}_T \prec \mathcal{L}_M$, as demonstrated by the less accuracy gap between forgetting and affected retaining data than that of Retrained in the left-middle panel of Figure 2; and the *false retaining* data (which belong to the target concept but are not identified) are under-represented by the given forgetting data when $\mathcal{L}_D \prec \mathcal{L}_T$, as evident by the non-zero accuracy on target concept in the right panel of Figure 2.

## 3.2 SYSTEMATIC EXPLORATION ON FORGETTING DYNAMICS

The mismatch of label domains affects the construction of model representation in unlearning, which requires us to explore it further to understand the underlying mechanism of the performance gaps. We delve into the relationship between the representation and forgetting dynamics, for which we first derive the formal analytical results (a full proof can refer to Appendix C) as follows, and then provide empirical verification in Figure 3 with corresponding interpretations on different kind of mismatch.

**Assumption 3.1** (Representation similarity). *Let $s_1$ and $s_2$ be two disjoint yet semantically related subsets of a dataset $D$ trained on a model $f_\theta$, $x_1 \in s_1$ and $x_2 \in s_2$ refer to samples drawn from them. Given the representation of an input $x$ at an intermediate layer be $h(x)$, the gradient differences at representation level can be controlled by assuming $\ell_h(\cdot)$ is Lipschitz smooth with constant $C_\ell$, then we have $||\nabla\ell_h(x_1) - \nabla\ell_h(x_2)|| \leq C_\ell||h(x_1) - h(x_2)|| = C_\ell d_h(x_1, x_2)$ for a local region.*

**Theorem 3.2** (Gravity effects on forgetting dynamic). *Let $\theta^0$ be the well-trained model parameters for unlearning, and we perform unlearning on $s_1$ via a gradient ascent update, i.e., $\theta^{t+1} = \theta^t + \nabla L_{s_1}(\theta^t)$ for epoch t, then we can the following dynamics given $\Delta L_{s_1,s_2}(\theta^{t+1}) = (L_{s_1}(\theta^{t+1}) - L_{s_2}(\theta^{t+1}))$,*

$$\Delta L_{s_1,s_2}(\theta^{t+1}) \leq (L_{s_1}(\theta^t) - L_{s_2}(\theta^t)) + \eta\lambda_{max}(J_{\theta^t}(x_1))C_\ell\mathbb{E}d_h(x_1, x_2) \cdot ||\nabla L_{s_1}(\theta^t)|| + \mathcal{O}(\eta^2), \quad (2)$$

*where $\lambda_{max}(J_{\theta^t})$ is the largest eigenvalue of the Jacobian matrix $J_\theta = \frac{\partial h(x)}{\partial\theta}$. Note that when $t \to 0$, the RHS mainly relies on the term measuring representation similarity as $(L_{s_1}(\theta^t) - L_{s_2}(\theta^t)) \to 0$.*

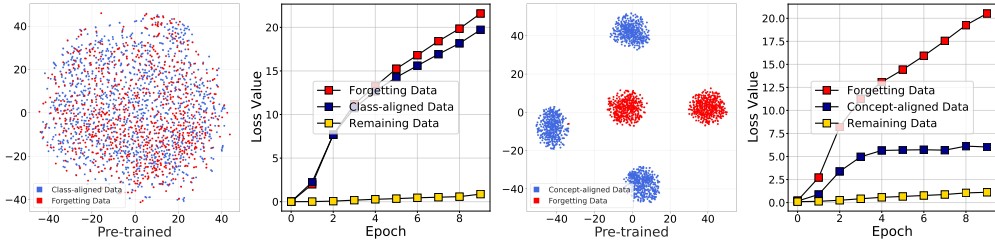

We present the tSNE visualization of the learned features from the pre-trained model trained by (left) superclass and (right) classes. We also show the averaged loss value of forgetting data, concept/class-aligned data, and the remaining data during GA on the two representations. In addition, the cluster-wise instance distance and accuracy dynamics can refer to Figure 9. Note that we only show the 5 classes in tSNE as the large number of remaining classes (e.g., 95 in yellow), we also provide the full results of the unlearned representations in the Appendix F.2.

Figure 3: Forgetting dynamics on entangled/under-entangled feature representations of trained model.

**Remark 3.1.** (Intuitive implication) The Theorem 3.2 connects the unlearning behaviors with the representation-level relationship in forgetting dynamics, specifically on Eq. (2) where the leading term shows the magnitude of loss change can be proportional to their representation distance. Intuitively, if two portions of data occupy nearby/far-apart regions in the latent space, pushing the model to forget the one will inadvertently/loosely affect the other, reflecting a gravity-style co-movement. This idea forms the basis of our later understanding of challenges in mismatch scenarios and our useful cues.

**Target or data mismatch.** In both tasks, we have $\mathcal{L}_D \prec \mathcal{L}_T$, which means that the forgetting data is a subset of the target concept, i.e., $\mathcal{D}_f \subset \mathcal{D}_t$. As indicated in right of Figure 3, partially relying on the forgetting or remaining data can not fully represent the target concept due to under-entangled representation, and leaves non-zero accuracy on false retaining data as shown in Figure 2.

**Remark 3.2.** (Insufficient representation) Given $\mathcal{L}_D \prec \mathcal{L}_T$ that indicates $\mathcal{D}_f \subset \mathcal{D}_t$, we can have $\mathcal{D}_f$ as $s_1$ and $\mathcal{D}_t \backslash \mathcal{D}_f$ as $s_2$ corresponds to Theorem 3.2, in which the sample $(x^u, y^u) \sim s_2$ and sample $(x, y) \sim s_1$ exhibit weak gravity effects on the forgetting dynamics due to the under-entangled or biased representation with large latent distance $d_h(x^u, x)$, i.e., the $\Delta L_{s_1,s_2}(\theta^{t+1}) - \Delta L_{s_1,s_2}(\theta^t)$ in Eq. 2 indicating loss update gap of two subsets can be also relatively large, that aligns with Figure 3. The forgetting set covers only part of target concept can't govern the whole concept forgetting.

**Model mismatch forgetting.** In this task, we have $\mathcal{L}_D = \mathcal{L}_T$ while $\mathcal{L}_T \prec \mathcal{L}_M$. Regarding the model trained by the superclass, it can be found in the left of Figure 3 that the features of forgetting data and affected retaining data are closely entangled, showing that the unlearning of the forgetting data can unavoidably affect the representation of the other part. In contrast, it is also notable in the left-middle of Figure 2 that the accuracy gap between forgetting data and affected retaining data is expected to be large in the retrained reference. We provide the following interpretation based on Theorem 3.2.

**Remark 3.3.** (Decomposition lacking) Given $\mathcal{L}_T \prec \mathcal{L}_M$ that indicates the broader representation region for $\mathcal{D}_z := \mathcal{D}_t \cup \mathcal{D}_{ar}$ within the same class $z$ (here $\mathcal{D}_{ar} \subset \mathcal{D}_r$ refer to the set of affected retaining data as illustrated in left-most of Figure 2), the entangled representation results in small latent distance $d_h(x^u, x)$ for the samples of $(x^u, y^u) \sim \mathcal{D}_z$ and sample $(x, y) \sim \mathcal{D}_t$, so $\Delta L_{s_1,s_2}(\theta^{t+1}) - \Delta L_{s_1,s_2}(\theta^t)$ is as small as evident in left of Figure 3, requiring bidirectional operation to disentangle it, as the representation is overly entangled that forgetting updates on target concept may spill over onto others.

**Forgetting dynamics with representation distance.** Despite the issues revealed by the observations under label domain mismatch, the forgetting performance varying obviously on different representations also provides clues on addressing them. Notably, we can find that GA achieves better forgetting efficacy on the data mismatch forgetting as the feature representation of the forgetting data and false retaining data is entangled. Through the effect of actively forgetting the given data on the other parts of data, we can also utilize the representation gravity defined as follow to identify false retaining data,

**Definition 3.3** (Representation gravity). Given the empirically supported gravity effects in Theorem 3.2, we can have $I_{con}(x, y, \theta)$ to reflect the similarity $d(x^u, x)$ in the model $\theta^t$ with a small $t$, e.g., $I_{con}(x, y, \theta) = |\ell(f_\theta(x), y) - \ell(f_{\theta^t}(x), y)|$, or we can calculate class-wise accuracy change.

It is empirically demonstrated in Figure 3 (also the latent representation distance and class accuracy trends in Figure 9), the corresponding changes in accuracy and loss values show that generally the smaller the distance in representation level, the similar forgetting dynamics the model would have on prediction. Regarding the issues of insufficient representation and decomposition missing, we can

utilize the gravity effects to identify the unidentified forgetting data in the remaining set, and reveal the needs of deconstructing entangled representation by simultaneously considering two parts.

## 3.3 ALGORITHM FRAMEWORK OF TARF

Based on the intuition, we introduce the whole framework of *TARget-aware Forgetting* (TARF), to enable the mismatched class-wise unlearning tasks. Given the identified forgetting data, we illustrate the overall process in Figure 4, and introduce its dynamic learning objective as follows:

$$L_{\text{TARF}} = \underbrace{k(t) \cdot \left( -\frac{1}{|\mathcal{D}_{\text{f}}|} \sum_{(x,y) \sim \mathcal{D}_{\text{f}}} \ell(f(x), y) \right)}_{\textbf{Annealed Forgetting } L_{\text{f}}(k)} + \underbrace{\frac{1}{|\mathcal{D}_{\text{un}}|} \sum_{(x,y) \sim \mathcal{D}_{\text{un}}} \ell(f(x), y) \cdot \tau(x, y, t)}_{\textbf{Target-aware Retaining } L_{\text{u}}(\tau)}, \quad (3)$$

where $k(t)$ serves as an annealing strategy to control the strength of the forgetting part. Along with training, we expect the overall objective to approximate the retraining ones $L_{\text{TARF}} \to L_{\text{retrain}}$ through,

$$L_{\text{f}}(k) \xrightarrow{t \to T} 0, \quad L_{\text{u}}(\tau) \xrightarrow{t \to T} L_{\text{retrain}}, \quad (4)$$

given the initially provided forgetting data $\mathcal{D}_{\text{f}}$ and the remaining set $\mathcal{D}_{\text{un}}$. Specifically, we design the two dynamic hyperparameters $k(t)$ and $\tau(x, y, t)$ as follows,

$$k(t) = \max\left[ \frac{k \cdot (T - t - t_0)}{T}, 0 \right], t \in [0, T]; \tau(x, y, t) = \begin{cases} 0 & I_{\text{con}}(x, y, \theta_{t_1}) > \beta \text{ or } t < t_1, \\ 1 & I_{\text{con}}(x, y, \theta_{t_1}) < \beta \text{ and } t \geq t_1, \end{cases} \quad (5)$$

where $T$ indicates the total training time (e.g., epochs), and the value of $k(t)$ decreases with the training process, $\beta$ can be estimated by the information about the specific unlearning request and the rank of loss/accuracy change (e.g., setting the threshold $\beta$ as the lowest value of top-10% data with in a descending order, to select the most influenced part) at $t_1$, $t_0$ and $t_1$ respectively control the end time of active forgetting and the begin time of retaining part. The whole process can refer to Figure 4 for an intuitive understanding how the previous objective controlled by $k$ and $\tau$ organically consists of three phases to tackling the revealed challenges in mismatched unlearning, and we also provide a functionality explanation about those factors in Appendix E and guidance based on empirical results.

**Phase I: Target Identification.** Before $t_1$, since $\tau(x, y, t) = 0$, Eq. 3 can be formalized as, $L_{\text{TARF-Phase-I}} = k(t) \cdot \left( -\frac{1}{|\mathcal{D}_{\text{f}}|} \sum_{(x,y) \sim \mathcal{D}_{\text{f}}} \ell(f(x), y) \right)$, in which the retaining part is waiting for the dynamic information revealed by this phase. As shown in Figure 3, the false retaining data in $\mathcal{D}_{\text{fr}}$ can be identified due to the similar forgetting dynamics with the forgetting data. We utilize the class label information in our main tasks as it is also available for unlearning. We can obtain the accuracy drop of each class and estimate $\beta$ (refer to Appendix E for details). In Figure 5(a), we show the selected classes in accuracy drop and identification efficacy. Specif-

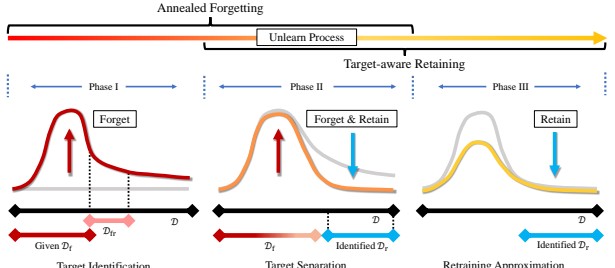

The overall framework consists of two objective parts, e.g., annealed forgetting and target-aware retaining, which can be regarded as three phases to enable all the class-wise unlearning tasks through the view of the unlearning process. (a) Phase I utilizes the gradient ascent to construct dynamic information for all class data; (b) Phase II simultaneously considers gradient ascent on forgetting data and gradient descent on remaining data that is hard to affect to separate target concept; (c) Phase III conducts gradient descent on the selected data to approximate the retraining.

Figure 4: Overview of the proposed framework TARF.

ically, the left shows that classes belonging to the target concept (blue) experience a significantly larger accuracy drop than the remaining classes (yellow), which serves as an indicator for target identification; the right presents the performance with different amounts of given forgetting classes.

**Phase II: Target Separation.** After phase I, the retaining part is engaged with the forgetting part with the identified data $\mathcal{D}_{\text{fr}}$ and the remaining retaining data $\mathcal{D}_{\text{r}}$. By simultaneously considering the forgetting and retaining part as Eq. 3, $L_{\text{TARF-Phase-II}}$ encourages the model to deconstruct the target concept and reconstruct the feature representation of the retaining part, which can effectively decouple the entangled feature in the model mismatch forgetting. In the first panel of Figure 5(b), we compare the accuracy gap on RA and UA, which indicates the success (refer the dashed line of

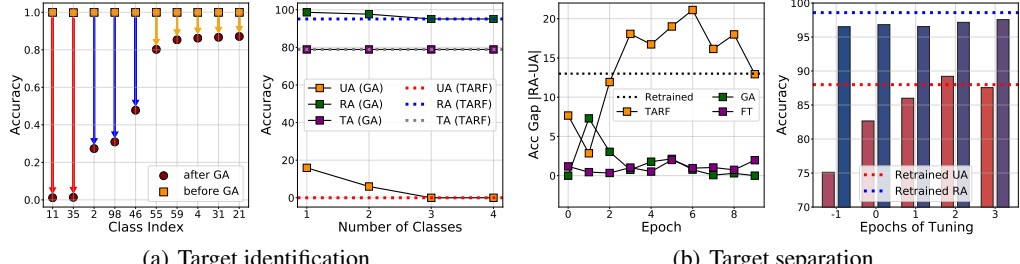

(a) Target identification                        (b) Target separation

We show (a) accuracy changes in target identification, and unlearning performance in data mismatch forgetting; (b) accuracy gap of retaining and forgetting part of the same class, as well as the need of reconstruction.

Figure 5: Target identification and target separation for unlearning under mismatch.

Retrained reference) of disentanglement. It validates the rationality of our method, which jointly applies gradient ascent and decent to deconstruct the entangled representation, achieving the expected accuracy gap (e.g., isolating the target concept with affected retaining data as shown in Figure 2).

**Phase III: Retraining Approximation.** After $t_0$, we focus on retaining in the current phase, which approximates the retraining objective as follows, $L_{\text{TARF-Phase-III}} = \frac{1}{|\mathcal{D}_{\text{un}}|} \sum_{(x,y)\sim\mathcal{D}_{\text{un}}} \ell(f(x), y) \cdot \tau(x, y, t)$, where we use $\tau$ at $t_1$ to indicate the identified hard-to-effect retaining data, and continually reconstruct the representations. Since the general goal of unlearning considered in our work is similar to retraining, this phase can prevent excessive forgetting. In the second panel of Figure 5(b), we compare the performance using different lengths of this phase to show retrain approximation. Note that in Phase-II, our TARF may induce over-deconstruction (larger Acc Gap than that of the dashed line for Retrained reference), so it demonstrates the necessity of our Phase-III focusing purely on retraining to approximate the Retrained reference by using different epochs of this stage.

**Remark 3.3.** Note that the three-phase are interpreted from a unified framework rather than an ad-hoc pipeline. Each phase builds on the previous insights: Phase I identifies potential forgetting targets, Phase II separates entangled representation, and Phase III approximates retraining on those generalizable knowledge. The whole process enables a flexible framework for all mismatch scenarios.

## 4 EXPERIMENTS

### 4.1 EXPERIMENTAL SETUP

**Datasets and models.** In our experiments, we mainly explore unlearning for conventional image classification tasks. To build an easy-to-adopt testbed for our new settings, we adopt the benchmarked datasets, e.g., CIFAR-10/CIFAR-100 (Krizhevsky, 2009) with their superclass information (refer to Tables 13, 14 and 15) in the main experiments. Note the coarse-to-fine label structure of CIFAR-10 is obtained by grouping based on semantic proximity (Dhakad et al., 2024) to enable the controllable experiments. We train two models based on the original classes and its superclass respectively, and instantiate four tasks (as illustrated in Figure 1). More details are summarized in Appendix D.4. Following (Jia et al., 2023; Fan et al., 2023), we use ResNet-18 (He et al., 2016) as the main architecture to obtain original models with standard learning, and then set it to be the basis for unlearning. And we also adopt TinyImageNet and ImageNet (Krizhevsky et al., 2012) for large-scale experiments, and adopt ImageNette (Howard, 2019) and TOFU (Maini et al., 2024) for case studies.

**Evaluation metrics.** The general goal of unlearning considered in this work is to approximate the Retrained model. To give a comprehensive evaluation, we adopt 5 specific evaluation metrics in classification tasks following previous works (Jia et al., 2023; Fan et al., 2023). We utilize Unlearning Accuracy (UA) to evaluate the accuracy of the unlearning targeted subset; Retaining Accuracy (RA) to evaluate the accuracy of the retaining subset; Testing Accuracy (TA) to evaluate the generalization ability of the model; Membership Inference Attack (MIA) to evaluate the efficacy of unlearning by the confidence-based predictor. Note that any single indicator does not represent optimally in the approximation of a Retrained reference. All the above will be compared with that of the Retrained model and summarized in a "Gap" value (averaged gap with Retrained, i.e., $\frac{1}{4}\sum |\mathcal{R}_{\theta_{\text{un}}} - \mathcal{R}_{\theta^{\text{r}}}|$) to indicate the overall performance (the lower the better), and we also adopt TIME to present the computational time. Detailed evaluations of different scenarios are provided in Appendix B.2.

Table 3: Main Results (%). All are trained on the same backbone and initialization (except for the reference Retrained from scratch). Bold numbers are superior results, and we also indicate the second-best results of "Gap" for readability, ↓ indicates smaller values are better (Complete results with mean and std values in Appendix F.7). Overall TARF performs well across various tasks.

| Type / $\mathcal{D}$ | Dataset | CIFAR-10 | | | | | | CIFAR-100 | | | | | |
|---|---|---|---|---|---|---|---|---|---|---|---|---|---|
| | Method / Metrics | UA | RA | TA | MIA | Gap↓ | TIME↓ | UA | RA | TA | MIA | Gap↓ | TIME↓ |
| **All matched** | Retrained (Ref.) | 0.00 | 99.51 | 94.69 | 100.00 | - | 43.3 | 0.00 | 97.85 | 76.03 | 100.00 | - | 43.2 |
| | FT (Warnecke et al., 2023) | 1.07 | 98.62 | 92.36 | 100.00 | 1.07 | 4.43 | 0.67 | 96.32 | 72.34 | 100.00 | 1.47 | 5.02 |
| | RL (Toneva et al., 2018) | 4.13 | 97.65 | 91.23 | 100.00 | 2.36 | 4.88 | 1.00 | 96.09 | 72.00 | 100.00 | 1.70 | 4.96 |
| | GA (Ishida et al., 2020) | 0.49 | 95.24 | 88.17 | 99.78 | 2.88 | **0.25** | 1.33 | 94.74 | 68.56 | 99.89 | 3.01 | **0.06** |
| | IU (Izzo et al., 2021) | 0.22 | 88.15 | 82.38 | 99.96 | 5.99 | 0.45 | 0.00 | 37.61 | 29.58 | 100.00 | 26.67 | 0.51 |
| | BS (Chen et al., 2023) | 25.04 | 87.94 | 80.90 | 88.67 | 15.43 | 0.82 | 4.60 | 90.18 | 63.66 | 99.55 | 6.27 | 0.78 |
| | $L_1$-sparse (Jia et al., 2023) | 0.00 | 94.20 | 89.77 | 100.00 | 2.56 | 4.39 | 0.00 | 94.60 | 71.57 | 100.00 | 1.93 | 4.39 |
| | SalUn (Fan et al., 2023) | 0.00 | 91.32 | 86.87 | 100.00 | 4.00 | 5.65 | 0.00 | 75.34 | 62.14 | 100.00 | 9.10 | 5.75 |
| | SCRUB (Kurmanji et al., 2023) | 0.00 | 99.94 | 91.00 | 100.00 | 1.03 | 2.88 | 0.00 | 99.98 | 76.75 | 100.00 | **0.71** | 3.23 |
| | **TARF** (ours) | 0.00 | 98.23 | 91.95 | 100.00 | **1.01** | 4.21 | 0.00 | 96.90 | 72.53 | 100.00 | 1.11 | 4.68 |
| **Model mismatch** | Retrained (Ref.) | 87.76 | 99.58 | 95.91 | 20.57 | - | 43.8 | 88.22 | 98.58 | 78.50 | 25.78 | - | 43.8 |
| | FT (Warnecke et al., 2023) | 94.67 | 98.53 | 93.56 | 9.56 | 5.33 | 4.29 | 92.67 | 95.02 | 79.34 | 16.33 | 4.58 | 4.86 |
| | RL (Toneva et al., 2018) | 53.69 | 97.85 | 92.39 | 96.60 | 28.84 | 4.82 | 80.11 | 95.83 | 79.83 | 99.00 | 21.35 | 4.93 |
| | GA (Ishida et al., 2020) | 5.76 | 86.99 | 82.20 | 94.98 | 45.68 | **0.25** | 6.78 | 94.83 | 76.96 | 97.78 | 39.68 | **0.06** |
| | IU (Izzo et al., 2021) | 23.69 | 87.34 | 82.57 | 89.87 | 39.74 | 0.44 | 34.67 | 96.83 | 79.08 | 86.44 | 29.14 | 0.49 |
| | BS (Chen et al., 2023) | 10.29 | 50.77 | 49.39 | 95.96 | 62.05 | 0.79 | 18.11 | 95.90 | 72.28 | 95.22 | 37.14 | 0.89 |
| | $L_1$-sparse (Jia et al., 2023) | 93.11 | 94.76 | 91.63 | 14.44 | 5.15 | 4.24 | 90.22 | 94.78 | 78.81 | 18.88 | 3.25 | 5.00 |
| | SalUn (Fan et al., 2023) | 8.91 | 93.95 | 84.38 | 99.32 | 43.69 | 6.04 | 66.33 | 78.83 | 70.78 | 77.00 | 25.15 | 5.97 |
| | SCRUB (Kurmanji et al., 2023) | 95.14 | 99.81 | 94.22 | 15.38 | 3.61 | 3.06 | 91.44 | 99.74 | 79.23 | 21.11 | 2.45 | 4.12 |
| | **TARF** (ours) | 91.11 | 97.49 | 92.49 | 17.82 | **2.90** | 4.31 | 86.67 | 97.05 | 80.07 | 26.00 | **1.21** | 4.81 |
| **Target mismatch** | Retrained (Ref.) | 0.00 | 99.38 | 93.85 | 100.00 | - | 52.1 | 0.00 | 97.85 | 73.72 | 100.00 | - | 53.2 |
| | FT (Warnecke et al., 2023) | 50.43 | 98.47 | 91.65 | 50.44 | 25.78 | 4.38 | 58.18 | 96.32 | 72.53 | 46.76 | 28.54 | 5.00 |
| | RL (Toneva et al., 2018) | 51.25 | 97.56 | 90.90 | 56.23 | 24.95 | 4.79 | 58.89 | 96.05 | 72.20 | 46.98 | 28.81 | 4.93 |
| | GA (Ishida et al., 2020) | 40.82 | 97.01 | 89.51 | 64.32 | 20.80 | 0.26 | 58.04 | 96.64 | 70.22 | 90.67 | 8.86 | 0.05 |
| | IU (Izzo et al., 2021) | 44.51 | 88.07 | 81.80 | 58.73 | 27.29 | 0.44 | 30.62 | 37.19 | 29.58 | 63.69 | 42.93 | 0.50 |
| | BS (Chen et al., 2023) | 53.62 | 88.65 | 75.39 | 76.33 | 26.62 | 0.82 | 34.67 | 98.32 | 68.66 | 85.16 | 15.20 | 0.97 |
| | $L_1$-sparse (Jia et al., 2023) | 49.47 | 93.61 | 88.83 | 51.24 | 27.26 | 4.38 | 56.09 | 94.63 | 72.00 | 48.04 | 28.25 | 4.78 |
| | SalUn (Fan et al., 2023) | 46.63 | 91.08 | 86.31 | 60.94 | 25.38 | 5.90 | 59.64 | 75.52 | 62.37 | 65.96 | 27.35 | 5.81 |
| | SCRUB (Kurmanji et al., 2023) | 49.98 | 99.94 | 92.10 | 50.18 | 25.53 | 2.89 | 59.64 | 99.99 | 75.32 | 44.89 | 29.90 | 3.52 |
| | **TARF** (ours) | 0.06 | 97.57 | 90.81 | 100.00 | **1.23** | 4.23 | 0.31 | 97.35 | 73.68 | 100.00 | **0.21** | 4.85 |
| **Data mismatch** | Retrained (Ref.) | 0.00 | 99.54 | 95.56 | 100.00 | - | 52.1 | 0.00 | 98.50 | 80.15 | 100.00 | - | 53.2 |
| | FT (Warnecke et al., 2023) | 96.79 | 98.49 | 93.26 | 6.48 | 48.41 | 4.32 | 82.62 | 95.66 | 79.77 | 37.24 | 37.15 | 4.93 |
| | RL (Toneva et al., 2018) | 76.47 | 97.68 | 91.93 | 49.81 | 33.04 | 4.76 | 89.78 | 96.82 | 79.90 | 70.76 | 30.49 | 4.97 |
| | GA (Ishida et al., 2020) | 8.69 | 96.41 | 90.78 | 93.03 | 5.89 | 0.25 | 6.00 | 97.65 | 79.23 | 98.04 | 2.43 | 0.05 |
| | IU (Izzo et al., 2021) | 22.84 | 95.50 | 89.54 | 88.57 | 11.08 | 0.44 | 31.51 | 98.96 | 78.20 | 88.09 | 11.46 | 0.48 |
| | BS (Chen et al., 2023) | 16.70 | 61.21 | 49.76 | 92.24 | 22.37 | 0.82 | 15.38 | 98.50 | 72.28 | 96.22 | 6.76 | 0.96 |
| | $L_1$-sparse (Jia et al., 2023) | 95.76 | 94.31 | 91.08 | 9.52 | 48.99 | 4.78 | 88.31 | 94.91 | 79.02 | 22.49 | 42.64 | 5.03 |
| | SalUn (Fan et al., 2023) | 51.77 | 93.87 | 90.46 | 63.52 | 24.75 | 5.72 | 72.93 | 78.87 | 71.04 | 54.13 | 36.89 | 5.72 |
| | SCRUB (Kurmanji et al., 2023) | 97.13 | 99.89 | 95.03 | 10.99 | 46.76 | 2.94 | 95.50 | 99.79 | 79.68 | 15.11 | 45.54 | 3.68 |
| | **TARF** (ours) | 0.00 | 98.17 | 93.09 | 100.00 | **0.96** | 4.22 | 0.00 | 95.01 | 78.98 | 100.00 | **1.17** | 4.78 |

## 4.2 PERFORMANCE EVALUATION

In this part, we present the main comparison results with those considered as baselines in the four unlearning tasks. We also report results under multiple runs in Appendix F.7 with std values.

**In conventional benchmarks**, all the retrained models (termed Retrained) are trained with the fully aligned retaining data. In Table 3, we can find the previous unlearning methods achieved satisfactory performance in conventional all matched forgetting, but did not perform well on the other three newly considered tasks with the label domain mismatch. Note that UA of Retrained (Ref.) in the model mismatch scenario is not equal to 0 since it is evaluated with superclass label. Specifically, since the previous methods partially rely on forgetting data or remaining data, it results in ineffective or excessive forgetting due to the insufficient representation or decomposition missing. For example, FT can retain a similar RA with the Retrained but be less effective in forgetting, while GA reaches the lowest UA across different tasks but sacrifices too much performance on the retaining dataset. In contrast, TARF can generally perform better (or comparable with the best method). We also present Table 2 to show a fine-grained evaluation on unlearning target within superclass in model mismatch.

Table 2: Fine-grained evaluation on superclass.

| Model Mismatch | Method | UA-F | UA-R | RA | TA | MIA | Gap |
|---|---|---|---|---|---|---|---|
| **CIFAR-10** (UA-F: automobile, UA-R: truck) | Retrained (Ref.) | 77.48 | 98.04 | 99.58 | 95.91 | 20.57 | – |
| | FT | 92.09 | 97.25 | 98.53 | 93.56 | 9.56 | 5.96 |
| | RL | 48.69 | 58.69 | 97.85 | 92.39 | 96.60 | 29.88 |
| | GA | 0.00 | 11.52 | 86.99 | 82.20 | 94.98 | 52.94 |
| | BS | 7.79 | 12.45 | 50.77 | 49.39 | 95.96 | 65.20 |
| | $L$-sparse | 91.40 | 94.82 | 94.76 | 91.63 | 14.44 | 6.47 |
| | SCRUB | 91.07 | 99.21 | 99.81 | 94.22 | 15.38 | 4.37 |
| | **TARF** (ours) | 85.24 | 96.98 | 97.49 | 92.49 | 17.82 | **3.42** |
| **CIFAR-100** (UA-F: boy,girl; UA-R: man,woman,baby) | Retrained (Ref.) | 77.56 | 95.25 | 98.58 | 78.50 | 25.78 | – |
| | FT | 90.33 | 94.23 | 95.02 | 79.34 | 16.33 | 5.53 |
| | RL | 74.04 | 84.16 | 95.83 | 79.83 | 99.00 | 18.38 |
| | GA | 5.64 | 7.54 | 94.83 | 76.96 | 97.78 | 47.38 |
| | BS | 17.00 | 18.85 | 95.90 | 72.28 | 95.22 | 43.06 |
| | $L$-sparse | 86.69 | 92.58 | 94.78 | 78.81 | 18.88 | 4.56 |
| | SCRUB | 81.26 | 98.23 | 99.74 | 79.23 | 21.11 | 2.65 |
| | **TARF** (ours) | 74.70 | 94.65 | 97.05 | 80.07 | 26.00 | **1.36** |

**For verification on large-scale datasets**, we evaluate the method on Tiny-ImageNet and ImageNet-1k with lager models. Due to the space, we show the results on ImageNet-1k in main text, and leave other results in Appendix F.5, and forgetting multiple classes in Appendix F.9. It shows that our TARF can achieve satisfactory performance with respect to the overall gap with Retrained references.

Table 4: Results (%). Comparison with the unlearning baselines on ImageNet-1k. All matched forgetting: unlearn 1 class; Target mismatch forgetting: unlearn three classes belonging to "fish".

| Type / $\mathcal{D}$ | Dataset | All matched | | | | | | Target mismatch | | | | | |
|---|---|---|---|---|---|---|---|---|---|---|---|---|---|
| | Method / Metrics | UA | RA | TA | MIA | Gap↓ | TIME↓ | UA | RA | TA | MIA | Gap↓ | TIME↓ |
| | Retrained (Ref.) | 0.00 | 79.77 | 77.64 | 100.00 | - | 7075.48 | 0.00 | 80.09 | 77.54 | 100.00 | - | 7777.54 |
| | FT (Warnecke et al., 2023) | 0.00 | 70.18 | 71.98 | 100.00 | 3.82 | 608.11 | 0.79 | 70.26 | 72.07 | 100.00 | 4.02 | 608.62 |
| | RL (Toneva et al., 2018) | 81.38 | 70.22 | 71.79 | 19.46 | 44.29 | 969.44 | 79.69 | 69.98 | 71.77 | 23.03 | 43.14 | 972.02 |
| | GA (Ishida et al., 2020) | 0.00 | 66.25 | 67.36 | 100.00 | 5.95 | 8.76 | 0.00 | 31.21 | 37.74 | 0.00 | 47.17 | 17.38 |
| | BS (Chen et al., 2023) | 0.00 | 31.15 | 36.33 | 100.00 | 22.48 | 9.03 | 0.00 | 21.57 | 27.56 | 99.97 | 27.13 | 23.75 |
| | $L_1$-sparse (Jia et al., 2023) | 0.00 | 67.98 | 70.70 | 100.00 | 4.68 | 603.21 | 0.00 | 67.24 | 70.28 | 100.00 | 5.03 | 601.27 |
| | SCRUB (Kurmanji et al., 2023) | 29.77 | 74.92 | 75.66 | 81.77 | 13.71 | 655.42 | 22.44 | 74.87 | 75.60 | 82.77 | 11.71 | 681.53 |
| | **TARF** (ours) | 0.00 | 70.53 | 72.23 | 100.00 | **3.66** | 600.11 | 0.00 | 69.93 | 71.79 | 100.00 | **3.97** | 628.87 |
| **ImageNet-1k** | Dataset | Model matched | | | | | | Data mismatch | | | | | |
| | Method / Metrics | UA | RA | TA | MIA | Gap↓ | TIME↓ | UA | RA | TA | MIA | Gap↓ | TIME↓ |
| | Retrained (Ref.) | 79.15 | 80.00 | 70.29 | 25.69 | - | 6501.27 | 0.00 | 80.36 | 70.38 | 100.00 | - | 6493.16 |
| | FT (Warnecke et al., 2023) | 83.31 | 70.38 | 64.05 | 19.00 | 6.68 | 695.42 | 0.00 | 69.99 | 63.76 | 100.00 | 4.24 | 693.18 |
| | RL (Toneva et al., 2018) | 87.62 | 69.43 | 63.26 | 15.23 | 9.13 | 959.84 | 88.21 | 70.33 | 63.81 | 12.21 | 48.15 | 956.13 |
| | GA (Ishida et al., 2020) | 0.00 | 66.62 | 58.91 | 100.00 | 44.56 | 17.44 | 0.00 | 15.35 | 14.34 | 0.00 | 55.26 | 17.58 |
| | BS (Chen et al., 2023) | 0.00 | 45.81 | 40.84 | 100.00 | 54.28 | 19.69 | 0.00 | 13.00 | 12.10 | 100.00 | 31.41 | 23.70 |
| | $L_1$-sparse (Jia et al., 2023) | 82.00 | 67.94 | 62.58 | 19.15 | 7.29 | 1091.29 | 0.00 | 66.37 | 61.03 | 100.00 | 5.84 | 1071.41 |
| | SCRUB (Kurmanji et al., 2023) | 86.08 | 74.82 | 68.04 | 14.69 | 6.34 | 663.61 | 14.18 | 74.84 | 67.92 | 93.10 | 7.27 | 689.82 |
| | **TARF** (ours) | 80.62 | 70.27 | 64.04 | 19.46 | **5.92** | 601.28 | 0.00 | 70.10 | 63.97 | 100.00 | **4.17** | 602.62 |

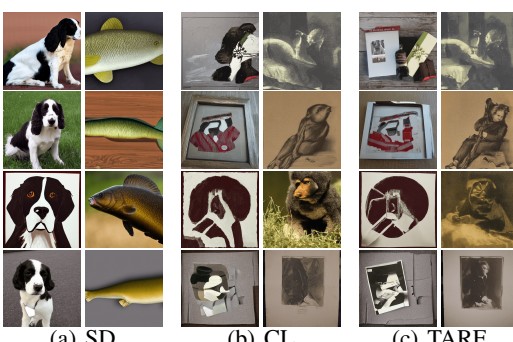

| | | | |
|---|---|---|---|
| (a) SD | (b) CL | (c) TARF | |

Figure 6: Application on data mismatch concept removal of image generation with stable diffusion. Full results with more Tables are in Appendix E.3.

Table 5: Application on information removal on LLM with TOFU dataset for real-world application. More discussions are in Appendix F.8.

| TOFU | Setting/Request Metric | All-matched | | Target Mismatch | |
|---|---|---|---|---|---|
| | | QA Prob on F. (↓) | QA Prob on R. (↑) | QA Prob on F. (↓) | QA Prob on R. (↑) |
| | GA | **0.0009** | 0.1604 | **0.0000** | 0.0000 |
| | **TARF** (GA) | 0.0198 | 0.3218 | 0.1756 | **0.4301** |
| | NPO | 0.0792 | 0.6824 | **0.0095** | 0.0104 |
| | **TARF** (NPO) | **0.0762** | **0.6977** | 0.2597 | 0.4343 |
| **LLama3.2 1B-instruct** | Setting/Request Metric | Representation Mismatch | | Data Mismatch | |
| | | QA Prob on F. (↓) | QA Prob on R. (↑) | QA Prob on F. (↓) | QA Prob on R. (↑) |
| | GA | **0.0000** | 0.0000 | **0.0048** | 0.1768 |
| | **TARF** (GA) | **0.0000** | 0.4034 | 0.1101 | **0.5942** |
| | NPO | 0.0074 | 0.0105 | 0.2482 | **0.6856** |
| | **TARF** (NPO) | 0.1421 | **0.3881** | 0.1238 | 0.6530 |
| | Setting/Request Metric | All-matched | | Target Mismatch | |
| | | QA Prob on F. (↓) | QA Prob on R. (↑) | QA Prob on F. (↓) | QA Prob on R. (↑) |
| | GA | **0.0002** | 0.1814 | **0.0000** | 0.0000 |
| | **TARF** (GA) | 0.0016 | **0.4730** | 0.1716 | **0.4854** |
| | NPO | 0.0080 | 0.4924 | **0.0000** | 0.0000 |
| | **TARF** (NPO) | 0.0113 | **0.6209** | 0.2703 | **0.5643** |
| **LLama3.2 8B-instruct** | Setting/Request Metric | Representation Mismatch | | Data Mismatch | |
| | | QA Prob on F. (↓) | QA Prob on R. (↑) | QA Prob on F. (↓) | QA Prob on R. (↑) |
| | GA | **0.0000** | 0.0000 | 0.0296 | 0.1826 |
| | **TARF** (GA) | **0.0000** | 0.4839 | **0.0038** | 0.3909 |
| | NPO | **0.0000** | 0.0000 | 0.1274 | 0.4949 |
| | **TARF** (NPO) | 0.0987 | **0.5630** | **0.0201** | **0.5994** |

**For case study on real-world application**, we apply our TARF in the scenario of concept removal with stable-diffusion (Ho et al., 2020) and personal information removal with LLama3.2 (Grattafiori et al., 2024). Considering the practical data mismatch forgetting on where users report some undesirable examples to represent the unwanted concept, we show the efficacy of unlearning the "springer" and "tench" in Figure 6. We also constructing the four similar mismatch scenarios using TOFU dataset. Due to the limited space, we leave more details in Appendixes E.3 and F.8.

## 4.3 ABLATIONS AND FURTHER EXPLORATION

In this part, we provide further exploration of the three class-wise unlearning tasks and conduct various ablation studies to characterize TARF. More results and discussions are provided in Appendix F.

**Weighted control in annealed gradient ascent.** To analyze the annealed gradient ascent, we present the results on the left of Figure 7 to show the effects of initialized strength $k$ on the all matched setting (results on other settings can refer to Figure 17) using the CIFAR-100 dataset. The results show that a proper $k$ (e.g., about 0.05) can achieve a satisfactory performance. However, the larger $k$ results in lower retaining performance and higher Gap value as the strength increases feature deconstruction. For the hyperparameters, we discuss the computational stability from a functionality understanding and also synthesize a practical guideline in Appendix E.1 with the empirical results of ablation study.

**Constant or dynamic gradient ascent for forgetting.** In the middle-left of Figure 7, we study whether we need the learning-rate-reduced $k$ for the forgetting part. Specifically, we compare it with using constant $k$ and learning-rate-increased $k$ on two model mismatch forgetting tasks. The results demonstrate that annealed gradient ascent can achieve more similar performance with the Retrained on forgetting data. The gradient ascent is considered simultaneously with gradient descent for restricting the forgetting region, while we adopt the annealed one since the unlearning target is to approximate the retrained model instead of continually maximizing the loss of forgetting data.

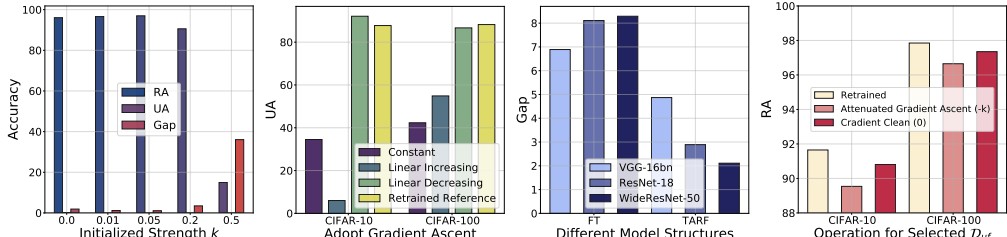

Figure 7: Ablation studies: *Left:* performance using different initialized $k$ on all matched forgetting; *middle-left:* effects of constant or different dynamic gradient ascent controlled by $k(t)$; *middle-right:* comparison of forgetting with different model structures; *right:* comparison of using different operations on the selected forgetting data. More experimental details can refer to Appendix F.

**Unlearning on models trained by different structures.** In the middle-right of Figure 7, we investigate unlearning with different structures, e.g., ResNet-18 (He et al., 2016), VGG-16bn (Simonyan & Zisserman, 2015), and WideResNet-50 (Zagoruyko & Komodakis, 2016). The results on the model mismatch forgetting demonstrate that TARF can achieve the lower performance gap than FT, evaluated with the retrained reference. With the increasing model capacity on the original training tasks, we can also find the model with a smaller capacity makes it harder to decompose the entangled feature representation for achieving the unlearning target, which increases the representation complexity.

**Different operations on the selected forgetting data.** In the right of Figure 7, we present the ablation on the specific gradient operation on the identified false retaining data $\mathcal{D}_{\text{fr}}$. We compare using the gradient ascent ($-k(t)$) and cleaning (0) with the Retrained reference in target mismatch forgetting. Except for the similar forgetting efficacy achieved by the three trials, major differences exist in the performance evaluated by RA. The results show that gradient cleaning may be a better choice for $\mathcal{D}_{\text{fr}}$ to not deconstruct the features too much and affect the retaining accuracy.

**Broader explorations of unlearning with TARF.** Beyond the performance comparison and ablation on major benchmarks, we also conduct broader exploration on our TARF to give a balance view on the unlearning capabilities. Specifically, we also investigate the performance robustness under varied false-retaining set size for quantile-choice in Appendix E; discuss and check the computational cost of TARF in target identification stage in Appendix E.2; verify the robustness of TARF under the weakly-supervised scenario or more challenging multiple concept unlearning scenarios in Appendix F.

## 5 CONCLUSION

In this work, we decouple the class label and target concept in class-wise unlearning. By introducing the label domain mismatch among forgetting data, model output, and target concept, we uncover three additional tasks beyond the conventional all matched forgetting, e.g., target mismatch, model mismatch, and data mismatch forgetting. We identify the insufficient representation and decomposition lacking of restrictively forgetting the target concept, and reveal the crucial forgetting dynamics in the representation level for the feasibility of these unlearning requests. Based on that, we propose the TARF that assigns an annealed gradient ascent on the identified forgetting data and the normal gradient descent on the selected retaining data. By collaboratively considering the forgetting/retaining target, TARF is more accurate in unlearning while maintaining the rest. We hope our work can provide new insights and draw more attention toward the practical scenarios of machine unlearning.

**Open challenge and future work discussion.** Representation gravity, that relies on the forgetting dynamics, is central to the ability of TARF to identify latent target concepts. In challenging regimes where concepts are inherently ambiguous, weakly clustered, or attribute-entangled (e.g., certain long-tailed or multi-attribute scenarios), the underlying representation structure itself becomes less separable. It affects all existing unlearning methods as the ambiguity originates from the nature of the data. In our exploration, we also observe a few preliminary cases where the gravity signal becomes weaker and the ranking slightly noisier. We therefore view these situations as inherent difficulties when the target concept is not well-defined in the representation space. At the same time, these cases also suggest promising avenues for future research, such as incorporating external knowledge (e.g., text embeddings, semantic priors, or multi-modal cues) to assist more challenging unlearning.

## ACKNOWLEDGMENT

JNZ and BH were supported by RGC General Research Fund No. 12200725, RIKEN Collaborative Research Fund, and HKBU CSD Departmental Incentive Scheme. JCY is supported by National Natural Science Foundation of China (No. 62306178) and STCSM (No. 22DZ2229005). JLX was supported by RGC Grant C1043-24GF. MS was supported by JST ASPIRE Grant Number JPMJAP2405.

## ETHICS STATEMENT

This work adheres fully to the Code of Ethics. Our study does not involve human or animal subjects, and all datasets and models used are publicly available as detailed in experimental section and appendix. We have taken care to ensure that our methodology does not propagate sensitive, private, or personally identifiable information. The research is intended purely for advancing scientific understanding on machine unlearning and poses no foreseeable risks of misuse or harm. We confirm compliance with legal, fairness, transparency, and research integrity standards.

## REPRODUCIBILITY STATEMENT

We have made extensive efforts to ensure the reproducibility of our results. A detailed version of reproducibility statement can be found in Appendix, where we summarize critical aspects to facilitate verification. In addition, we also provide an anonymous repository containing code, training scripts, and instructions for reproducible results. Detailed descriptions of models, datasets, and experimental setups are provided in the Section 4.1 and Appendix F.1. And we also introduce the construction of unlearning task instantiation in detail in Appendix D.4 for a specific reference.

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

APPENDIX

The whole Appendix is structured in the following manner. In Appendix A, we provide a detailed discussion with related works of machine unlearning and other aspects. In Appendix B, we review the representative baseline methods in machine unlearning, which are considered in our experimental comparisons. And we also detailed the evaluation for different unlearning scenarios with full results on challenge and representation analysis. In Appendix D, we introduce the complete scenarios considering the mismatch issues in machine unlearning, going beyond the four basic scenarios presented in the main text. In Appendix E, we formally present the algorithm implementation of our proposed TARF with its variant, and further explanation of the rationality of TARF in unlearning. In Appendix F, we provide additional experimental results to characterize forgetting dynamics and the properties of TARF. In Appendix G, we discuss the potential broader impact and limitations of our work.

## LLM Usage Statement

In this paper, no large language models (LLMs) were used in the conception, execution, or analysis of this work. All research ideas, experiments, and text were developed and written solely by the authors.

## Reproducibility Statement

Below we summarize some critical aspects to facilitate reproducible results:

- **Datasets.** The datasets we used are all publicly accessible, which is introduced in Section 4.1. For our newly introduced unlearning scenarios, we provide the specific dataset construction in our code, implemented as described in Section 4.1 and Appendix D.4.

- **Assumption.** Following the previous work (Warnecke et al., 2023; Jia et al., 2023; Fan et al., 2023), We set our experiments to a tuning scenario where a well-trained model is available, and all the training samples are available but limited samples are labeled as "to be unlearned".

- **Open source.** The code repository will be available in an anonymous repository for the reviewing purposes. We provide a series of unlearning methods considered in our work and also the pre-trained model for unlearning.

- **Environment.** All experiments are conducted with multiple runs on NVIDIA Tesla V100-SXM2-32GB GPUs with Python 3.8 and PyTorch 1.8. More detailed requirements can also refer to the environment descriptions in our aforementioned source codes.

## A  Discussion about Related Work

In this section, we discuss the related literature on machine unlearning, and provide more detailed comparisons of some work with their approaches and motivations.

### A.1  Machine Unlearning

Machine unlearning targets to adjust a trained model to scrub the data influence (Koh & Liang, 2017; Shaik et al., 2023; Xu et al., 2023). It is initially proposed to protect data privacy (Cao & Yang, 2015; Bourtoule et al., 2021; Ginart et al., 2019), and a series of studies explore probabilistic methods through the differential privacy (Ginart et al., 2019; Guo et al., 2018; Neel et al., 2021; Ullah et al., 2021; Sekhari et al., 2021). Although having the provable guarantee on the unlearning errors, the strong algorithmic assumptions hinders the practical effectiveness (Jia et al., 2023). Current research (Golatkar et al., 2020; Thudi et al., 2022b;a; Fan et al., 2023; Chen et al., 2023; Yao et al., 2023; Gandikota et al., 2023; Zhang et al., 2023) focus more on developing more effective and efficient unlearning methods to approximate the Retrained model, with the given trained model. As for the assumption on data generation, prior works (Golatkar et al., 2020; Warnecke et al., 2023; Jia et al., 2023; Chen et al., 2023) mainly consider all matched forgetting targets, with similar features on the original training tasks. As for the assumption on label generation, most prior works (Bourtoule et al., 2021; Graves et al., 2021; Thudi et al., 2022a; Jia et al., 2023; Fan et al., 2023; 2024) assume

the accessibility on the fully identified forgetting dataset, and the complementary is the remaining dataset. One recent related work (Yoon et al., 2022) considers unlearning with only a few forgetting samples but requires another generative model to generate approximated data. Our work considers a more practical scenario in which we can conduct mismatched forgetting and use limited identified forgetting data with the remaining set. Another work (Shen et al., 2024) proposes label-agnostic forgetting to enable supervision-free unlearning in deep models, while effective, but the method shows the incompatibility of the assumption since it assumes access to fully identified forgetting set and retaining set during optimization which is not aligned with our formulation. With the increasing attention to safety and regulation in foundation models, machine unlearning has received growing interest, machine unlearning also draw more research interests and some recent studies (Li et al., 2025; Muhamed et al., 2025) also explore the post-adjustment for foundation model oriented unlearning or concept erasure, and the focuses is structural specified and also follow the conventional assumption having the well-aligned forgetting target. While such advances have pushed the field forward, our work focuses on a fundamentally different challenge: label-domain mismatch, where the target concept to be forgotten does not coincide with the model's original class taxonomy. We aim to provide a rigorous treatment of this overlooked yet practically important setting, including both theoretical insights and a general unlearning framework capable of addressing these mismatches.

## A.2 Positive-unlabeled Learning

Positive-unlabeled learning (du Plessis et al., 2014; Menon et al., 2015) tries to learn a binary classifier from a few labeled positive samples with the rest unlabeled ones. A series of PU algorithms (Liu et al., 2002; Du & Cai, 2015; du Plessis et al., 2015) are developed to train an accurate binary classifier, and can be roughly divided into two categories (Bekker & Davis, 2020). The first branch is cost-sensitive learning, which is related to importance weighting (Liu & Tao, 2015). Given the estimated class prior, these methods (du Plessis et al., 2015; Kiryo et al., 2017; Chen et al., 2020b) can develop an unbiased or consistent risk estimator for PU learning. Another branch of PU learning adopts two heuristic steps to perform binary classification. Such methods (Liu et al., 2002; Yu et al., 2004) first identify reliable negative and positive examples from the unlabeled data, and then conduct semi-supervised learning. The model trained using cost-sensitive learning can also be a recognizer for positive or negative samples (Hsieh et al., 2019). Different from PU learning focusing on binary classification tasks, our work tries to enable more practical scenarios in class-wise unlearning (Shaik et al., 2023) where the class labels and target concepts are decoupled, and we consider the label domain mismatch.

## B  Details about Considered Baselines and Metrics

In this section, we provide details about the considered representative baselines for machine unlearning methods, as well as their general intuitions with specific objectives. For the specific hyperparameters adopted in different methods, we keep the same setting with previous related works (Jia et al., 2023; Fan et al., 2023), and the specific values are listed in detail in our source codes. In addition, we introduce the evaluation metrics in detail, corresponding to the implementations in different unlearning scenarios.

### B.1  Unlearning Methods

**Finetune (FT).**  Utilizing the catastrophic forgetting (Kirkpatrick et al., 2017) in the model (e.g., existed in the continual learning), FT (Warnecke et al., 2023) fine-tunes the given trained model partially on $\mathcal{D}_r$ with few training epochs to obtain the $\theta^*_{un}$ with the following objective function,

$$L_{FT} = \frac{1}{|\mathcal{D}_r|} \sum_{(x,y) \sim \mathcal{D}_r} \ell(f(x), y). \tag{6}$$

**Gradient Ascent (GA).**  Different from the normal gradient descent, GA reverses the gradient signal on $\mathcal{D}_f$ to conduct maximization with ascended gradients, resulting in the increasing loss of the forgetting data to obtain the $\theta^*_{un}$. The objective is given as follows,

$$L_{GA} = -\frac{1}{|\mathcal{D}_f|} \sum_{(x,y) \sim \mathcal{D}_f} \ell(f(x), y). \tag{7}$$

With reverse optimization to maximize the loss on the specific data, the model can approximate $\theta^*$ by directly forgetting the learned knowledge represented by the forgetting data.

**Random Label (RL).**    Similar to GA, RL (Golatkar et al., 2020) assign the random labels $Y^*$ on the forgetting data in $\mathcal{D}_\text{f}$ and fine-tune the given model with it to obtain the unlearned model $\theta_\text{un}^*$,

$$L_\text{RL} = \frac{1}{|\mathcal{D}_\text{f}|} \sum_{(x,y) \sim \mathcal{D}_\text{f}} \ell(f(x), y^*). \tag{8}$$

Instead of using the original training label on the forgetting data in $\mathcal{D}_\text{f}$, RL can destroy the learned feature by using the random label $y^*$ on $\mathcal{D}_\text{f}$, which violate the minimized loss value.

**Influence Unlearning (IU).**    IU adopts the influence function (Koh & Liang, 2017) to estimate the change if the training point is removed from the training loss. It is designed for random data unlearning (Shaik et al., 2023) with the provable guarantee on the unlearning effects. In general, IU estimates the change in model parameters of $\theta_\text{un}^* - \theta$ and adds the weight perturbation to the given model to obtain the unlearned one. However, it usually requires additional model information and training assumptions for the theoretical guarantee and may suffer hyperparameter tuning with inaccurate hessian estimation (Jia et al., 2023; Fan et al., 2023).

**Boundary Shrink (BS).**    BS (Chen et al., 2023) is recently proposed for class-wise unlearning, especially on the all matched forgetting. It focuses on the decision spaces (Goodfellow et al., 2016) of the given trained model. The critical idea is to shift the original decision boundary to imitate the decision behavior of the model retrained from scratch. Motivated by adversarial attacks (Madry et al., 2018), it proposes a neighbor searching method to identify the nearest but incorrect class labels $y_\text{near}$ for $\mathcal{D}_\text{f}$ to guide the model to unlearn the existing class and shift the decision boundary. Using the adversarial attack to find the nearest incorrect label, the objective of BS can be formulated as follows,

$$L_\text{BS} = \frac{1}{|\mathcal{D}_\text{f}|} \sum_{(x,y) \sim \mathcal{D}_\text{f}} \ell(f(x), y_\text{near}), \tag{9}$$

where $y_\text{near}$ is obtained by first perturbing the forgetting data and getting the newly predicted result as,

$$\begin{aligned} x' &= x + \epsilon \cdot \text{sign}(\nabla \ell(f(x), y)) \\ y_\text{near} &\leftarrow \text{softmax}(f(x')) \end{aligned} \tag{10}$$

$L_1$**-sparse.**    Developed based on the conventional FT, $L_1$-sparse (Jia et al., 2023) investigate the model sparsity on machine unlearning. It figures out that model sparsification can benefit the unlearning performance on different perspectives via first pruning and then conducting unlearning. By carrying out pruning and unlearning simultaneously, $L_1$-sparse proposes the sparsity-aware unlearning utilizing the $L_1$ norm-based penalty. The objective is as follows with a hyperparameter $\gamma$,

$$L_{L_1\text{-sparse}} = \frac{1}{|\mathcal{D}_\text{r}|} \sum_{(x,y) \sim \mathcal{D}_\text{r}} \ell(f(x), y) + \gamma ||\theta^*||, \tag{11}$$

and the general sparsity-aware penalty can also be added to different unlearning methods. In this work, we mainly compare the $L_1$-sparse FT as the previous work (Jia et al., 2023; Fan et al., 2023) considered.

**SalUn.**    With the concern on unlearning stability and cross-domain applicability, SalUn (Fan et al., 2023) introduces the concept of weight saliency in machine unlearning. This innovation directs the attention of unlearning into specific model weights for specific data that need to be unlearned. In general, it first generates the gradient-based weight saliency map inspired by model sparsification (Jia et al., 2023) with gradient-value thresholding, where the specific generation method is defined as,

$$m_s = \mathbf{1}(|\nabla_\theta \ell(\theta; \mathcal{D}_f)|_{\theta = \theta_o}| \geq \gamma), \quad \theta_u = m_s \odot (\delta\theta + \theta_o) + (1 - m) \odot \theta_o, \tag{12}$$

in which $\mathbf{1}(g \geq \gamma)$ is an element-wise indicator function that yields a value of $\mathbf{1}$ for the i-th element if and 0 otherwise, $|\cdot|$ is an element-wise absolute value operation, and $\gamma > 0$ is a hard threshold. and then conducts saliency-based unlearning using the generated saliency map. Specifically, SalUn

adopts RL (Golatkar et al., 2020) to fine-tune the forgetting data in $\mathcal{D}_f$ on the salience map, and the extended objective is given as follows,

$$L_{\text{SalUn}} = \frac{1}{|\mathcal{D}_f|} \sum_{(x,y)\sim\mathcal{D}_f} \ell_{\theta_u}(f(x), y^*) + \alpha \frac{1}{|\mathcal{D}_r|} \sum_{(x,y)\sim\mathcal{D}_r} \ell(f(x), y), \tag{13}$$

More detailed operations can refer to (Fan et al., 2023), and we keep the same hyperparameter used in (Fan et al., 2023) to conduct the class-wise unlearning tasks.

**SCRUB.** SCRUB is a newly proposed unlearning algorithm based on a novel casting of the problem into a teacher-student framework (Kurmanji et al., 2023). It is designed to meet the desiderata of unlearning: efficiently forgetting without hurting the model utility. As the general target of SCRUB in forgetting is application-dependent, it is proposed with a recipe that works across applications: SCRUB is first to strive for maximal forget error, which is desirable in some scenarios like removing bias or restricted contents but not in others like user privacy protection. To address the latter case, SCRUB is integrated with a rewinding procedure that can reduce the forget set error appropriately when required.

Given the original model $\theta^o$ as the teacher model, the goal of SCRUB is formatting as training a student model $\theta^u$ that selectively obeys the teacher. The overall objective can be divided into two folds, the first is to remember $\mathcal{D}_r$ under the teacher model's guide while the second is to forget $\mathcal{D}_f$ by disobeying the teacher model's guide. To measure the degree to which the student model obeys the teacher model, SCRUB utilizes the following distance measure,

$$d(x; \theta^u) = D_{\text{KL}}(p(f(x; \theta^o))||p(f(x; \theta^u))), \tag{14}$$

where $D_{\text{KL}}$ is the KL-divergence and the overall measures of the distance between the student model's and teacher model's prediction distribution. With the aforementioned distance, the objective of SCRUB is as follows,

$$L_{\text{SCRUB}} = \min_{\theta^u} \frac{\alpha}{N_r} \sum_{x_r \in \mathcal{D}_r} d(x_r; \theta^u) + \frac{\gamma}{N_r} \sum_{(x_r, y_r)\in\mathcal{D}_r} \ell(f(x_r; \theta^u), y_r) - \frac{1}{N_f} \sum_{x_f \in \mathcal{D}_f} d(x_f; \theta^u), \tag{15}$$

where the first two parts can be regarded as a variant of distillation from a teacher model on $\mathcal{D}_r$ and the third part is encouraging the student model to disobey the teacher model to forget the target data.

Table 6: Comparison with additional recent class-wise unlearning methods on CIFAR-100.

| All Matched | UA | RA | TA | MIA | Gap↓ | Target Mismatch | UA | RA | TA | MIA | Gap↓ |
|---|---|---|---|---|---|---|---|---|---|---|---|
| Retrained (Ref.) | 0.00 | 97.85 | 76.03 | 100.00 | - | Retrained (Ref.) | 0.00 | 97.85 | 73.72 | 100.00 | - |
| FT | 0.67 | 96.32 | 72.34 | 100.00 | 1.47 | FT | 58.18 | 96.32 | 72.53 | 46.76 | 28.54 |
| LAU | 4.11 | 80.44 | 61.64 | 95.78 | 10.03 | LAU | 46.71 | 88.65 | 68.19 | 66.49 | 23.74 |
| SFR-on | 0.00 | 99.21 | 74.26 | 100.00 | 0.78 | SFR-on | 59.21 | 99.13 | 74.28 | 48.32 | 28.18 |
| SG | 0.00 | 95.21 | 71.23 | 100.00 | 1.86 | SG | 58.21 | 96.26 | 72.18 | 46.24 | 28.78 |
| SCRUB | 0.00 | 99.98 | 76.75 | 100.00 | **0.71** | SCRUB | 59.64 | 99.99 | 75.32 | 44.89 | 29.90 |
| TARF | 0.00 | 96.90 | 72.53 | 100.00 | 1.11 | TARF | 0.31 | 97.35 | 73.68 | 100.00 | **0.21** |
| **Model Mismatch** | UA | RA | TA | MIA | Gap↓ | **Data Mismatch** | UA | RA | TA | MIA | Gap↓ |
| Retrained (Ref.) | 88.22 | 98.58 | 78.50 | 25.78 | - | Retrained (Ref.) | 0.00 | 98.50 | 80.15 | 100.00 | - |
| FT | 92.67 | 95.02 | 79.34 | 16.33 | 4.58 | FT | 82.62 | 95.66 | 79.77 | 37.24 | 37.15 |
| LAU | 80.00 | 96.74 | 79.86 | 45.78 | 7.86 | LAU | 85.73 | 96.96 | 80.00 | 40.40 | 36.76 |
| SFR-on | 92.12 | 99.21 | 79.21 | 20.65 | 2.59 | SFR-on | 92.68 | 99.21 | 79.23 | 18.21 | 44.03 |
| SG | 89.27 | 93.52 | 73.45 | 19.31 | 4.41 | SG | 87.52 | 93.25 | 73.21 | 23.08 | 44.16 |
| SCRUB | 91.44 | 99.74 | 79.23 | 21.11 | 2.45 | SCRUB | 95.50 | 99.79 | 79.68 | 15.11 | 45.54 |
| TARF | 86.67 | 97.05 | 80.07 | 26.00 | **1.21** | TARF | 0.00 | 95.01 | 78.98 | 100.00 | **1.17** |

In addition, we also incorporate three more recent unlearning methods into our comparison, e.g., LAU (Shen et al., 2024), SFR-on (Huang et al., 2024), and SG (Di et al., 2025), in Table 6. These methods propose some advancements for class-wise unlearning in different aspects while not considering the mismatched challenges. Both have been evaluated under the same mismatched unlearning scenarios introduced in our paper, using identical training budgets and evaluation protocols to ensure a fair comparison. The additional results also validate the effectiveness and generality of our TARF framework across all tasks, as the unified framework design enables target identification and separation. TARF maintains robust unlearning performance, due to its flexible capabilities of handling various mismatched unlearning scenarios.

## B.2 EVALUATION METRICS REGARDING DIFFERENT SCENARIOS

In this part, we summarize the following list and tables of the evaluation metrics (adopted from the previous work (Jia et al., 2023; Fan et al., 2023)) and the used labels in different unlearning scenarios,

- Unlearning Accuracy (**UA**): the accuracy of the unlearned model $\theta_u$ on the dataset of target concept $D_t$. $UA = \frac{1}{|D_t|} \sum_{(x,y) \in D_t} \mathbf{1}[\hat{y}_\theta(x) = y]$

- Retaining Accuracy (**RA**): the accuracy of the unlearned model $\theta_u$ on retaining dataset $D_r$. $RA = \frac{1}{|D_r|} \sum_{(x,y) \in D_r} \mathbf{1}[\hat{y}_\theta(x) = y]$

- Testing Accuracy (**TA**): the accuracy of the unlearned model $\theta_u$ on test dataset $D_{test}$ excluding the data belonging to the target concept. $TA = \frac{1}{|D_{test} \setminus D_t|} \sum_{(x,y) \in D_{test} \setminus D_t} \mathbf{1}[\hat{y}_\theta(x) = y]$

- Model Inversion Attack (**MIA**): the MIA success rate by a confidence-based MIA predictor of the model $\theta_u$ on the dataset of target concept $D_t$. We follow (Jia et al., 2023) to implement it to find how many samples in $D_t$ can be correctly predicted as a non-training sample by the MIA predictor against $\theta_u$. First, we sample a balanced dataset from the retaining dataset $D_r$ and the test dataset excluding the forgetting data to train the MIA predictor, then it is used to count the rate of true negative predictions for forgetting data of the target concept.

- Gap: $\frac{1}{4} \cdot \left( |UA_{\theta^r} - UA_{\theta^{un}}| + |RA_{\theta^r} - RA_{\theta^{un}}| + |TA_{\theta^r} - TA_{\theta^{un}}| + |MIA_{\theta^r} - MIA_{\theta^{un}}| \right)$

**Remark.** We follow previous work (Jia et al., 2023) to adopt the **TIME** which report the wall-clock training time required to perform unlearning from the original model initialization. The **termination** of each unlearning methods is defined by the recommended hyperparameter for training epochs (e.g., early stopping 10 epochs for FT-style method as performance plateaus or a fixed maximum epoch cap like 5 for GA-style method) on specific unlearning tasks.

Generally, in the evaluation phase, we adopt the same labels used in pre-training to measure the unlearned model. Note that in the model mismatch forgetting, as the model is trained with superclass labels, the UA is also calculated using the superclass label. Hence, the UA of the Retrained reference is not equal to 0 as indicated in Table 3, and we compare the methods mainly on the averaged performance "Gap" (calculated based on the previous four metrics) to the Retrained reference.

Table 7: The label used in evaluation metrics on different forgetting scenarios.

| Used Label | All matched | Target mismatch | Model mismatch | Data mismatch |
|---|---|---|---|---|
| UA | Class Label | Class Label | Superclass Label | Superclass Label |
| RA | Class Label | Class Label | Superclass Label | Superclass Label |
| TA | Class Label | Class Label | Superclass Label | Superclass Label |
| MIA | Class Label | Class Label | Superclass Label | Superclass Label |

In Table 7, we summarize the specific label used in different unlearning scenarios. To provide an intuitive example that corresponds to the instantiated unlearning tasks like Figure 1, we present Table 8 to give overall information about the data and labels considered in each metric.

Table 8: The evaluation data (label number) of different forgetting scenarios with CIFAR-100.

| Data (classes number) | All matched | Target mismatch | Model mismatch | Data mismatch |
|---|---|---|---|---|
| UA ($D_t$) | "boy", "girl" (2) | "boy", "girl", "man", "woman", "baby" (5) | part of "people" (1), which is data of "boy" and "girl" but with superclass label | "people" (1) |
| RA ($D_r$) | Other classes (98) | Other classes (95) | other part of "people" (1) with the rest superclasses (19) | Other superclasses (19) |
| TA ($D_{test}$) | Other classes (98) | Other classes (95) | other part of "people" (1) with the rest superclasses (19) | Other superclasses (19) |
| MIA ($D_t$) | "boy", "girl" (2) | "boy", "girl", "man", "woman", "baby" (5) | part of "people" (1), which is data of "boy" and "girl" but with superclass label | "people" (1) |

## B.3 FULL RESULTS OF FIGURE 2 AND FIGURE 3

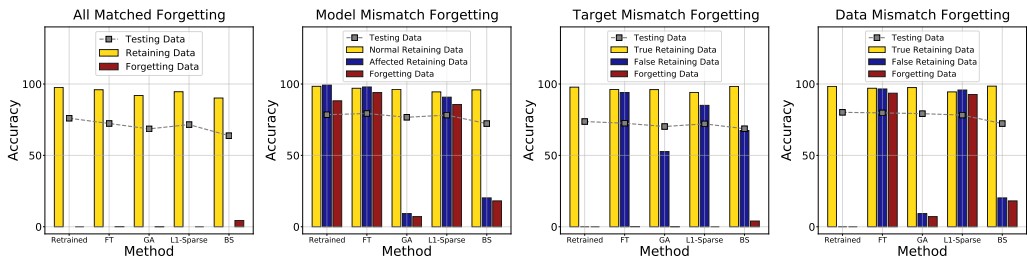

Figure 8: Unlearning results across four tasks using different representative methods.

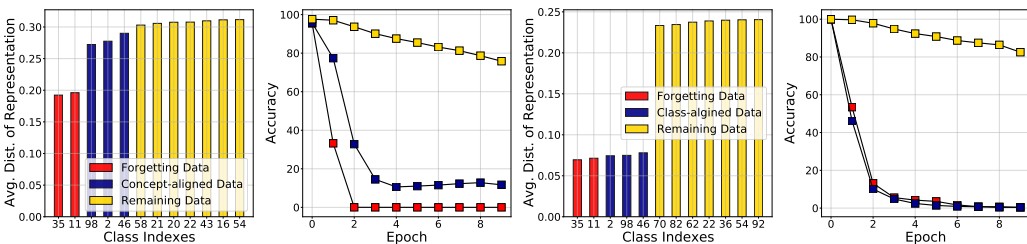

Figure 9: Forgetting dynamics on entangled (left) and under-entangled (right) feature representations.

## C  PROOF OF THEOREM 3.2

Here we provide the complete proof of Theorem 3.2. The proof sketch is from Taylor expansion (Bottou et al., 2018) on $L_{s_1}(\theta^{t+1})/L_{s_2}(\theta^{t+1})$, subtracting two formulas, and bounding the differences term. It intuitively reveals the data representation can affect the forgetting dynamic on the loss differences. As our primary goal is to explore the new settings, we believe it worth future work to establish more systematical analysis beyond the current gravity effects on the forgetting dynamics.

First, we can have the following assumption about the representation similarity of $d_h(\cdot, \cdot)$, which is also adopted and assumed in the various previous works (Belkin & Niyogi, 2003; Chen et al., 2020a).

**Assumption C.1** (Representation Similarity). Let $s_1$ and $s_2$ be two disjoint subsets of a dataset $D$ trained on a model $f_\theta$. Given the representation of an input x at an intermediate layer be $h(x)$, the gradient differences at representation level can be controlled by assuming $\ell_h(\cdot)$ is Lipschitz smooth with constant $C_\ell$, then we have $||\nabla \ell_h(x_1) - \nabla \ell_h(x_2)|| \le C_\ell ||h(x_1) - h(x_2)|| = C_\ell d_h(x_1, x_2)$.

**Theorem C.2** (gravity effects on unlearning update). *Let $\theta^0$ be the well-trained model parameters for unlearning, and we perform unlearning on $s_1$ via a gradient ascent update, i.e., $\theta^{t+1} = \theta^t + \nabla L_{s_1}(\theta^t)$ for epoch t, then we can the following dynamics given $\Delta L_{s_1, s_2}(\theta^{t+1}) = (L_{s_1}(\theta^{t+1}) - L_{s_2}(\theta^{t+1}))$,*

$$\Delta L_{s_1, s_2}(\theta^{t+1}) \le (L_{s_1}(\theta^t) - L_{s_2}(\theta^t)) + \eta \lambda_{max}(J_{\theta^t}(x_1)) C_\ell \mathbb{E} d_h(x_1, x_2) \cdot ||\nabla L_{s_1}(\theta^t)|| + \mathcal{O}(\eta^2),$$
(16)

*where $\lambda_{max}(J_{\theta^t})$ is the largest eigenvalue of the Jacobin matrix $J_\theta = \frac{\partial h(x)}{\partial \theta}$. Note that when $t \to 0$, the RHS mainly relies on the term measuring representation similarity as $(L_{s_1}(\theta^t) - L_{s_2}(\theta^t)) \to 0$.*

*Proof.* Let $s_1$ and $s_2$ be two disjoint subsets with associated empirical losses $L_{s_1}(\theta)$ and $L_{s_2}(\theta)$ for model parameters $\theta$. Suppose the representations $h_\theta(x)$ of inputs $x_1 \sim s_1$ and $x_2 \sim s_2$ are similar:

$$\mathbb{E}_{x_1 \sim S_1, x_2 \sim S_2} ||h_\theta(x_1) - h_\theta(x_2)|| \le \epsilon.$$

Assume the loss $\ell_h(\cdot)$ is Lipschitz smooth with constant $C_\ell$, and $J_{\theta^t}(x) = \frac{\partial h_\theta(x)}{\partial \theta}$ is the Jacobian of the representation. Suppose an update $\theta^{t+1} = \theta^t + \Delta\theta^t$ is applied (e.g., for unlearning). Then by Taylor expansion we have,

$$L_{s_i}(\theta^{t+1}) = L_{s_i}(\theta^t) + \nabla L_{s_i}(\theta)^t(\theta^{t+1} - \theta^t) + \frac{1}{2}(\theta^{t+1} - \theta^t)^T H_{S_i}(\theta^{t+1} - \theta^t)$$

then subtracting expansions for $s_1$ and $s_2$,

$$L_{s_1}(\theta^{t+1}) - L_{s_2}(\theta^{t+1}) = (L_{S_1}(\theta^t) - L_{S_2}(\theta^t)) + (\nabla L_{s_1}(\theta^t) - \nabla L_{s_2}(\theta^t))^T \Delta\theta^t$$
$$+ \frac{1}{2}\Delta\theta^{tT}(H_{s_1} - H_{s_2})\Delta\theta^t$$

using the chain rule,

$$\nabla L_{s_i}(\theta) = \mathbb{E}_{x \sim s_i}[J_{\theta^t}(x)^T \nabla_h \ell(h(x))],$$

then,

$$\nabla L_{s_1}(\theta^t) - \nabla L_{s_2}(\theta^t) = \mathbb{E}_{x_1, x_2}[J_{\theta^t}(x_1)^T \nabla_h \ell(x_1) - J_{\theta^t}(x_2)^T \nabla_h \ell(x_2)]$$

we can split this as,

$$\mathbb{E}_{x_1, x_2}[J_{\theta^t}(x_1)^T(\nabla_h \ell(x_1) - \nabla_h \ell(x_2)) + (J_{\theta^t}(x_1)^T - J_{\theta^t}(x_2)^T)\nabla_h \ell(x_2)]$$

then, by triangle inequality and operator norms,

$$\nabla L_{s_1} - \nabla L_{s_2} \le \lambda_{\max}(J_{\theta^t}(x_1)) \cdot \mathbb{E}_{x_1, x_2} ||\nabla_h \ell(x_1) - \nabla_h \ell(x_2)|| + O(\epsilon),$$

assuming $\ell_h(\cdot)$ is Lipschitz smooth with constant $C_\ell$,

$$||\nabla \ell_h(x_1) - \nabla \ell_h(x_2)|| \leq C_\ell ||h(x_1) - h(x_2)|| = C_\ell d_h(x_1, x_2),$$

so

$$\nabla L_{s_1} - \nabla L_{s_2} \leq \lambda_{\max}(J_{\theta^t}(x_1)) C_\ell \mathbb{E} d_h(x_1, x_2),$$

and,

$$(\nabla L_{s_1}(\theta^t) - \nabla L_{s_2}(\theta^t))^T \Delta \theta^t \leq \lambda_{\max}(J_{\theta^t}(x_1)) C_\ell \mathbb{E} d_h(x_1, x_2) \cdot ||\nabla L_{s_1}(\theta^t)||$$

combining everything and given $\Delta L_{s_1, s_2}(\theta^{t+1}) = (L_{s_1}(\theta^{t+1}) - L_{s_2}(\theta^{t+1}))$,

$$\Delta L_{s_1, s_2}(\theta^{t+1}) \leq (L_{s_1}(\theta^t) - L_{s_2}(\theta^t)) + \eta \lambda_{\max}(J_{\theta^t}(x_1)) C_\ell \mathbb{E} d_h(x_1, x_2) \cdot ||\nabla L_{s_1}(\theta^t)|| + \mathcal{O}(\eta^2).$$

$\square$

The loss difference between $s_1$ and $s_2$ after an unlearning update on $s_1$ is bounded above by a term proportional to their representation similarity. This shows that representation similarity implies loss entanglement, where unlearning one set will influence the loss on another if they share similar representations.

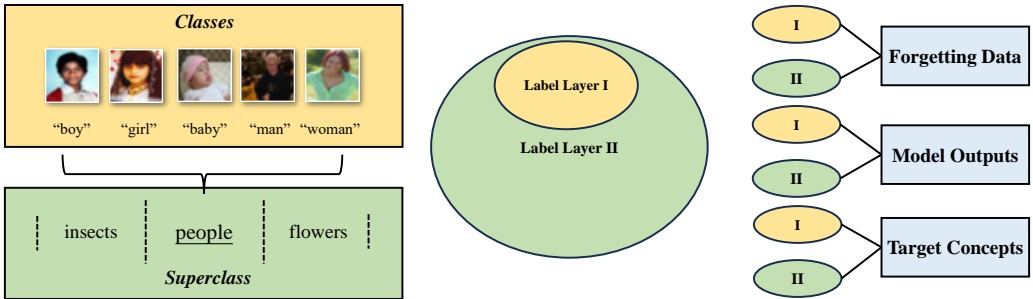

The left panel shows an example of two-layer label domains; The middle panel is the Venn diagram to show the hierarchical relation; The right panel illustrates the potentials of three critical class-wise unlearning aspects.

Figure 10: **Label domain mismatch with the two-layer illustration.**

Table 9: Mismatching in the label domain of three critical aspects with a two-layer label structure.

| No. | Forgetting data | Model output | Target concept | Comment |
|---|---|---|---|---|
| 1 | Class label | Class label | Class label | **All matched** |
| 2 | Class label | Class label | Superclass | **Target mismatch** |
| 3 | Class label | Superclass | Class label | **Model mismatch** |
| 4 | Class label | Superclass | Superclass | **Data mismatch** |
| 5 | Superclass | Class label | Class label | Impractical since $\mathcal{L}_D \succ \mathcal{L}_T$ |
| 6 | Superclass | Class label | Superclass | Similar to all matched |
| 7 | Superclass | Superclass | Class label | Impractical since $\mathcal{L}_D \succ \mathcal{L}_T$ |
| 8 | Superclass | Superclass | Superclass | **All matched** |

## D  FULL DISCUSSION ABOUT LABEL DOMAIN MISMATCH

In this section, we discuss the full scenarios of label domain mismatch in class-wise unlearning (Warnecke et al., 2023; Golatkar et al., 2020; Chen et al., 2023; Jia et al., 2023; Fan et al., 2023). Specifically, we will start by why focusing on class-wise unlearning, and then discuss the motivation for investigating its label domain mismatch, with the newly introduced setting being friendly for empirical analysis and further research. In addition, we provide detailed information on our instantiated four tasks using the benchmarked datasets (Krizhevsky, 2009). Finally, we discuss the commonalities of mismatch forgetting scenarios and the general principle of unified framework design.

To begin with, machine unlearning (Cao & Yang, 2015; Thudi et al., 2022b; Xu et al., 2023; Shaik et al., 2023) is originally proposed in response to "the right to be forgotten" to protect the data privacy, and recently deep machine unlearning is a timely research topic associated with foundation models which use massive of data to train (Kurmanji et al., 2023; Bommasani et al., 2021). The ensuing data regulation concerns have also expanded the original privacy-protecting goal to more general needs and scenarios (Yao et al., 2023; Maini et al., 2024; Gandikota et al., 2023). As stated in (Shah et al., 2023; Kurmanji et al., 2023; Jia et al., 2023), unlearning a subset of the training set has received increasing attention (like removing sensitive information, and inappropriate content). However, the previous scenarios mainly consider the coinciding class labels with the target concept to be unlearned. Although achieving promising results in forgetting, it is still not enough in practice.

Considering the problem setups of unlearning, we have three critical aspects, e.g., the well-trained machine learning model $\theta$, and the reported data $\mathcal{D}_f$ to be unlearned, as well as the target concept. In previous studies, the three aspects are mainly considered to be under the same label taxonomy. In other words, the unlearning tasks are aligned with the pre-training task, where the latter trains a multi-class classification model, and the former aims to unlearn a training class. However, in practice, the unlearning request may violate the taxonomy of the pre-training tasks, while the

specific target concepts always exhibit a unified property for specific forgetting data. It naturally motivates us to consider different label domains of the three aspects of unlearning. As listed in Table 10, the label domain of data $\mathcal{L}_D$, the label domain of model output $\mathcal{L}_M$, and the label domain of target concept $\mathcal{L}_T$. To begin with, we introduce the relations between two label domains, i.e., $\mathcal{L}_1$ matches $\mathcal{L}_2$ ($\mathcal{L}_1 = \mathcal{L}_2$), $\mathcal{L}_1$ is the subclass domain of $\mathcal{L}_2$ ($\mathcal{L}_1 \prec \mathcal{L}_2$)[2] and $\mathcal{L}_1$ is the superclass domain of $\mathcal{L}_2$ ($\mathcal{L}_1 \succ \mathcal{L}_2$)[3], and we have a practical assumption on the relation between label domains of forgetting data and target concept, i.e., $\mathcal{L}_D \preceq \mathcal{L}_T$, indicating that the reported forgetting data should be included in the target concept (as intuitively illustrated in the middle panel of Figure 10). Considering $\mathcal{L}_D = \mathcal{L}_T$, we can have two possibilities on $\mathcal{L}_M$, e.g., $\mathcal{L}_M = \mathcal{L}_T$ and $\mathcal{L}_M \neq \mathcal{L}_T$, where the former is regarded as all matched when $\mathcal{L}_D = \mathcal{L}_M = \mathcal{L}_T$ and the latter is the model mismatch. To be more specific, we consider model mismatch forgetting as $\mathcal{L}_D = \mathcal{L}_T \prec \mathcal{L}_M$, since $\mathcal{L}_M \prec \mathcal{L}_T$ will have no additional effects on the unlearning when $\mathcal{L}_D = \mathcal{L}_T$ and we can regard it as similar to the all matched case. Considering $\mathcal{L}_D \prec \mathcal{L}_T$, we can have the target mismatch forgetting when $\mathcal{L}_D = \mathcal{L}_M$ and data mismatch forgetting when $\mathcal{L}_M = \mathcal{L}_T$.

We summarize the mainly considered mismatch cases in Table 10, which can serve as a general reference for further research on constructing the unlearning tasks. In the following, we further explain the procedure of task instantiating and discuss the other potential scenarios with the typical two-layer label structure considered in the main text and an additional three-layer label structure.

| Label Domain $\mathcal{L}$ | Relation of Data $\mathcal{L}_D$, Model $\mathcal{L}_M$, and Target $\mathcal{L}_T$ |
|---|---|
| All matched | $\mathcal{L}_D \quad = \quad \mathcal{L}_T \quad = \quad \mathcal{L}_M$ |
| Target mismatch | $\mathcal{L}_M \quad = \quad \mathcal{L}_D \quad \prec \quad \mathcal{L}_T$ |
| Model mismatch | $\mathcal{L}_D \quad = \quad \mathcal{L}_T \quad \prec \quad \mathcal{L}_M$ |
| Data mismatch | $\mathcal{L}_D \quad \prec \quad \mathcal{L}_T \quad = \quad \mathcal{L}_M$ |

Table 10: considering **label domain** relations of three critical aspects in class-wise unlearning.

## D.1  A TWO-LAYER LABEL STRUCTURE OF MISMATCH

In Figure 10, we first show the illustration of a two-layer label structure and the three aspects of unlearning, i.e., forgetting data, model outputs, and target concept. Without losing generality, we utilize the class labels and superclass information (refer to the official information in CIFAR-100 (Krizhevsky, 2009)) for consideration. Then we have a two-layer label structure representing different knowledge regions.

Given two potential label domains in each aspect, we can totally get the 8 scenarios list in Table 9. The first 4 scenarios are mainly considered and detailedly introduced in the main text. For the rest 4 scenarios (i.e., No. 5-8), we consider some (i.e., No. 5 and No. 7) to be impractical as the label domain of forgetting data is larger than the target concept, which means that the unlearning requests identify more forgetting data than the true target concept. It should be more reasonable that only limited forgetting data are identified by server users or internal examiner (Kovashka et al., 2016) in real-world applications. Therefore, we mainly consider the forgetting data $\mathcal{D}_f$ belongs a part of or equals to the overall data $\mathcal{D}_t$ of the target concept. As for No. 6 and No. 8 cases, the former is similar to the conventional all matched forgetting since the forgetting data has the same label domains with the target concept while the model output has a fine-grained label domain (e.g., class label) that will not affect the unlearning, and the latter is exactly same as the all matched forgetting.

## D.2  A THREE-LAYER LABEL STRUCTURE OF MISMATCH

Since in more extreme cases, some unlearning requests would exhibit only several instances of forgetting an abstract concept not aligned with the pre-training tasks. We then consider an extra label layer (e.g., the sub-set level inside a class) to construct a three-layer structure beyond the previous one. In Figure 11, we illustrate it with some samples and a Venn diagram.

Considering each aspect can have three potential label domains, we can totally get 27 scenarios in Table 11. In general, we have three rough categories for analysis. First, due to the aforementioned constraint that the target concept should include the forgetting data, we consider several cases (e.g.,

---

[2]$\mathcal{L}_1 \prec \mathcal{L}_2$: For any label $y \in \mathcal{L}_1$, there exist label $y' \in \mathcal{L}_2$ that instance being labeled with $y$ can also being labeled with $y'$, but not all instance being labeled with $y'$ can be labeled with $y$.

[3]$\mathcal{L}_1 \succ \mathcal{L}_2$: For any label $y \in \mathcal{L}_2$, there exist label $y' \in \mathcal{L}_1$ that instance being labeled with $y$ can also being labeled with $y'$, but not all instance being labeled with $y'$ can be labeled with $y$.

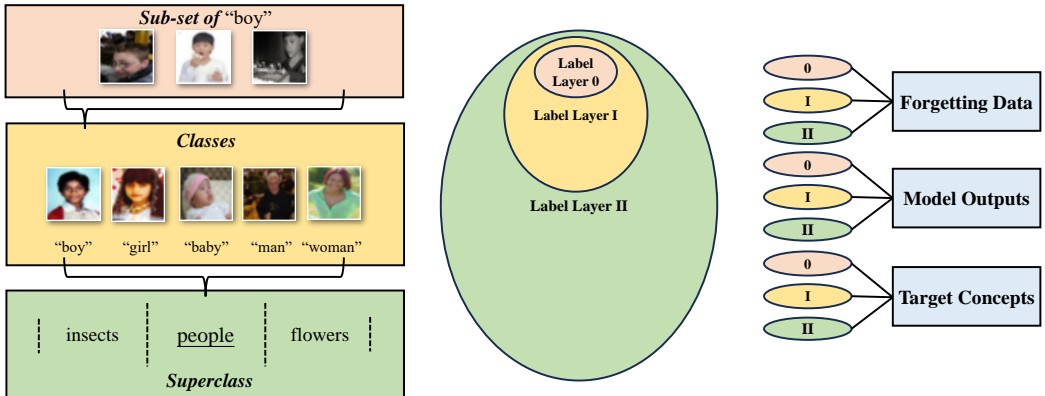

The left panel shows an example of three-layer label domains extended from the ordinary setting considered in our main text, where the sub-set is sampled from the "boy" class; The middle panel is the Venn diagram to show the hierarchical relation; The right panel illustrates the potentials of three critical class-wise unlearning aspects.

Figure 11: **Label domain mismatch with the three-layer illustration.**

Table 11: Mismatching in the label domain of three critical aspects with a three-layer label structure.

| No. | Forgetting data | Model output | Target concept | Comment |
|-----|-----------------|--------------|----------------|---------|
| 1 | Sub-set | Sub-set | Sub-set | **All matched** |
| 2 | Sub-set | Sub-set | Class label | **Target mismatch** |
| 3 | Sub-set | Class label | Sub-set | **Model mismatch** |
| 4 | Sub-set | Class label | Class label | **Data mismatch** |
| 5 | Sub-set | Sub-set | Superclass | Different |
| 6 | Sub-set | Superclass | Sub-set | Different |
| 7 | Sub-set | Superclass | Superclass | Different |
| 8 | Sub-set | Class label | Superclass | Different |
| 9 | Sub-set | Superclass | Class label | Different |
| 10 | Class label | Class label | Class label | **All matched** |
| 11 | Class label | Class label | Superclass | **Target mismatch** |
| 12 | Class label | Superclass | Class label | **Model mismatch** |
| 13 | Class label | Superclass | Superclass | **Data mismatch** |
| 14 | Class label | Sub-set | Sub-set | Impractical since $\mathcal{L}_D \succ \mathcal{L}_T$ |
| 15 | Class label | Sub-set | Class label | Similar to all matched |
| 16 | Class label | Sub-set | Superclass | Different |
| 17 | Class label | Class label | Sub-set | Impractical since $\mathcal{L}_D \succ \mathcal{L}_T$ |
| 18 | Class label | Superclass | Sub-set | Impractical since $\mathcal{L}_D \succ \mathcal{L}_T$ |
| 19 | Superclass | Superclass | Superclass | **All matched** |
| 20 | Superclass | Class label | Class label | Impractical since $\mathcal{L}_D \succ \mathcal{L}_T$ |
| 21 | Superclass | Class label | Superclass | Similar to all matched |
| 22 | Superclass | Superclass | Class label | Impractical since $\mathcal{L}_D \succ \mathcal{L}_T$ |
| 23 | Superclass | Sub-set | Sub-set | Impractical since $\mathcal{L}_D \succ \mathcal{L}_T$ |
| 24 | Superclass | Sub-set | Class label | Impractical since $\mathcal{L}_D \succ \mathcal{L}_T$ |
| 25 | Superclass | Sub-set | Superclass | Similar to all matched |
| 26 | Superclass | Class label | Sub-set | Impractical since $\mathcal{L}_D \succ \mathcal{L}_T$ |
| 27 | Superclass | Superclass | Sub-set | Impractical since $\mathcal{L}_D \succ \mathcal{L}_T$ |

No. 14, 17, 18, 20, 22-24, and 26-27) to be impractical. Second, the three-layer structure also includes a group of scenarios that also existed in the two-layer structure, so No. 1-4 and 19 are the same as the four scenarios (i.e., all matched, target mismatch, model mismatch, and data mismatch). Third, for the rest scenarios, we regarded them to be novel cases than those considered in the main text.

To be more specific, there are two groups of cases in the third part. For No. 5, 6, and 7, since they also can be represented using a two-layer structure, the forgetting dynamics are similar to that in target, model, and data mismatch forgetting. By contrast, in No. 8, 9, and 16, all three label domains exist in the three aspects of class-wise unlearning, which is worthy of further discussion.

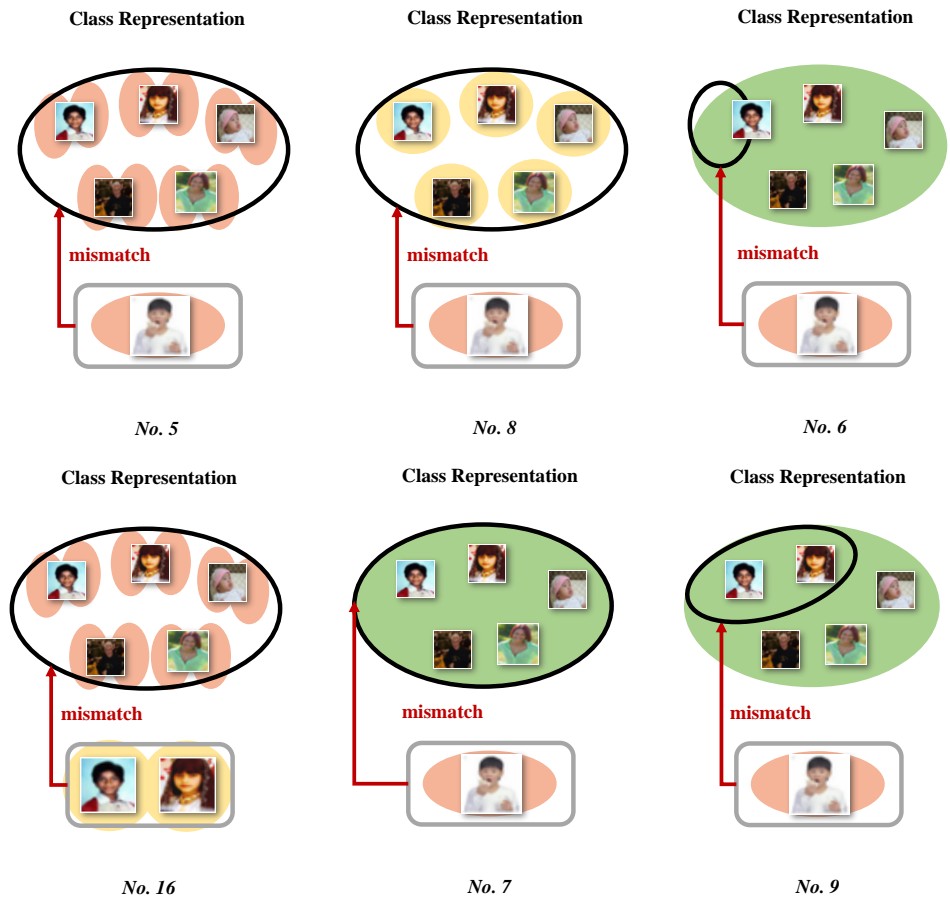

Figure 12: Illustration of 6 scenarios different from the four major tasks according to Table 11.

## D.3 FURTHER EXPLORATION ON THE OTHER 6 DIFFERENT SCENARIOS

In this part, we further discuss the 6 different scenarios discovered by constructing the three-layer label structure. We illustrated these forgetting tasks in Figure 12 and discuss them as follows,

- **No. 5&16** In the two scenarios, the model output has the most fine-grained label domain (e.g., sub-set as illustrated in Figure 11) for representation. At the same time, the target concept is broader than both model output and identified forgetting data. Different from the aforementioned target mismatch, the mismatch degree of this task is larger (e.g., superclass level) than the previous one (e.g., class level). In other words, the model output further loses the entanglement of feature representation of the samples belonging to target concept (compared with the original setups of target mismatch). To simulate the case, we employ the same model pre-trained by class in target mismatch, but enlarge the target concept (consists of 7 classes with similar semantic features, instead of the original 5) and change the forgetting data (2 class as the given forgetting data in No.5 and 3 classes in No.16).

- **No. 8&7.** Similar to the previous No. 5, the target concept in these tasks is also broader than the label domains of the identified forgetting data. However, in these two scenarios, the model output is varied which controls the entanglement of target samples. To construct these two forgetting tasks, we respectively adopt the models pre-trained by class labels and superclass, and use the same forgetting data (with 1 class) to investigate the performance change using our TARF and other baselines.

- **No. 6&9.** In the last two scenarios, the forgetting tasks are more similar to the previous model mismatch forgetting. However, the distinguishable difference is that the label domain of model outputs can be much different from the identified forgetting data. In the No. 6 task, the target concept is aligned with the identified forgetting data, while since the remaining data is more than the original

Table 12: Results (%) of unlearning with different model structures. All methods are trained on the same backbone, i.e., the basis of unlearning initialization is the same (except for retraining from scratch). Values are percentages. Bold numbers are superior results. ↓ indicates smaller are better.

| CIFAR-100 | Metric | UA | RA | TA | MIA | Gap↓ | Metric | UA | RA | TA | MIA | Gap↓ |
|---|---|---|---|---|---|---|---|---|---|---|---|---|
| Retrained | | 0.00 | 97.85 | 73.72 | 100.00 | - | | 0.00 | 97.85 | 73.72 | 100.00 | - |
| FT (Warnecke et al., 2023) | | 67.52 | 96.43 | 72.96 | 41.14 | 32.72 | | 53.11 | 94.64 | 71.23 | 52.70 | 26.53 |
| RL (Toneva et al., 2018) | No. 5 | 68.57 | 96.12 | 72.58 | 41.17 | 33.15 | No. 16 | 53.90 | 96.94 | 73.07 | 53.56 | 25.48 |
| GA (Ishida et al., 2020) | | 38.03 | 97.00 | 70.98 | 76.92 | 16.75 | | 32.24 | 95.73 | 69.99 | 77.62 | 15.12 |
| **TARF** (ours) | | 0.00 | 96.58 | 72.03 | 100.00 | **0.74** | | 0.00 | 96.98 | 72.87 | 100.00 | **0.43** |
| Retrained | | 0.00 | 97.85 | 73.72 | 100.00 | - | | 0.00 | 98.50 | 80.15 | 100.00 | - |
| FT (Warnecke et al., 2023) | | 74.09 | 97.19 | 74.01 | 36.71 | 34.58 | | 95.16 | 94.98 | 78.68 | 13.06 | 46.77 |
| RL (Toneva et al., 2018) | No. 8 | 76.04 | 96.76 | 72.88 | 36.00 | 35.49 | No. 7 | 91.51 | 96.98 | 80.11 | 47.24 | 36.46 |
| GA (Ishida et al., 2020) | | 49.47 | 98.92 | 72.94 | 77.96 | 18.34 | | 15.91 | 98.64 | 80.27 | 93.82 | 5.59 |
| **TARF** (ours) | | 0.00 | 96.22 | 72.43 | 100.00 | **0.73** | | 0.00 | 96.54 | 79.23 | 100.00 | **0.65** |
| Retrained | | 88.22 | 98.52 | 84.42 | 22.22 | - | | 88.22 | 98.58 | 78.50 | 25.78 | - |
| FT (Warnecke et al., 2023) | | 94.33 | 95.00 | 78.77 | 13.67 | 5.96 | | 91.78 | 95.02 | 78.90 | 18.44 | 3.72 |
| RL (Toneva et al., 2018) | No. 6 | 84.22 | 96.96 | 80.18 | 65.77 | 13.34 | No. 9 | 96.97 | 70.22 | 80.24 | 94.67 | 26.94 |
| GA (Ishida et al., 2020) | | 18.44 | 96.06 | 78.20 | 92.67 | 37.23 | | 19.11 | 95.27 | 77.56 | 91.56 | 34.79 |
| **TARF** (ours) | | 92.21 | 98.43 | 82.32 | 19.17 | **2.31** | | 89.12 | 97.23 | 79.21 | 24.32 | **1.11** |

model mismatch forgetting, the task separation could be harder than the previous. In the No. 9 task, we can find that it is a complex scenario where the target concept is broader than the forgetting data but included in the same superclass. In both tasks, we use 1 class data as the forgetting data.

To further understand the properties of unlearning in these tasks, we conducted additional experiments and summarized the results in Table 12. We can find the empirical results well demonstrate the conceptual conjectures in the previous discussion, and the representative baselines exhibit varied performance gap with the Retrained reference. Among them, our TARF can consistently achieve the better performance regarding to the Gap.

### D.4 SPECIFIC INFORMATION OF THE INSTANTIATED TASKS

For the four major scenarios (i.e., conventional all matched forgetting, target mismatch forgetting, model mismatch forgetting, and data mismatch forgetting) considered in our work, we provide the dataset construction and partition details in this section. Note that we focus on class-wise unlearning in this work, which is different from random data forgetting that uniformly samples the forgetting target of all classes in the training dataset.

**Forgetting target.** In previous works (Warnecke et al., 2023; Chen et al., 2023), the target concept to be forgotten is mainly considered as all matched where $\mathcal{D}_t = \mathcal{D}\{y = y_f\}$ has the same label domains (exactly same labels) with the pre-training task and forgetting data $\mathcal{D}_f = \mathcal{D}\{y = y_f\}$. In contrast, we assume that the target concept can be decoupled from the class label in practical unlearning requests. As illustrated in Figure 1, we further instantiate with three forgetting tasks given $\mathcal{D}_f = \mathcal{D}\{y = y_f\}$ with the superclass labels $\mathcal{Y}'$ of $\mathcal{Y}$ (classes): i) model mismatch forgetting, e.g., $\mathcal{D}_t = \mathcal{D}\{y = y_t\}$ and $y_t \subseteq y'_f$ where $y'_f \in \mathcal{Y}'$ given the model trained on $\mathcal{Y}'$; ii) target mismatch forgetting, e.g., $\mathcal{D}_t = \mathcal{D}\{y = y'_f\}$ given the model trained on $\mathcal{Y}$; iii) data mismatch forgetting, e.g., $\mathcal{D}_t = \mathcal{D}\{y = y'_f\}$ given the model trained on $\mathcal{Y}'$.

To ease the research investigation and empirical verification, we adopt the commonly used (Kurmanji et al., 2023; Jia et al., 2023; Fan et al., 2023; 2024) benchmark CIFAR-10 and CIFAR-100 for constructing the pre-training task for unlearning. Specifically, the official class labels are kept as classes for ordinary setup, and we provide the superclass information referring to the pre-defined lists (Krizhevsky, 2009) of CIFAR-100. Since there is no official superclass information for CIFAR-10 dataset, we manually grouped the classes of CIFAR-10 according to their semantic feature similarity and finalized 5 superclass clusters consisting of 2 classes in each. The full structured label layers information is summarized in Tables 14 and 15. For all the unlearning scenarios where the label domain of model output is the superclass, we will first use the superclass information to train the 20-class and 5-class classification models respectively. For the specific data partition in unlearning requests, we randomly sampled two classes in CIFAR-100 and one class in CIFAR-10 as forgetting data and kept the setup across the four forgetting tasks as well as other experiments. For other additional experimental setups, we will state them at the near positions.

Table 13: Basic setup about unlearning scenarios. More illustrations can be found in Appendix D.4.

| Dataset | Forgetting Data | Setup | All matched | Model mismatch | Target mismatch | Data mismatch |
|---------|----------------|-------|-------------|----------------|-----------------|---------------|
| CIFAR-10 | "automobile" | Training Class | 10 | 5 | 10 | 5 |
| | | Target Concept | "automobile" | "automobile" | "vehicle" | "vehicle" |
| CIFAR-100 | "boy", "girl" | Training Class | 100 | 20 | 100 | 20 |
| | | Target Concept | "boy", "girl" | "boy", "girl" | "people" | "people" |

Table 14: Full list of the 20-class classification on CIFAR-100 with its official superclass labels (Krizhevsky, 2009).

| Superclass (20) | Classes (5 for each superclass) |
|-----------------|--------------------------------|
| aquatic mammals | beaver, dolphin, otter, seal, whale |
| fish | aquarium fish, flatfish, ray, shark, trout |
| flowers | orchids, poppies, roses, sunflowers, tulips |
| food containers | bottles, bowls, cans, cups, plates |
| fruit and vegetables | apples, mushrooms, oranges, pears, sweet peppers |
| household electrical devices | clock, computer keyboard, lamp, telephone, television |
| household furniture | bed, chair, couch, table, wardrobe |
| insects | bee, beetle, butterfly, caterpillar, cockroach |
| large carnivores | bear, leopard, lion, tiger, wolf |
| large man-made outdoor things | bridge, castle, house, road, skyscraper |
| large natural outdoor scenes | cloud, forest, mountain, plain, sea |
| large omnivores and herbivores | camel, cattle, chimpanzee, elephant, kangaroo |
| medium-sized mammals | fox, porcupine, possum, raccoon, skunk |
| non-insect invertebrates | crab, lobster, snail, spider, worm |
| people | baby, boy, girl, man, woman |
| reptiles | crocodile, dinosaur, lizard, snake, turtle |
| small mammals | hamster, mouse, rabbit, shrew, squirrel |
| trees | maple, oak, palm, pine, willow |
| vehicles 1 | bicycle, bus, motorcycle, pickup truck, train |
| vehicles 2 | lawn-mower, rocket, streetcar, tank, tractor |

Table 15: Full list of the 5-class classification on CIFAR-10 with its manually set superclass (Krizhevsky, 2009).

| Superclass (5) | Classes (2 for each superclass) |
|----------------|--------------------------------|
| 1 | airplane, bird |
| 2 | automobile, truck |
| 3 | cat, dog |
| 4 | deer, frog |
| 5 | horse, ship |

Table 16: Specific training set data partition corresponding to four major forgetting tasks.

| Forgetting Tasks | Identified | Unidentified |
|------------------|-----------|--------------|
| All matched | $\mathcal{D}_f = \mathcal{D}_t$ | $\mathcal{D}_{un} = \mathcal{D}_r$ |
| Target mismatch | $\mathcal{D}_f \subset \mathcal{D}_t$ | $\mathcal{D}_{un} = \mathcal{D}_{fr} \cup \mathcal{D}_r$ |
| Model mismatch | $\mathcal{D}_f = \mathcal{D}_t$ | $\mathcal{D}_{un} = \mathcal{D}_r$ |
| Data mismatch | $\mathcal{D}_f \subset \mathcal{D}_t$ | $\mathcal{D}_{un} = \mathcal{D}_{fr} \cup \mathcal{D}_r$ |

## D.5 DISCUSSION ON THE PRACTICALITY OF LABEL DOMAIN MISMATCH

Machine unlearning is originally proposed in response to data regulations (Cao & Yang, 2015; Hoofnagle et al., 2019), which are primarily motivated by a desire to protect data owners' right to withdraw from the learning process. However, regarding its technical nature of mitigating the data influence from a trained model (Xu et al., 2023), unlearning is actually given broader significance in the context of trustworthy AI [4,5], like studies for mitigating bias and unfairness [6], addressing safety issues [7], erasing the NSFW generation [8,9]. It is worth noting that these trustworthy requirements may generally exhibit different concerns from the original training tasks. Motivated by the research problem raised in Section 3.1, our work focuses on a critical problem from the

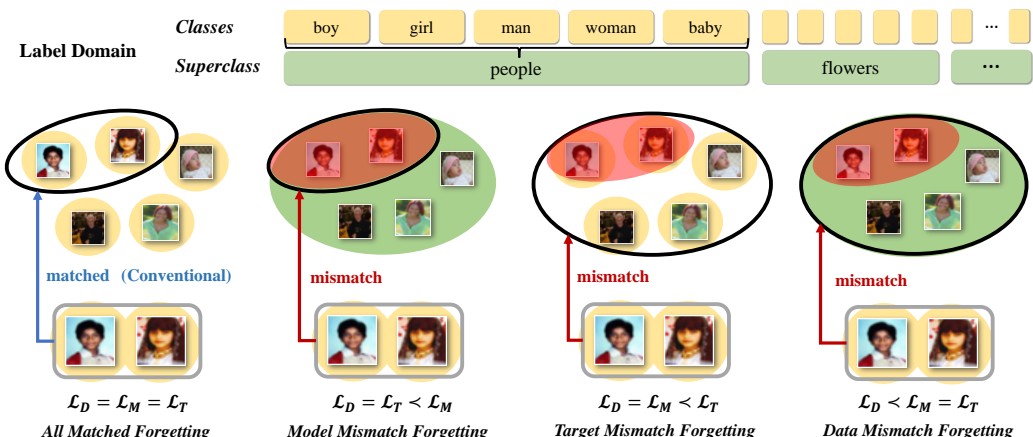

Taking the *CIFAR-100* (Krizhevsky, 2009) dataset, we instantiate four unlearning tasks given the same forgetting data with the class labels of "boy" and "girl": a) *all matched forgetting* (conventional scenario): unlearn "boy" and "girl" with the model trained on the classes; b) *target mismatch forgetting*: unlearn "people" with the model trained on the classes; c) *model mismatch forgetting*: unlearn "boy" and "girl" with the model trained on the superclass; d) *data mismatch forgetting*: unlearn "people" with the model trained on the superclass.

Figure 13: Illustrations of class representation with the four unlearning scenarios.

assumption view, i.e., the unlearning request may have a different taxonomy from the original tasks, for which we model the three mismatched scenarios for systematical exploration.

Here, we discuss some practical use cases for the three newly proposed settings. For example, 1) the label domain mismatch may exist in some recommendation tasks (Ge et al., 2022) or other generative tasks like image generation (Gandikota et al., 2023) with diversified user feedback (for which we have presented a case study on concept-forgetting on Stable Diffusion in Appendix E). 2) When the users raise unlearning requests for some representative disliked item with a message of "don't recommend this kind of thing", it is similar to target mismatch forgetting. In addition, 3) when debugging the pre-trained model with some spurious correlation or safety concerns (Bommasani et al., 2021; Gandikota et al., 2023; Li et al., 2024), it is similar to model or data mismatch forgetting as we only have the forgetting cases (e.g., some figures including NSFW content or adversarial features) that may not be aligned with the taxonomy of the pre-training task. We hope our exploration can provide insights for further consideration of specific practical applications.

### D.6 Discussion on the Scenario Commonalities and Framework Principles

The three mismatch scenarios, i.e., target mismatch forgetting, data mismatch forgetting, and model mismatch forgetting, share the common challenge of representation mismatch between the pre-trained model, the identified forgetting data, and the target concept to be forgotten. It breaks the assumption in all-matched scenarios that the three are matched (Shah et al., 2023; Xu et al., 2023; Fan et al., 2023; Jia et al., 2023) and can result in extra-/ineffective- forgetting in unlearning tasks, as demonstrated in our Figure 2. Specifically, in the target/data mismatch forgetting, the target concept can be wider than the identified forgetting data; while in the model mismatch forgetting, it can be smaller than the coarse-grained model representation.

To build a unified framework like our TARF, it requires considering the aforementioned two issues, i.e., insufficient representation in target/data mismatch, and decomposition lacking in the model mismatch. The former requires a flexible controller for forgetting strength while and latter requires a simultaneous consideration on forgetting and retaining. Thus, based on the general equation in Eq. (3), we set two sub-objectives (annealed forgetting and target-aware retaining) to decompose the learned representation and control the forgetting strength by the instance-wise weighting mechanism which selects the targeted-aware forgetting data. Then, TARF becomes a unified framework for the three mismatched scenarios. Note that although TARF is illustrated with three phases to better explain its functionality, while it is not three independent parts but unified in a general objective.

# E  ALGORITHM IMPLEMENTATION AND EXPLANATION

In this section, we present the pseudo-code of our proposed TARF and its variant, as well as additional discussions to enhance the understanding of our methods. Here we summarize the detailed procedure of algorithm implementation in Algorithm 1 and Algorithm 2. In detail, Algorithm 1 identifies the potential target using the class labels, while Algorithm 2 can use the instance level information.

As introduced in Section 3.3, the objective of our TARF is defined as follows,

$$L_{\text{TARF}} = \underbrace{k(t) \cdot \left( -\frac{1}{|\mathcal{D}_{\text{f}}|} \sum_{(x,y)\sim\mathcal{D}_{\text{f}}} \ell(f(x),y) \right)}_{\text{Annealed Forgetting } L_{\text{f}}(k)} + \underbrace{\frac{1}{|\mathcal{D}_{\text{un}}|} \sum_{(x,y)\sim\mathcal{D}_{\text{un}}} \ell(f(x),y) \cdot \tau(x,y,t)}_{\text{Target-aware Retaining } L_{\text{u}}(\tau)}, \tag{17}$$

where $k(t)$ and $\tau(x,y,t)$ are two training-time-related hyperparameters to deal with the mismatch issues raised in our new settings. Specifically, we set a learning-rate-reduced $k(t)$ as,

$$k(t) = k \cdot (T - t - t_0)/T, \quad t \in [0,T], \tag{18}$$

where $T$ indicates the total training time (e.g., epochs), and the value of $k(t)$ decreases with the training process. On the other hand, we have the following indicator to measure the model prediction consistency with the training data $I_{\text{con}}(x,y,\theta) = |\ell_{f_\theta}(x,y) - \ell_{f_{\theta^*}}(x,y)|$, with which we set $\tau(x,y,t)$ as follows,

$$\tau(x,y,t) = \begin{cases} 0 & I_{\text{con}}(x,y,\theta_{t_1}) > \beta \text{ or } t < t_1 \quad {}^*\text{Unconf. Retain,} \\ 1 & I_{\text{con}}(x,y,\theta_{t_1}) < \beta \text{ and } t \geq t_1 \quad {}^*\text{Conf. Retain,} \end{cases} \tag{19}$$

where $t_1$ is a time stamp to control the start of pursuing the retaining part. The overall two dynamic hyperparameters can divide the whole unlearning process into three phases as illustrated in Figure 4.

**Remark on $\beta$.** The intuition of setting $\beta$ is identifying those false remaining data in our Phase-I: Target Identification based on the gravity effects of forgetting dynamics. According to the dynamic information revealed by Phase-I (e.g., before $t_1$), $\beta$ is set to thresholding those data most influential by unlearning the given forgetting data, so is a computable value given the pre-assumed forgetting range. For the task

Table 17: Sensitive check of quantile-based choice on varied false-retain size using CIFAR-100.

| Forgetting Support | Size | UA | RA | TA | MIA | Gap |
|---|---|---|---|---|---|---|
| Retrained (Ref.) | 450 | 0.00 | 97.76 | 74.28 | 100.00 | - |
| GA (large) | 450 | 6.35 | 92.32 | 70.12 | 94.53 | 5.36 |
| TARF (large) | 450 | 0.00 | 96.42 | 72.13 | 100.00 | **0.87** |
| Retrained (Ref.) | 2250 | 0.00 | 98.03 | 73.42 | 100.00 | - |
| GA (Small) | 2250 | 35.07 | 91.81 | 66.39 | 75.91 | 18.10 |
| TARF (Small) | 2250 | 23.37 | 85.53 | 70.68 | 77.82 | **15.20** |

feasibility, we will generally assume the amount of false remaining data or classes is known at our target/data mismatch forgetting, following a similar setup in learning from label noise (Han et al., 2018) where the noise rate can be estimated and utilized as prior information. In the implementation, the $\beta$ value is estimated with the ranked accuracy difference of each class, once at time step $t_1$ (e.g., the end of Phase-I illustrated in Figure 4) and remain fixed afterward throughout training, as we already identify those potential false remaining data. Specifically, we estimate $\beta$ by ranking samples in the remaining set. For example, setting $\beta$ as the lowest $\mathcal{I}_{con}$ of top-5% data/class with the most loss/accuracy. In Table 17, we further check the performance robustness of quantile-based choice under varied false-retain size on CIFAR-100. When the false-retain set is very small, these data remain reliably captured by the top quantiles due to their strong semantic and representation alignment with the forgetting set, and the unlearning performance is good. In contrast, when the false-retain set is significantly enlarged, the ranking could include a number of noisy samples as their semantic similarity to the forgetting set is inherently weak. Although we can not achieve accurate forgetting (e.g., achieve 0% in UA), we can still perform better than plain gradient ascent on the given forgetting data. This situation also corresponds to the challenging settings where representation-gravity cues become ambiguous when the false-retain samples are disproportionately large and given forgetting data can not be representative anymore. Nevertheless, across all feasible mismatch scenarios introduced in the paper, the quantile-based ranking choice remains structurally robust, and quantile-based choice of $\beta$ can be valid if the forgetting dynamics can well capture the semantic representativeness.

**Annealed Forgetting.**  For the forgetting target, we adopt the gradient ascent on the given forgetting data to unlearn it. However, to approximate the retrained model, the intuition is not to pursue the maximization of the risk on this part of the data but to destroy the learned feature on the given model.

So we introduce a learning-rate-reduced $k(t)$ to realize the annealed gradient ascent where $t_0 = 1$ is adopted for target or data mismatch forgetting, and the value of $k(t)$ decreases with the training process. Resulting in destroyed features, gradient ascent on this part of data also constructs the dynamic information for differentiating the data of different consistency on its loss values, making the risks of the false retaining data higher than the rest, and helping to filter retaining data.

**Target-aware Retaining.** For the retaining part, we need to selectively learn the data from the remaining set, since the complementary dataset may be biased with unidentified forgetting data. Compared with other remaining data, the false retaining data is easy to be affected by similar feature representation as indicated in Figure 3. Thus, we can have $\tau(x, y, t)$ where we can divide the remaining set into unconfident/confident parts to note the estimated retaining data like Figure 5. $t_1 = 2$ is adopted at target and data mismatch tasks, and $\beta$ can be estimated by the prior information about the specific unlearning request and the rank of loss values. By simultaneously conducting gradient ascent on forgetting data and selective gradient descent on confident retaining data, we can better restrict the forgetting region and deconstruct the entangled feature representation (refer to the middle of Figure 5 where we reveal the feature decomposition in deeper layers of model structure using ResNet). Finally, with the partial objective of retaining, it can approximate the retrained reference (refer to the right of Figure 5).

### E.1 DISCUSSION ON THE FUNCTIONALITY OF HYPERPARAMETERS AND TUNING PRINCIPLE

**Discussion on the computational stability.** Conceptually, the hyperparameters introduced in TARF are structurally constrained by their functionality, although they introduced extra tuning flexibility to enable the capability of handling different mismatched scenarios. We can understand from an induction view of our unlearning objective, where $k$ and $\tau$ respectively control the strength of forgetting and the scope of retaining. Generalized from the Phase-II of target separation in Figure 4, $t_1$ and $t_0$ enable the Phase-I for target identification on target/data mismatch and the Phase-III for retaining approximation. Note that as discussed, $\beta$ is an automatic ranking threshold in realizing the index $\tau$. Thus, we only need to decide the proper value of $k$, $t_1$ and $t_0$, which is guided by specific unlearning scenarios. Given that intuition, we have several tuning principles: 1) the initial forgetting strength $k$ can be tuned from a smaller value to avoid extra feature distortion; 2) $t_1$ is generally set to be Epoch 1 as dynamic information can be captured by ranking mechanism; 3) $t_0$ can be also tuned by extending the Phase-III to fix the potential feature distortion induced by forgetting. Empirically, the above intuition and tuning principles benefit the computational stability of our framework, as demonstrated in our ablations which consistently shows that TARF is stable across a wide but reasonable range of choices. As shown in Figures 7 and 17, sweeping the values of initial $k$, $t_1$, and $t_0$ leads to highly similar outcomes unless: 1) $k$ is set to an unrealistically large value that aggressively destroys the representation; 2) $t_1$ is set to an extremely large value obscure large quota of retaining; 3) $t_0$ is set to near 0 without considering fixing representation destroy. This is governed by the forgetting dynamics revealed in our theoretical analysis, where the early steps dominate the separation of target representations, while the later can induce unrelated feature distortion.

**Practical guideline for hyperparameters.** Based on the aforementioned conceptual discussion, we can synthesize the guideline as: 1) $k$ should be initialized to a small value and increased cautiously; a modest early ascent is sufficient to reveal representation gravity, while overly large values cause unnecessary distortion; 2) $t_1$ can generally be set very early (typically Epoch 1), since the gravity-based ranking relies on the first few steps of forgetting. 3) $t_0$ can be tuned by extending Phase III when additional retaining approximation is needed, ensuring that any feature distortion is fixed.

### E.2 DISCUSSION ON THE ALGORITHM COMPUTATION COST

We would acknowledge that TARF may require more time in unlearning compared with some methods like GA, which only uses the forgetting data (which sometimes can be extremely limited than other retaining data) for unlearning, while those methods may suffer from excessive forgetting and results in inaccurate unlearning across different scenarios. Regarding the metric "TIME", it originally means to avoid some methods that consume too much time compared with that of Retrained (Ref.). From this perspective, these current methods and TARF actually fall in the acceptable time range, and the efficiency gap between existing explorations in that range is indeed not a bottleneck based on Table 1.

---

**Algorithm 1** TARF

**Input:** Training dataset $D = \{(x_i, y_i, s_i)\}_{i=1}^n$, where $s_i = 1$ indicates the identified forgetting dat, otherwise the data is recognized to be unlabeled for unlearning, learning rate $\eta$, number of epochs $T$, batch size $m$, number of batches $M$, data $x \in \mathcal{X}$, label $y \in \mathcal{Y}$, original trained model $\theta$, loss function $\ell$, initialized indicator value $\tau$ with the threshold $\beta$, time indicator $t_0$ and $t_1$ related to Eq. 5.
**Output:** model $\theta^T$;

1: **for** mini-batch $= 1, \ldots, M$ **do**
2:     Sample a mini-batch $\{(x_i, y_i)\}_{i=1}^m$ from $D$
3:     $\{\ell(x_i, y_i)\}_{i=1}^m \leftarrow \theta.\text{forward}(f_\theta, \{(x_i, y_i)\}_{i=1}^m)$,
4:     Collect the initial training accuracy in each class based on $\{\ell(x_i, y_i)\}_{i=1}^m$,
5:     $\tau \leftarrow 0$
6: **end for**
7: **for** epoch $= 0, \ldots, T$ **do**
8:     Update $k(t)$ according to Eq. 5,
9:     **if** epoch $= t_1$ **then**
10:         compute $\beta$ in Eq. 5 according to the rank of class accuracy difference, and update $\tau$.
11:     **end if**
12:     **for** mini-batch $= 1, \ldots, M$ **do**
13:         Sample a mini-batch $\{(x_i, y_i, s_i)\}_{i=1}^m$ from $D$
14:         Assign different weights for identified target samples and the rest retaining data,
15:         $L_{\text{TARF}} = k(t) \cdot \left( - \frac{1}{|\mathcal{D}_\text{f}|} \sum_{(x,y) \sim \mathcal{D}_\text{f}} \ell(f(x), y) \right) + \frac{1}{|\mathcal{D}_\text{un}|} \sum_{(x,y) \sim \mathcal{D}_\text{un}} \ell(f(x), y) \cdot \tau(x, y, t)$,
16:         $\theta \leftarrow \theta - \eta \nabla_\theta L_{\text{TARF}}(D, D_\text{f}, f, \tau)$
17:     **end for**
18: **end for**

---

From the methodology perspective, the three separately presented phases are integrated in a unified framework, instead of adding extra phases before and after Phase II. Compared to other approximate unlearning methods, the unique operation is target identification by comparing the output information of the unlearned

Table 18: Computational overhead of target identification compared with unlearning procedure.

| Method | TIME-I | TIME | UA | RA | TA | MIA | Gap |
|---|---|---|---|---|---|---|---|
| CIFAR-10 | 0.18 | 4.23 | 0.06 | 97.57 | 90.81 | 100.00 | 1.23 |
| CIFAR-100 | 0.18 | 4.85 | 0.31 | 97.35 | 73.68 | 100.00 | 0.21 |
| Tiny-ImageNet | 0.95 | 32.81 | 1.08 | 94.78 | 69.91 | 100.00 | 1.37 |
| ImageNet | 11.52 | 628.87 | 0.00 | 69.93 | 71.79 | 100.00 | 3.97 |

model with the original model for weight assignment, which has similar or less computation than other advanced designs that consider sparse regularization (Jia et al., 2023) or compute the gradient mask for the original model (Fan et al., 2023). Empirically, we check the computation overhead of identification in Table 18. Specifically, we report the computation time (min) of identification (TIME-In) including the forwarding pass and ranking operation, compared with that (TIME-Un) of the whole unlearning process. The resulting overhead is relatively limited compared with the unlearning cost across the datasets. While extremely large-scale settings could benefit from optional structural priors or sampling-based accelerations, our implementation already shows favorable scalability.

**Practical optimization strategies.** Considering the future application scenario like foundation-model-scale target identification, the single forward pass could also be costive for the entire remaining dataset for monitoring the dynamic information change. Here we further discuss potential optimization strategies. First, sampling-based approaches like uniform or stratified sampling are a viable strategy in large-scale applications, especially when the dataset contains structural information such as category tags or class labels. TARF depends only on relative changes in the gravity shift, rather than exact global estimates, so monitoring the shift using a small, class-balanced validation subset (e.g., a few thousand examples) is feasible. The primary trade-off is representational fidelity: if the sampled subset fails to capture the intra-class structural variation, the estimated gravity shift may introduce mild variance. Second, a lightweight surrogate model may be used to approximate gravity monitoring. Since TARF relies on representation gravity for target identification, we may also compute the gravity shift using a compressed encoder if available (e.g., a distilled representation model or low-rank projection of the backbone). This reduces monitoring costs by optimizing the forwarding pass, with the trade-off that the surrogate model may introduce slight bias in the absolute magnitude of the

---

**Algorithm 2** TARF-I: generalized version on instance-wise identification

---

**Input:** Training dataset $D = \{(x_i, y_i, s_i)\}_{i=1}^n$, where $s_i = 1$ indicates the identified forgetting dat, otherwise the data is recognized to be unlabeled for unlearning, learning rate $\eta$, number of epochs $T$, batch size $m$, number of batches $M$, data $x \in \mathcal{X}$, label $y \in \mathcal{Y}$, original trained model $\theta$, loss function $\ell$, initialized indicator value $\tau$ with the threshold $\beta$, time indicator $t_0$ and $t_1$ related to Eq. 5.
**Output:** model $\theta^T$;

1: **for** mini-batch $= 1, \ldots, M$ **do**
2:    Sample a mini-batch $\{(x_i, y_i)\}_{i=1}^m$ from $D$
3:    $\{\ell(x_i, y_i)\}_{i=1}^m \leftarrow \theta.\text{forward}(f_\theta, \{(x_i, y_i)\}_{i=1}^m)$,
4:    Collect the initial loss values in each training samples based on $\{\ell(x_i, y_i)\}_{i=1}^m$, $\tau \leftarrow 0$
5: **end for**
6: **for** epoch $= 0, \ldots, T$ **do**
7:    Update $k(t)$ according to Eq. 5,
8:    **if** epoch $= t_1$ **then**
9:       compute $\beta$ in Eq. 5 according to the rank of difference in instance loss values, and update $\tau$.
10:   **end if**
11:   **for** mini-batch $= 1, \ldots, M$ **do**
12:      Sample a mini-batch $\{(x_i, y_i, s_i)\}_{i=1}^m$ from $D$
13:      Assign different weights for identified target samples and the rest retaining data,
14:      $$L_{\text{TARF}} = k(t) \cdot \left( -\frac{1}{|\mathcal{D}_f|} \sum_{(x,y) \sim \mathcal{D}_f} \ell(f(x), y) \right) + \frac{1}{|\mathcal{D}_{\text{un}}|} \sum_{(x,y) \sim \mathcal{D}_{\text{un}}} \ell(f(x), y) \cdot \tau(x, y, t),$$
15:      $\theta \leftarrow \theta - \eta \nabla_\theta L_{\text{TARF}}(D, D_f, f, \tau)$
16:   **end for**
17: **end for**

---

gravity shift. We may also need consider the time cost of constructing such a surrogate model based on the existing literature. It could also be a promising future direction for efficient target identification.

### E.3 CASE STUDY FOR UNLEARNING GENERATION CONCEPT

To demonstrate the compatibility, we also extend the idea of this work and investigate the performance of TARF on the specific text-to-image generation task with stable diffusion (Gandikota et al., 2023; Fan et al., 2023), and presented the generated images by the original model and unlearned model in Tables 19 and 20.

In detail, we aim to unlearn the image generation of a concept with its specific prompt like "a photo of a tench". To simulate the practical unlearning request (e.g., the user raises the request of unlearning a specific concept with some identified generation examples, and the developer needs to adjust the model to forget the concept), we construct the given dataset consisting of limited forgetting data and the unidentified remaining data for unlearning, which corresponds to the data mismatch forgetting task. Then we compare the image generation on the original stable diffusion, the unlearned model with certain label (CL) mismatching (Fan et al., 2023), and that with our TARF. Note that here we recognize ESD (Gandikota et al., 2023) as a performance upper bound and do not compare it, since it is the same for all matched settings with fully identified forgetting data (as it directly encourages the model to unlearn the concept from text semantics). For this exploration of TARF, we adopt the instance-wise identification during the forgetting process as described in Algorithm 2, to unlearn the target concept with the given limited forgetting data and pursue retaining the selected remaining data with lower loss values.

The results in Tables 19 and 20 demonstrate that our TARF can achieve better forgetting results given the limited identified forgetting data, with proper target identification in the remaining set, while CL using only identified forgetting data can not unlearn the concept well as the generated examples still maintain some semantic features belongs to the target concept (like "tench" or "English springer").

### E.4 DISCUSSION ON TARF WITH LIMITED CLASS INFORMATION

The phase 1 of TARF is for target identification in the target mismatch forgetting where the target concept is wider than the given forgetting data (e.g., forgetting "people" given "boy" and "girl"). The

class information may affect the accurate identification of the target concept but not the rationality of our framework. In our experimental setup, the class information is available in TARF as the class labels are used in pre-training, while the information of the target concept is given by the number of extra classes instead of the superclass label. Regarding the unavailable or implicit class information, first, if the class (i.e., the subclasses w.r.t. target concept) is not available, TARF may also utilize the model prediction to obtain the pseudo labels to conduct the task; Second, if the extra forgetting target beyond the identified data is not restricted as classes, it may require that given forgetting data can well represent the target concept (e.g., the false retaining data should be easier affected than the other retaining data). We acknowledge that both scenarios would lead to a larger performance gap with Retrained reference, as it is a generally more challenging scenario affecting the task achievability to all of the approximate unlearning methods. We believe it worth future effort to explore.

### E.5 DISCUSSION ON THE ASSUMPTION OF SUPER-/SUB-CLASS INFORMATION

Though we provide the full hierarchical label structure in benchmarks like CIFAR-10/CIFAR-100 to enable controllable experiments for research purposes. It does assume some available structure or proxy signal to distinguish between the forgetting target and the retained knowledge. In practice, this can be: 1) class labels from user requests (e.g., "please unlearn boy/girl but not man/woman"); 2) semantic similarity (e.g., via pretrained embeddings or clustering in representation space); 3) model behaviors (e.g., gradients or output confidence shifts, used in our target-aware selection mechanism). Thus, our method is compatible with approximate or user-defined taxonomies as long as the information can reflect the representation similarity, and does not strictly require canonical super/sub-class structural information.

Table 19: Image generation results of unlearned Stable Diffusion in the **Data mismatch forgetting**, compared with the original stable diffusion, certain label (CL) unlearning (Fan et al., 2023), and our **TARF**. The specific prompt used in the image generation is "a photo of tench".

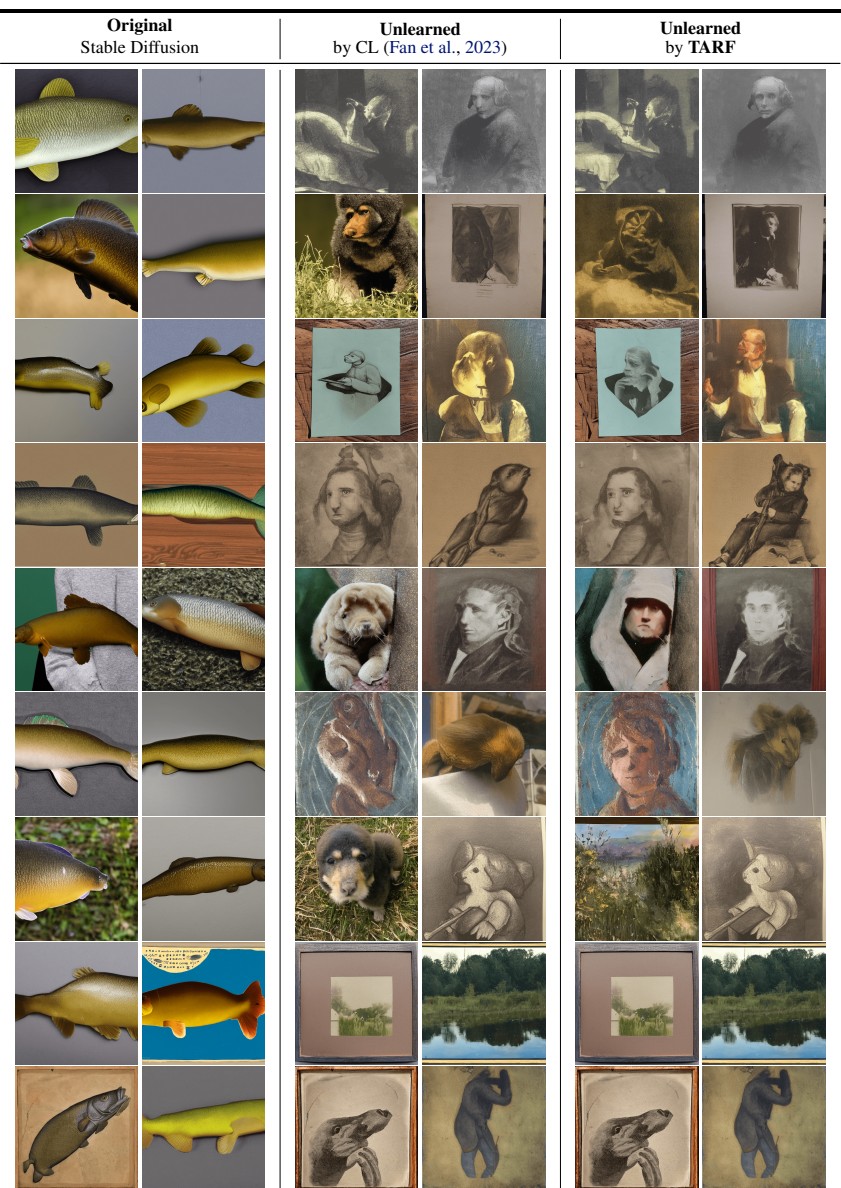

Table 20: Image generation results of unlearned Stable Diffusion in the **Data mismatch forgetting**, compared with the original stable diffusion, certain label (CL) unlearning (Fan et al., 2023), and our **TARF**. The specific prompt used in the image generation is "a photo of English springer".

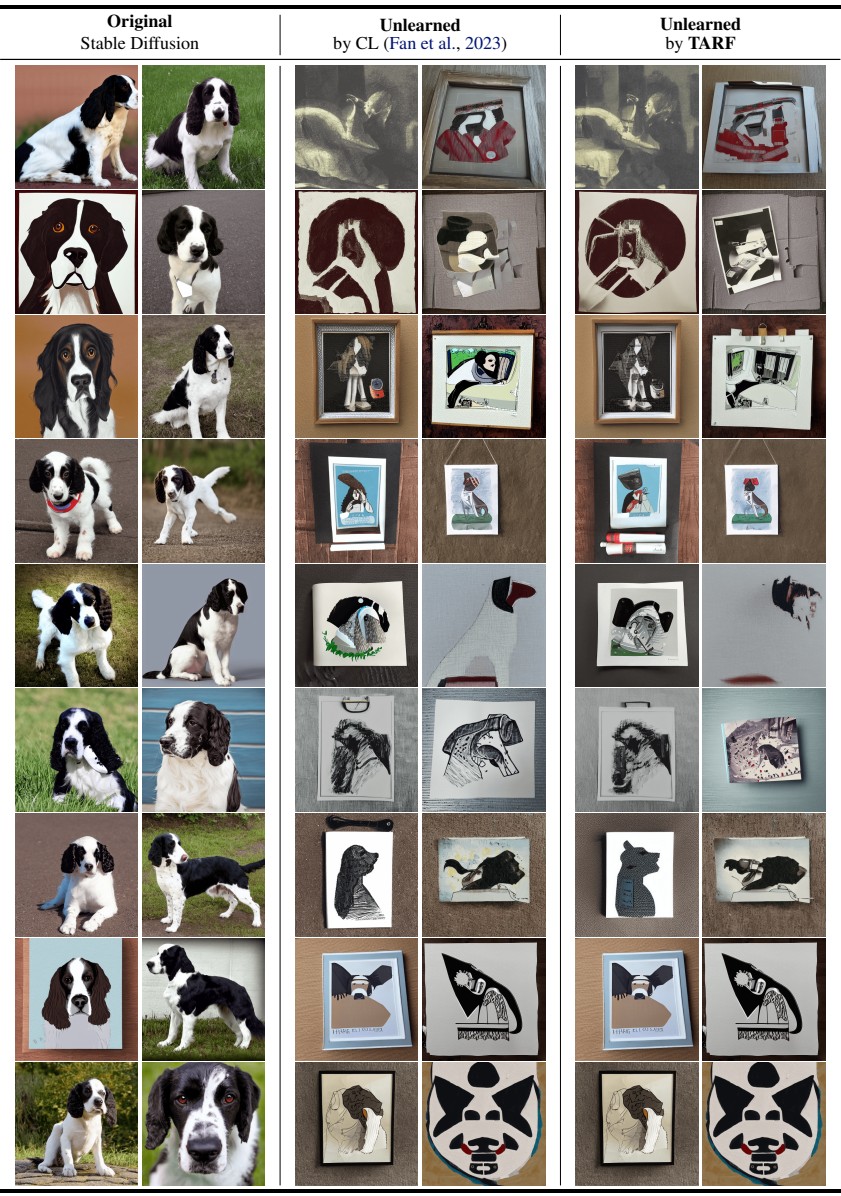

# F ADDITIONAL EXPERIMENTAL RESULTS

In this section, we provide additional experimental results of our work.

- In Appendix F.1, we summarize the additional experimental setups.

- In Appendix F.2, we discuss the crucial target identification in unlearning.

- In Appendix F.3, we discuss and compare TARF with the advanced method in all matched scenario.

- In Appendix F.4, we discuss potential ways to extend unlearning to the scenario without class labels.

- In Appendix F.5, we verify unlearning on large-scale datasets trained with large models.

- In Appendix F.6, we present unlearning with different model structures.

- In Appendix F.7, we present the full results under multiple runs with the four forgetting tasks.

- In Appendix F.8, we present the real-world unlearning application with LLM.

- In Appendix F.9, we discuss forgetting multiple class using TARF.

## F.1 EXTRA EXPERIMENTAL SETUPS

We introduce additional experimental details in the specific unlearning tasks. In our TARF, In general, we set $t_1 = 1$ for all the target identification parts, and we adopt $k = 0.04$, $t_0 = 2$ in model mismatch forgetting, and $k = 0.02$, $t_0 = 2$ for all matched, target mismatch and data mismatch forgetting in the unlearning request on CIFAR-10 classification task; for the CIFAR-100 classification task, we adopt $k = 0.5$, $t_0 = 2$ in model mismatch forgetting, and $k = 0.05$, $t_0 = 2$ for all matched, target mismatch and data mismatch forgetting. For the other hyperparameters, we follow the previous works (Jia et al., 2023; Kurmanji et al., 2023; Fan et al., 2023) to set the specific values. All the forgetting trails use 10 epochs for the total unlearning process except for GA (use 5 epochs) and IU (use the specific fixed step for optimization). The specific parameters and the pre-trained models (unlearn base) are provided in our source codes.

## F.2 DISCUSSION ABOUT TARGET IDENTIFICATION IN UNLEARNING

In this part, we further discuss the important factors for the achievability of the unlearning tasks. To be more specific, for the target or data mismatch forgetting, the scenario assumes that the identified forgetting data is part of the whole samples belonging to the target concept, which means there are other forgetting data included in the remaining set that need to be found. Thus, target identification is important for effective unlearning. As demonstrated in Section 3.2, the representation gravity can be a useful clue in forgetting dynamics to identify the other false retaining data. An implicit assumption is that those false retaining data have similar semantic features to the initially provided forgetting data, which has smaller representation distance than the retaining part of data as illustrated in Figure 14. Empirically, the model can have similar prediction changes on those false retaining data with the initial forgetting data. However, not all of the superclasses officially defined for the CIFAR-100 dataset are suitable for constructing the unlearning request, as some superclasses are not semantically separable like "aquatic mammals" and "fish". It can be found in Figure 16, where we check the Top-10 classes with the most accuracy changes after gradient ascent for each superclass in the CIFAR-100 dataset, some false retaining data (class-level indicated by blue arrows) are not easily identified given the two initially provided forgetting data classes (indicated by red arrows). One interesting future problem can be how to handle the spurious correlation given the insufficient representative samples.

## F.3 DISCUSSION ABOUT TARF ON ALL MATCHED SCENARIO

Regarding the all matched scenario, there is no need for the target identification part to identify extra forgetting data in the all-matched scenario as the target concept matches the forgetting data, then TARF degenerates into a general framework using the given forgetting data to forget, and the rest to retain. The performance of TARF is comparable to the existing best counterpart like SCRUB regarding the "Gap↓" in Table 3. It can be found that the overall performance of the unlearned models

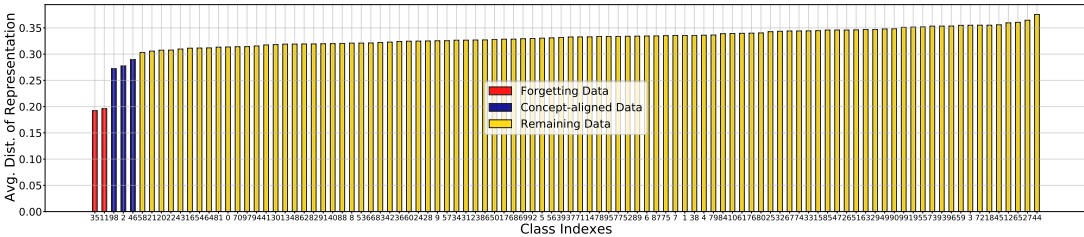

(a) Inter-classes distance in the model trained by classes

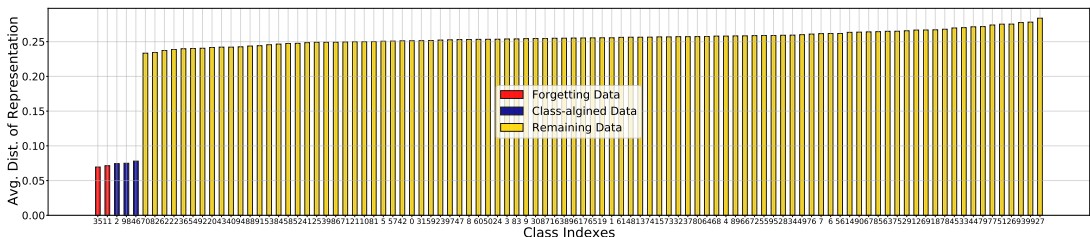

(b) Inter-superclass distance in the model trained by superclass

The distance is calculated at the feature representation extracted from the penultimate layer of the model for each class, which measures the averaged Euclidean distance to the cluster center (averaged by the forgetting data).

Figure 14: Inter-class distance and Inter-superclass distance for the unlearning assumption.

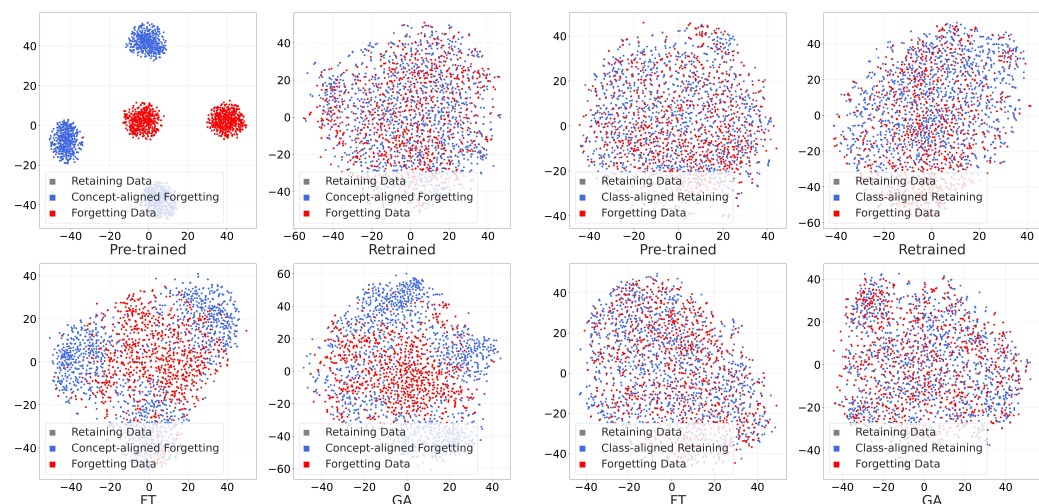

We present the tSNE visualization (Van der Maaten & Hinton, 2008) of the learned features, using two representative unlearning methods, i.e., finetune (FT) (Warnecke et al., 2023) and gradient ascent (GA) (Thudi et al., 2022a) with the pre-trained and retrained ones.

Figure 15: The entangled/under-entangled feature representations visualized by tSNE.

has already closely approximated the Retrained reference. Furthermore, since TARF is a general framework, we can also adopt the KL divergence loss with the original model as designed in SCRUB to further improve the performance, for which we present the comparison in Table 21.

## F.4 DISCUSSION ABOUT TARF ON WEAKLY-SUPERVISED SCENARIO

Our current work mainly focus on expanding the scope of conventional class-wise unlearning. Regarding the existing approximate unlearning studies (Shah et al., 2023; Kurmanji et al., 2023; Chen et al., 2023; Jia et al., 2023), considering the all matched forgetting scenario with full supervision, we

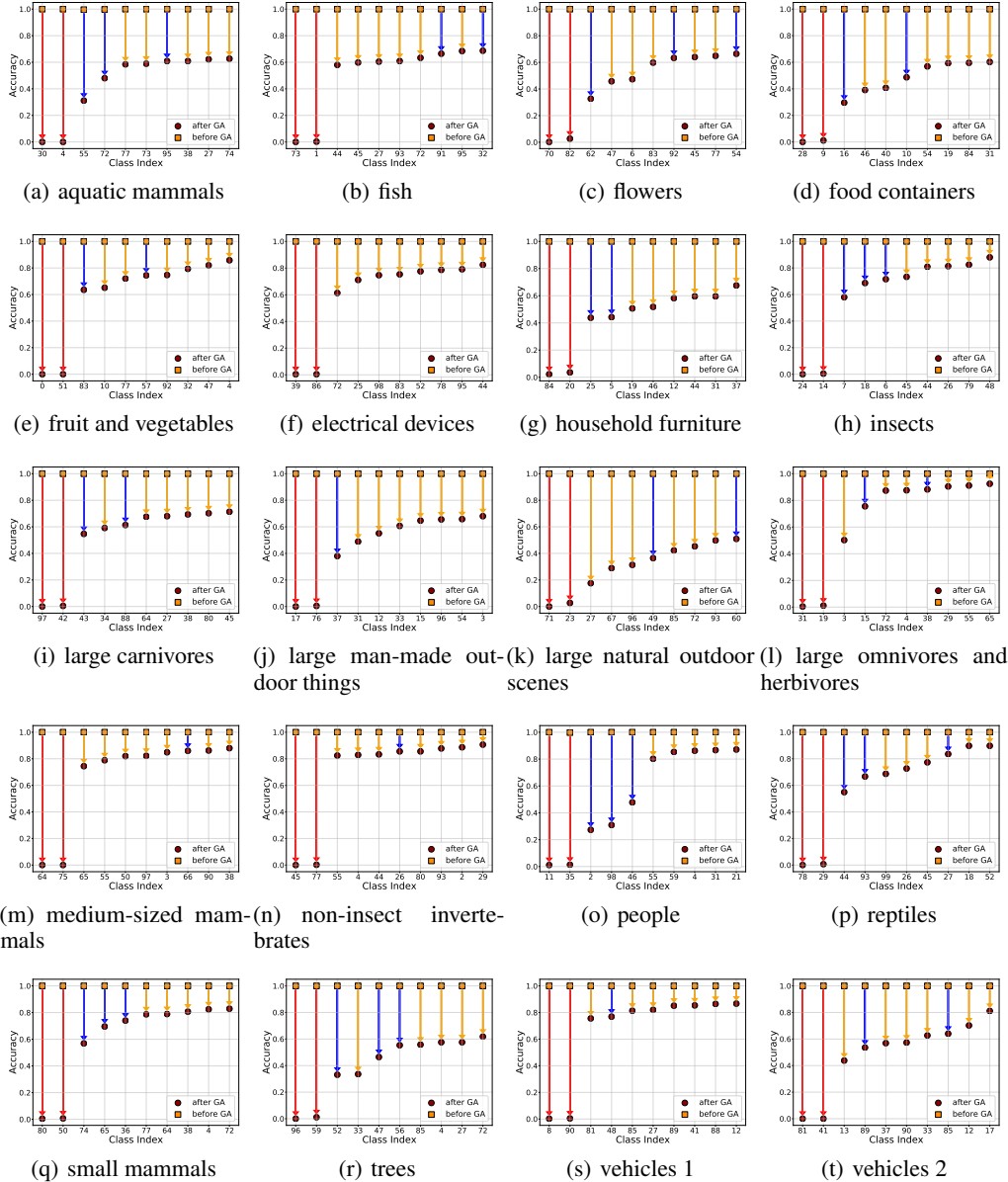

(a) aquatic mammals   (b) fish   (c) flowers   (d) food containers

(e) fruit and vegetables   (f) electrical devices   (g) household furniture   (h) insects

(i) large carnivores   (j) large man-made outdoor things   (k) large natural outdoor scenes   (l) large omnivores and herbivores

(m) medium-sized mammals   (n) non-insect invertebrates   (o) people   (p) reptiles

(q) small mammals   (r) trees   (s) vehicles 1   (t) vehicles 2

Target identification results with different unlearning requests and the minimum identified forgetting data on the CIFAR-100 dataset. Note that some target concepts are not successfully identified by the identified data.

Figure 16: Task Identification using the CIFAR-100 dataset for target mismatch forgetting.

push it towards more practical settings via decoupling the class labels and the target concept. For machine unlearning under weak supervision, there are limited studies (Shah et al., 2023) to our best knowledge, and we believe it is worth an in-depth exploration in future work.

Given that if a model is trained with semi-supervised or other weak supervision, we can obtain the pseudo labels by the model prediction for its unlearning phase. Instead of using the predicted label, we can also utilize the distillation objective to encourage the unlearned model's output to be far away from (or close to) the original ones. With the guide of model prediction, the data belonging to the same superclass with the forgetting data can be figured out to constrain the unlearning target. In Table 22, we present the results of our methods when only the given forgetting data are labeled, demonstrating our framework can be extended to achieve satisfactory performance.

Table 21: Performance comparison in the all matched scenario when TARF with CE loss/KL divergence (refer to Eq. (14)) with the original model for the retaining part.

| Type / $\mathcal{D}$ | Dataset | CIFAR-10 | | | | | | CIFAR-100 | | | | | |
|---|---|---|---|---|---|---|---|---|---|---|---|---|---|
| | Method / Metrics | UA | RA | TA | MIA | Gap↓ | TIME↓ | UA | RA | TA | MIA | Gap↓ | TIME↓ |
| | Retrained (Ref.) | 0.00 | 99.51 | 94.69 | 100.00 | - | 43.3 | 0.00 | 97.85 | 76.03 | 100.00 | - | 43.2 |
| | FT (Warnecke et al., 2023) | 1.07 | 98.62 | 92.36 | 100.00 | 1.07 | 4.43 | 0.67 | 96.32 | 72.34 | 100.00 | 1.47 | 5.02 |
| | SCRUB (Kurmanji et al., 2023) | 0.00 | 99.94 | 91.00 | 100.00 | 1.03 | 2.88 | 0.00 | 99.98 | 76.75 | 100.00 | 0.71 | 3.23 |
| Semi-supervised Scenarios | TARF (with CE) | 0.00 | 98.23 | 91.95 | 100.00 | 1.01 | 4.21 | 0.00 | 96.90 | 72.53 | 100.00 | 1.11 | 4.68 |
| | TARF (with KL) | 0.00 | 98.81 | 93.33 | 100.00 | **0.52** | 4.32 | 0.00 | 96.95 | 74.98 | 100.00 | **0.49** | 4.89 |

Table 22: A case study (%) on the unlearning on CIFAR-100 under the weakly-supervised scenario (e.g., using the pseudo-label generated by model prediction to handle unlabeled retaining data).

| Type / $\mathcal{D}$ | Dataset | Model mismatch | | | | | | Data mismatch | | | | | |
|---|---|---|---|---|---|---|---|---|---|---|---|---|---|
| | Method / Metrics | UA | RA | TA | MIA | Gap↓ | TIME↓ | UA | RA | TA | MIA | Gap↓ | TIME↓ |
| | Retrained | 88.22 | 98.58 | 78.50 | 25.78 | - | 43.8 | 0.00 | 98.50 | 80.15 | 100.00 | - | 53.2 |
| | FT | 92.67 | 95.02 | 79.34 | 16.33 | 4.58 | 4.86 | 82.62 | 95.66 | 79.77 | 37.24 | 37.15 | 4.93 |
| | RL | 80.11 | 95.83 | 79.83 | 99.00 | 21.35 | 4.93 | 89.78 | 96.82 | 79.90 | 70.76 | 30.49 | 4.97 |
| Semi-supervised Scenarios | GA | 6.78 | 94.83 | 76.96 | 97.78 | 39.68 | 0.06 | 6.00 | 97.65 | 79.23 | 98.04 | 2.43 | 0.05 |
| | BS | 18.11 | 95.90 | 72.28 | 95.22 | 37.14 | 0.89 | 15.38 | 98.50 | 78.50 | 96.22 | 6.76 | 0.96 |
| | $L_1$-sparse | 82.11 | 85.17 | 75.22 | 20.00 | 7.15 | 5.00 | 84.53 | 85.13 | 75.22 | 17.02 | 46.45 | 5.03 |
| | TARF (full labels) | 86.67 | 97.05 | 80.07 | 26.00 | **1.21** | 4.81 | 0.00 | 95.01 | 78.98 | 100.00 | **1.17** | 4.78 |
| | TARF (unlabeled retain) | 90.22 | 96.58 | 80.01 | 22.54 | 2.17 | 4.84 | 1.33 | 95.30 | 78.12 | 99.34 | 1.45 | 4.85 |

Table 23: Results (%). Comparison with the baselines on TinyImageNet trained on ResNet101. (More results on large-scale dataset like ImageNet can refer to Appendix F.5)

| Type / $\mathcal{D}$ | Dataset | All matched | | | | | | Model mismatch | | | | | |
|---|---|---|---|---|---|---|---|---|---|---|---|---|---|
| | Method / Metrics | UA | RA | TA | MIA | Gap↓ | TIME↓ | UA | RA | TA | MIA | Gap↓ | TIME↓ |
| | Retrained (Ref.) | 0.00 | 74.32 | 63.13 | 100.00 | - | 217.0 | 34.80 | 71.26 | 64.29 | 66.90 | - | 256.14 |
| | FT (Warnecke et al., 2023) | 3.80 | 77.66 | 62.98 | 97.30 | 2.50 | 30.41 | 59.30 | 77.26 | 62.92 | 38.00 | 15.19 | 37.44 |
| | RL (Toneva et al., 2018) | 73.20 | 69.87 | 60.49 | 18.40 | 40.47 | 225.13 | 84.10 | 68.53 | 60.63 | 8.00 | 28.64 | 226.79 |
| | GA (Ishida et al., 2020) | 5.70 | 63.26 | 57.09 | 87.50 | 8.83 | 0.34 | 6.30 | 63.17 | 58.04 | 90.70 | 16.66 | 0.34 |
| | BS (Chen et al., 2023) | 0.30 | 43.96 | 40.23 | 97.70 | 13.97 | 1.2 | 0.10 | 33.94 | 31.82 | 99.10 | 34.17 | 0.62 |
| | $L_1$-sparse (Jia et al., 2023) | 3.70 | 76.63 | 62.55 | 72.00 | 2.28 | 40.79 | 59.40 | 76.30 | 62.80 | 38.80 | 14.81 | 37.05 |
| | SCRUB (Kurmanji et al., 2023) | 0.00 | 75.06 | 63.82 | 100.00 | **0.36** | 66.69 | 37.70 | 73.89 | 64.20 | 57.30 | 3.81 | 58.53 |
| | TARF (ours) | 0.00 | 75.47 | 62.79 | 100.00 | 0.37 | 28.22 | 34.00 | 74.28 | 62.60 | 65.00 | **1.85** | 38.21 |
| | Dataset | Target matched | | | | | | Data mismatch | | | | | |
| | Method / Metrics | UA | RA | TA | MIA | Gap↓ | TIME↓ | UA | RA | TA | MIA | Gap↓ | TIME↓ |
| | Retrained (Ref.) | 0.00 | 72.83 | 65.12 | 100.00 | - | 213.05 | 0.00 | 71.37 | 65.76 | 100.00 | - | 252.62 |
| | FT (Warnecke et al., 2023) | 29.67 | 75.94 | 62.97 | 69.30 | 16.41 | 30.41 | 64.33 | 75.45 | 62.96 | 30.60 | 35.15 | 37.44 |
| Tiny ImageNet | RL (Toneva et al., 2018) | 68.93 | 69.97 | 60.55 | 22.00 | 38.59 | 225.13 | 84.27 | 68.64 | 60.59 | 7.86 | 46.08 | 226.79 |
| | GA (Ishida et al., 2020) | 11.33 | 63.63 | 57.26 | 81.00 | 11.85 | 0.34 | 7.33 | 63.44 | 58.24 | 89.80 | 8.25 | 0.34 |
| | BS (Chen et al., 2023) | 1.00 | 44.00 | 40.42 | 96.70 | 14.46 | 1.2 | 0.00 | 34.10 | 31.98 | 99.30 | 17.94 | 0.62 |
| | $L_1$-sparse (Jia et al., 2023) | 28.93 | 75.18 | 62.55 | 69.60 | 16.06 | 40.79 | 63.90 | 74.80 | 62.80 | 31.30 | 34.75 | 37.05 |
| | SCRUB (Kurmanji et al., 2023) | 25.67 | 75.31 | 63.85 | 73.80 | 13.90 | 66.69 | 44.07 | 74.02 | 64.25 | 46.93 | 25.33 | 58.53 |
| | TARF (ours) | 5.07 | 75.78 | 62.72 | 97.53 | **3.22** | 32.81 | 0.00 | 74.85 | 62.59 | 100.00 | **1.66** | 37.92 |

## F.5 FORGETTING IN THE LARGE-SCALE DATASET

In this part, we present more experiments conducted on large-scale dataset like ImageNet-1k in Table 24, and also unlearning multiple classes in the large-scale datasets in Table 25.

## F.6 FORGETTING WITH DIFFERENT MODEL STRUCTURES

In this part, we further check the unlearning performance of our TARF on different pre-trained model structures compared with several baselines. We choose CIFAR-100 as the pre-training classification task and conduct all matched forgetting and model mismatch forgetting. The results are summarized in Table 26. The results validate that our TARF can robustly achieve better unlearning performance across different model structures.

## F.7 FULL RESULTS WITH DIFFERENT FORGETTING TASKS

In this section, we provide the full results of Table 3, which is conducted by setting different random seeds (for multiple runs) with the original trails and reported as the mean and std values for each evaluation metric. Tables 27 to 30 presents the performance of unlearning on CIFAR-10, and Tables 31 to 34 presents the performance of unlearning on CIFAR-100. The performance comparison of our

Table 24: Results (%). Comparison with the unlearning baselines on ImageNet-1k. All matched forgetting: unlearn 1 class; Target mismatch forgetting: unlearn three classes belonging to "fish".

| Type / $\mathcal{D}$ | Dataset | All matched | | | | | | Target mismatch | | | | | |
|---|---|---|---|---|---|---|---|---|---|---|---|---|---|
| | Method / Metrics | UA | RA | TA | MIA | Gap↓ | TIME↓ | UA | RA | TA | MIA | Gap↓ | TIME↓ |
| | Retrained | 0.00 | 79.77 | 77.64 | 100.00 | - | 7075.48 | 0.00 | 80.09 | 77.54 | 100.00 | - | 7777.54 |
| | FT (Warnecke et al., 2023) | 0.00 | 70.18 | 71.98 | 100.00 | 3.82 | 608.11 | 0.79 | 70.26 | 72.07 | 100.00 | 4.02 | 608.62 |
| | RL (Toneva et al., 2018) | 81.38 | 70.22 | 71.79 | 19.46 | 44.29 | 969.44 | 79.69 | 69.98 | 71.77 | 23.03 | 43.14 | 972.02 |
| | GA (Ishida et al., 2020) | 0.00 | 66.25 | 67.36 | 100.00 | 5.95 | 8.76 | 0.00 | 31.21 | 37.74 | 0.00 | 47.17 | 17.38 |
| | BS (Chen et al., 2023) | 0.00 | 31.15 | 36.33 | 100.00 | 22.48 | 9.03 | 0.00 | 21.57 | 27.56 | 99.97 | 27.13 | 23.75 |
| | $L_1$-sparse (Jia et al., 2023) | 0.00 | 67.98 | 70.70 | 100.00 | 4.68 | 603.21 | 0.00 | 67.24 | 70.28 | 100.00 | 5.03 | 601.27 |
| | SCRUB (Kurmanji et al., 2023) | 29.77 | 74.92 | 75.66 | 81.77 | 13.71 | 655.42 | 22.44 | 74.87 | 75.60 | 82.77 | 11.71 | 681.53 |
| | **TARF** (ours) | 0.00 | 70.53 | 72.23 | 100.00 | **3.66** | 600.11 | 0.00 | 69.93 | 71.79 | 100.00 | **3.97** | 628.87 |
| **ImageNet-1k** | Dataset | Model matched | | | | | | Data mismatch | | | | | |
| | Method / Metrics | UA | RA | TA | MIA | Gap↓ | TIME↓ | UA | RA | TA | MIA | Gap↓ | TIME↓ |
| | Retrained | 79.15 | 80.00 | 70.29 | 25.69 | - | 6501.27 | 0.00 | 80.36 | 70.38 | 100.00 | - | 6493.16 |
| | FT (Warnecke et al., 2023) | 83.31 | 70.38 | 64.05 | 19.00 | 6.68 | 695.42 | 0.00 | 69.99 | 63.76 | 100.00 | 4.24 | 693.18 |
| | RL (Toneva et al., 2018) | 87.62 | 69.43 | 63.26 | 15.23 | 9.13 | 959.84 | 88.21 | 70.33 | 63.81 | 12.21 | 48.15 | 956.13 |
| | GA (Ishida et al., 2020) | 0.00 | 66.62 | 58.91 | 100.00 | 44.56 | 17.44 | 0.00 | 15.35 | 14.34 | 0.00 | 55.26 | 17.58 |
| | BS (Chen et al., 2023) | 0.00 | 45.81 | 40.84 | 100.00 | 54.28 | 19.69 | 0.00 | 13.00 | 12.10 | 100.00 | 31.41 | 23.70 |
| | $L_1$-sparse (Jia et al., 2023) | 82.00 | 67.94 | 62.58 | 19.15 | 7.29 | 1091.29 | 0.00 | 66.37 | 61.03 | 100.00 | 5.84 | 1071.41 |
| | SCRUB (Kurmanji et al., 2023) | 86.08 | 74.82 | 68.04 | 14.69 | 6.34 | 663.61 | 14.18 | 74.84 | 67.92 | 93.10 | 7.27 | 689.82 |
| | **TARF** (ours) | 80.62 | 70.27 | 64.04 | 19.46 | **5.92** | 601.28 | 0.00 | 70.10 | 63.97 | 100.00 | **4.17** | 602.62 |

Table 25: Results (%). Comparison with the unlearning baselines on TinyImageNet-200 and ImageNet-1k with more (10+) forgetting classes in all matched forgetting scenarios.

| Scenarios / $\mathcal{D}$ | Unlearn request | forget 10 classes in Tiny-ImageNet | | | | | | forget 30 classes in Tiny-ImageNet | | | | | |
|---|---|---|---|---|---|---|---|---|---|---|---|---|---|
| | Method / Metrics | UA | RA | TA | MIA | Gap↓ | TIME↓ | UA | RA | TA | MIA | Gap↓ | TIME↓ |
| | Retrained | 0.00 | 71.00 | 60.29 | 100.00 | - | 251.43 | 0.00 | 65.26 | 57.60 | 100.00 | - | 181.13 |
| | FT | 2.04 | 70.63 | 59.04 | 98.26 | 1.35 | 27.10 | 2.79 | 72.41 | 60.36 | 97.38 | 3.71 | 35.00 |
| | GA | 17.76 | 61.74 | 56.12 | 76.90 | 13.57 | 1.37 | 28.95 | 59.72 | 57.54 | 57.06 | 19.37 | 3.49 |
| | **TARF** (ours) | 0.00 | 69.63 | 59.69 | 100.00 | **0.49** | 28.5 | 0.00 | 70.24 | 60.16 | 100.00 | **1.89** | 39.6 |
| **All matched Forgetting** | Unlearn request | forget 50 classes in Tiny-ImageNet | | | | | | forget 10 classes in ImageNet | | | | | |
| | Method / Metrics | UA | RA | TA | MIA | Gap↓ | TIME↓ | UA | RA | TA | MIA | Gap↓ | TIME↓ |
| | Retrained | 0.00 | 66.26 | 57.88 | 100.00 | - | 161.37 | 0.00 | 51.94 | 56.74 | 100.00 | - | 917.66 |
| | FT | 5.19 | 75.77 | 61.29 | 85.75 | 8.09 | 44.62 | 0.00 | 55.16 | 59.53 | 100.00 | 1.50 | 316.14 |
| | GA | 22.92 | 44.12 | 48.03 | 62.26 | 23.16 | 7.70 | 5.73 | 47.35 | 52.42 | 87.21 | 6.85 | 2.18 |
| | **TARF** (ours) | 0.00 | 71.68 | 60.89 | 100.00 | **2.11** | 46.97 | 0.00 | 50.69 | 55.83 | 100.00 | **0.54** | 353.69 |

TARF with other baseline across the four forgetting tasks (i.e., all matched, target mismatch, model mismatch, and data mismatch) demonstrated the general effectiveness of our algorithm framework.

## F.8 APPLICATION ON MISMATCHED FORGETTING WITH LLM

In Table 5, we adapt our introduced four mismatch setting under the context of Large Language Models (LLMs), we conduct experiments on the TOFU (Maini et al., 2024) dataset for real-world application on removing learned authors information which are private and IP-related. We modify the original TOFU forget set to our scenarios similarly as previous construction on conventional benchmark: specifically, the all matched setting we will have all the forgetting data while the data mismatch setting we use 80% identified forgetting data. Since there is no explicit concept in the context of TOFU target, we assume the amount of forget data implicitly represent the representation mismatch with original LLM (e.g., using forget01 to represent easy to cluster set and forget10 to represent hard to cluster set for different unlearn difficulties). The model adopt is LLama-3.2 and the evaluation metrics used is QA probability on forgetting and retaining set following (Maini et al., 2024). Given the complex LLM representation, previous representative methods like GA (Maini et al., 2024) and NPO (Zhang et al., 2024) can easily make the model collapse to achieve low Prob on both forget and retain set. While our TARF can perform robustly to achieve forgetting with better retaining performance resistance, we also find there are some trade-off between forget and retain part, which indicates complex entangled representation on it.

## F.9 FORGETTING MULTIPLE CLASS BY TARF

Our proposed TARF framework is applicable to forgetting multiple concepts or superclasses simultaneously, as we didn't restrict concept numbers in our algorithm design. Conceptually, the core challenges of mismathced unlearning still lie on insufficient representaiton or decomposition lacking

Table 26: Results (%) of unlearning with different model structure. All methods are trained on the same backbone, i.e., the basis of unlearning initialization is the same (except for retraining from scratch). Values are percentages. Bold numbers are superior results. ↓ indicates smaller are better.

| CIFAR-100 | Task | All matched | | | | | Model mismatch | | | | |
|---|---|---|---|---|---|---|---|---|---|---|---|
| | Metric | UA | RA | TA | MIA | Gap↓ | UA | RA | TA | MIA | Gap↓ |
| **VGG-19** | Retrained | 0.00 | 97.26 | 73.13 | 100.00 | - | 87.44 | 98.22 | 82.12 | 19.89 | - |
| | FT (Warnecke et al., 2023) | 0.00 | 90.92 | 66.86 | 100.00 | 3.15 | 95.22 | 95.17 | 77.71 | 7.56 | 6.89 |
| | RL (Toneva et al., 2018) | 0.00 | 90.29 | 66.16 | 100.00 | 3.48 | 96.22 | 95.26 | 77.71 | 98.56 | 23.71 |
| | GA (Ishida et al., 2020) | 0.00 | 79.27 | 56.03 | 100.00 | 8.77 | 0.00 | 93.09 | 74.30 | 100.00 | 45.13 |
| | **TARF (ours)** | 0.00 | 91.96 | 67.94 | 100.00 | **2.62** | 82.67 | 93.71 | 76.24 | 24.22 | **4.87** |
| **ResNet-18** | Retrained | 0.00 | 97.85 | 76.03 | 100.00 | - | 88.22 | 98.58 | 78.50 | 25.78 | - |
| | FT (Warnecke et al., 2023) | 0.66 | 96.55 | 71.97 | 100.00 | 1.51 | 98.22 | 96.79 | 80.14 | 6.78 | 8.11 |
| | RL (Toneva et al., 2018) | 0.11 | 95.90 | 71.57 | 100.00 | 1.63 | 94.11 | 96.70 | 80.17 | 96.89 | 20.14 |
| | GA (Ishida et al., 2020) | 1.89 | 95.26 | 69.14 | 99.89 | 2.87 | 9.33 | 95.13 | 77.22 | 96.89 | 38.68 |
| | **TARF (ours)** | 0.00 | 96.90 | 71.51 | 100.00 | **1.37** | 86.00 | 96.54 | 74.20 | 22.78 | **2.89** |
| **WideResNet** | Retrained | 0.00 | 97.71 | 76.95 | 100.00 | - | 88.11 | 98.37 | 83.61 | 23.56 | - |
| | FT (Warnecke et al., 2023) | 0.67 | 96.61 | 71.29 | 100.00 | 1.86 | 97.44 | 95.70 | 78.70 | 7.33 | 8.29 |
| | RL (Toneva et al., 2018) | 0.00 | 95.86 | 71.36 | 100.00 | 1.86 | 85.77 | 94.69 | 78.26 | 96.00 | 20.95 |
| | GA (Ishida et al., 2020) | 0.44 | 91.49 | 66.29 | 100.00 | 2.26 | 4.33 | 91.76 | 75.18 | 99.11 | 43.71 |
| | **TARF (ours)** | 0.00 | 96.51 | 71.77 | 100.00 | **1.60** | 88.00 | 95.50 | 79.06 | 22.67 | **2.11** |

Table 27: Main Results (%). Comparison with the unlearning baselines. All methods are trained on the same backbone, i.e., the basis of unlearning initialization is the same (except for retraining from scratch). Values are percentages. Bold numbers are superior results. ↓ indicates smaller are better.

| CIFAR-10 | Metric | UA | | RA | | TA | | MIA | | Gap↓ | |
|---|---|---|---|---|---|---|---|---|---|---|---|
| | Method | mean | std | mean | std | mean | std | mean | std | mean | std |
| **All matched** | Retrained | 0.00 | - | 99.51 | - | 94.69 | - | 100.00 | - | - | - |
| | FT (Warnecke et al., 2023) | 4.66 | 3.59 | 98.58 | 0.04 | 92.42 | 0.06 | 100.00 | 0.00 | 1.96 | 0.89 |
| | RL (Toneva et al., 2018) | 2.23 | 1.90 | 98.30 | 0.65 | 91.97 | 0.74 | 100.00 | 0.00 | 1.54 | 0.82 |
| | GA (Ishida et al., 2020) | 0.34 | 0.16 | 95.48 | 0.24 | 88.52 | 0.35 | 99.88 | 0.10 | 2.67 | 0.21 |
| | IU (Izzo et al., 2021) | 0.11 | 0.05 | 72.50 | 15.65 | 68.28 | 14.10 | 99.98 | 0.02 | 13.39 | 7.41 |
| | BS (Chen et al., 2023) | 24.72 | 0.32 | 88.91 | 0.97 | 81.84 | 0.94 | 89.23 | 0.56 | 14.74 | 0.70 |
| | $L_1$-sparse (Jia et al., 2023) | 0.00 | 0.00 | 94.18 | 0.03 | 90.01 | 0.24 | 100.00 | 0.00 | 2.50 | 0.05 |
| | SalUn (Fan et al., 2023) | 0.48 | 0.46 | 88.66 | 2.67 | 84.48 | 2.40 | 100.00 | 0.00 | 5.39 | 1.38 |
| | SCRUB (Kurmanji et al., 2023) | 1.23 | 0.58 | 99.92 | 0.02 | 91.23 | 0.56 | 100.00 | 0.00 | 1.28 | 0.23 |
| | **TARF (ours)** | 0.00 | 0.00 | 98.22 | 0.02 | 92.09 | 0.14 | 100.00 | 0.00 | **0.97** | 0.03 |

as revealed in our Section 3.2. For the model mismatch setting, it doesn't introduce extra algorithmic di"culty with multiple target concepts, as our objective simultaneously considers gradient ascent and descent to deconstruct the entangled representation. For the target/data mismatch setting, we can also identify those multiple target concepts by respectively utilizing our Phase-I: Target Identification with the base model, given the forgetting data as a representative support set for each concept.

To present the multiple-superclass forgetting, we also included new experiments in Table 36, where we conduct unlearning in the target mismatch and model mismatch settings using the CIFAR-100 dataset for two target concepts forgetting, e.g., "people" and "aquatic mammals". As a result, TARF demonstrates consistent performance effectiveness across all scenarios without harming the retention on retaining data, indicating its scalability to multi-concept forgetting.

Table 36: Forgetting Multiple Superclass by TARF using CIFAR-100.

| Method | UA | RA | TA | MIA | Gap |
|---|---|---|---|---|---|
| Retrained (Ref.) | 0.00 | 97.23 | 71.85 | 100.00 | - |
| FT | 45.68 | 94.65 | 71.16 | 58.87 | 22.52 |
| RL | 49.91 | 96.12 | 72.14 | 55.96 | 23.84 |
| GA | 18.07 | 92.62 | 67.75 | 91.29 | 8.88 |
| $L_1$-sparse | 43.40 | 87.68 | 68.90 | 60.31 | 23.90 |
| SCRUB | 50.55 | 99.61 | 78.91 | 29.12 | 24.77 |
| TARF | 1.08 | 94.78 | 69.91 | 100.00 | **1.37** |

## G  BROADER IMPACT

In this work, we explore the label domain mismatch in class-wise unlearning, which aims to enhance the flexibility of data regulation with the increasing concern about the trustworthiness of machine

Table 28: Main Results (%). Comparison with the unlearning baselines. All methods are trained on the same backbone, i.e., the basis of unlearning initialization is the same (except for retraining from scratch). Values are percentages. Bold numbers are superior results. ↓ indicates smaller are better.

| CIFAR-10 | Metric | UA | | RA | | TA | | MIA | | Gap↓ | |
|---|---|---|---|---|---|---|---|---|---|---|---|
| | Method | mean | std | mean | std | mean | std | mean | std | mean | std |
| | Retrained | 87.76 | - | 99.58 | - | 95.91 | - | 20.57 | - | - | - |
| | FT (Warnecke et al., 2023) | 94.78 | 0.11 | 98.65 | 0.12 | 93.77 | 0.21 | 10.42 | 0.86 | 5.06 | 0.27 |
| | RL (Toneva et al., 2018) | 48.25 | 5.43 | 98.01 | 0.12 | 93.03 | 0.21 | 98.10 | 0.64 | 30.37 | 1.53 |
| | GA (Ishida et al., 2020) | 6.49 | 0.73 | 86.91 | 0.08 | 82.03 | 0.18 | 94.39 | 0.59 | 45.41 | 0.27 |
| Model | IU (Izzo et al., 2021) | 15.84 | 7.86 | 85.89 | 1.45 | 81.08 | 1.49 | 93.58 | 3.71 | 43.36 | 3.62 |
| mismatch | BS (Chen et al., 2023) | 14.05 | 3.76 | 53.28 | 2.51 | 51.25 | 1.86 | 94.90 | 1.06 | 59.75 | 2.29 |
| | $L_1$-sparse (Jia et al., 2023) | 92.25 | 0.87 | 95.01 | 0.25 | 91.67 | 0.04 | 17.40 | 2.86 | 4.14 | 1.00 |
| | SalUn (Fan et al., 2023) | 16.31 | 7.40 | 92.91 | 1.05 | 86.50 | 2.12 | 99.24 | 0.09 | 41.55 | 2.14 |
| | SCRUB (Kurmanji et al., 2023) | 93.21 | 1.17 | 99.83 | 0.13 | 93.29 | 0.81 | 14.24 | 0.87 | 3.65 | 0.18 |
| | **TARF** (ours) | 89.91 | 1.20 | 97.73 | 0.24 | 92.66 | 0.17 | 20.36 | 2.54 | **2.45** | 0.46 |

Table 29: Main Results (%). Comparison with the unlearning baselines. All methods are trained on the same backbone, i.e., the basis of unlearning initialization is the same (except for retraining from scratch). Values are percentages. Bold numbers are superior results. ↓ indicates smaller are better.

| CIFAR-10 | Metric | UA | | RA | | TA | | MIA | | Gap↓ | |
|---|---|---|---|---|---|---|---|---|---|---|---|
| | Method | mean | std | mean | std | mean | std | mean | std | mean | std |
| | Retrained | 0.00 | - | 99.38 | - | 93.85 | - | 100.00 | - | - | - |
| | FT (Warnecke et al., 2023) | 52.23 | 1.80 | 98.43 | 0.05 | 91.74 | 0.09 | 50.59 | 0.15 | 26.18 | 0.40 |
| | RL (Toneva et al., 2018) | 50.63 | 0.62 | 98.21 | 0.65 | 91.51 | 0.61 | 53.88 | 2.36 | 25.06 | 0.12 |
| | GA (Ishida et al., 2020) | 41.64 | 0.82 | 97.05 | 0.04 | 89.68 | 0.17 | 63.23 | 1.10 | 21.23 | 0.43 |
| Target | IU (Izzo et al., 2021) | 45.32 | 0.81 | 70.25 | 17.82 | 65.67 | 2.76 | 55.98 | 2.76 | 36.66 | 9.37 |
| mismatch | BS (Chen et al., 2023) | 53.78 | 0.16 | 89.67 | 1.02 | 79.34 | 3.95 | 66.31 | 10.02 | 25.36 | 3.28 |
| | $L_1$-sparse (Jia et al., 2023) | 49.55 | 0.08 | 93.57 | 0.05 | 89.06 | 0.23 | 51.33 | 0.09 | 27.21 | 0.05 |
| | SalUn (Fan et al., 2023) | 47.85 | 1.22 | 87.84 | 3.25 | 83.38 | 2.94 | 58.10 | 2.85 | 27.40 | 1.10 |
| | SCRUB (Kurmanji et al., 2023) | 48.53 | 1.02 | 99.43 | 0.21 | 91.66 | 0.28 | 51.27 | 0.73 | 24.92 | 0.51 |
| | **TARF** (ours) | 0.05 | 0.02 | 97.65 | 0.08 | 91.28 | 0.47 | 100.00 | 0.00 | **1.09** | 0.14 |

learning. Pushing forward the practical usage of machine unlearning, our research provides a broader consideration of real-world unlearning scenarios and offers significant positive social impacts. It can enhance data privacy protection by allowing individuals to effectively remove their data, ensuring some sensitive data is not used for analysis. In addition, unlearning can remove bias or discrimination by correcting flawed datasets, promoting the development of fairness or other ethical considerations. This feature also enables enterprises to adhere to data protection standards such as GDPR (Rosen, 2011) and CCPA (Pardau, 2018), therefore promoting confidence among users. Our newly introduced unlearning setting, which decouples the class label and the target concept, is more general and discusses the achievability of various unlearning requests, which may often be different from the taxonomy of pre-training tasks.

Although we take a step forward in more practical class-wise unlearning by considering the label domain mismatch scenarios, it is not the end of this direction and there are still many problems to be addressed. Following the previous works (Warnecke et al., 2023; Golatkar et al., 2020; Jia et al., 2023; Chen et al., 2023), our work mainly focuses on the class-wise unlearning with the classification model for the exploration, future efforts can also be paid in the unlearning problem of the emerging and powerful generative models. On the technical level, although those compared unlearning methods and our framework can achieve the forgetting target, it all requires extra computational cost, and how to make it more efficient can be further studied.

Table 30: Main Results (%). Comparison with the unlearning baselines. All methods are trained on the same backbone, i.e., the basis of unlearning initialization is the same (except for retraining from scratch). Values are percentages. Bold numbers are superior results. ↓ indicates smaller are better.

| CIFAR-10 | Metric | UA | | RA | | TA | | MIA | | Gap↓ | |
|---|---|---|---|---|---|---|---|---|---|---|---|
| | Method | mean | std | mean | std | mean | std | mean | std | mean | std |
| | Retrained | 0.00 | - | 99.53 | - | 95.56 | - | 100.00 | - | - | - |
| | FT (Warnecke et al., 2023) | 96.85 | 0.06 | 98.62 | 0.13 | 93.47 | 0.21 | 6.93 | 0.45 | 48.23 | 0.18 |
| | RL (Toneva et al., 2018) | 73.62 | 2.86 | 97.90 | 0.22 | 92.59 | 0.66 | 52.04 | 2.23 | 31.55 | 1.49 |
| | GA (Ishida et al., 2020) | 9.82 | 1.13 | 96.14 | 0.28 | 90.46 | 0.33 | 90.46 | 0.95 | 6.56 | 0.67 |
| Data | IU (Izzo et al., 2021) | 15.19 | 7.66 | 94.80 | 0.70 | 89.08 | 0.46 | 92.83 | 4.26 | 8.39 | 2.69 |
| mismatch | BS (Chen et al., 2023) | 16.72 | 0.02 | 61.01 | 0.21 | 53.81 | 4.05 | 93.47 | 1.24 | 25.88 | 1.27 |
| | $L_1$-sparse (Jia et al., 2023) | 95.42 | 0.35 | 94.57 | 0.26 | 91.07 | 0.01 | 10.82 | 1.30 | 48.51 | 0.47 |
| | SalUn (Fan et al., 2023) | 55.52 | 3.76 | 92.68 | 1.19 | 89.25 | 1.22 | 60.23 | 3.30 | 27.12 | 2.37 |
| | SCRUB (Kurmanji et al., 2023) | 97.06 | 0.52 | 99.16 | 0.23 | 94.72 | 0.56 | 9.98 | 0.43 | 46.98 | 0.21 |
| | **TARF** (ours) | 0.00 | 0.00 | 98.35 | 0.18 | 93.42 | 0.34 | 100.00 | 0.00 | **0.83** | 0.13 |

Table 31: Main Results (%). Comparison with the unlearning baselines. All methods are trained on the same backbone, i.e., the basis of unlearning initialization is the same (except for retraining from scratch). Values are percentages. Bold numbers are superior results. ↓ indicates smaller are better.

| CIFAR-100 | Metric | UA | | RA | | TA | | MIA | | Gap↓ | |
|---|---|---|---|---|---|---|---|---|---|---|---|
| | Method | mean | std | mean | std | mean | std | mean | std | mean | std |
| | Retrained | 0.00 | - | 97.85 | - | 76.03 | - | 100.00 | - | - | - |
| | FT (Warnecke et al., 2023) | 0.67 | 0.01 | 96.44 | 0.12 | 72.16 | 0.19 | 100.00 | 0.00 | 1.49 | 0.02 |
| | RL (Toneva et al., 2018) | 0.56 | 0.45 | 96.00 | 0.10 | 71.79 | 0.22 | 100.00 | 0.00 | 1.66 | 0.03 |
| | GA (Ishida et al., 2020) | 1.61 | 0.28 | 95.00 | 0.26 | 68.85 | 0.29 | 99.89 | 0.00 | 2.93 | 0.07 |
| All matched | IU (Izzo et al., 2021) | 0.00 | 0.00 | 39.80 | 2.19 | 31.09 | 1.51 | 100.00 | 0.00 | 25.75 | 0.93 |
| | BS (Chen et al., 2023) | 4.83 | 0.05 | 90.17 | 0.06 | 64.30 | 0.64 | 99.45 | 0.12 | 6.20 | 0.22 |
| | $L_1$-sparse (Jia et al., 2023) | 0.00 | 0.00 | 94.25 | 0.57 | 71.35 | 1.27 | 100.00 | 0.00 | 1.92 | 0.46 |
| | SalUn (Fan et al., 2023) | 0.00 | 0.00 | 77.00 | 1.66 | 63.06 | 0.92 | 100.00 | 0.00 | 8.46 | 0.64 |
| | SCRUB (Kurmanji et al., 2023) | 0.00 | 0.00 | 99.72 | 0.26 | 76.69 | 0.06 | 100.00 | 0.00 | **0.64** | 0.08 |
| | **TARF** (ours) | 0.00 | 0.00 | 96.67 | 0.24 | 72.40 | 0.14 | 100.00 | 0.00 | 1.21 | 0.09 |

Table 32: Main Results (%). Comparison with the unlearning baselines. All methods are trained on the same backbone, i.e., the basis of unlearning initialization is the same (except for retraining from scratch). Values are percentages. Bold numbers are superior results. ↓ indicates smaller are better.

| CIFAR-100 | Metric | UA | | RA | | TA | | MIA | | Gap↓ | |
|---|---|---|---|---|---|---|---|---|---|---|---|
| | Method | mean | std | mean | std | mean | std | mean | std | mean | std |
| | Retrained | 88.22 | - | 98.58 | - | 78.50 | - | 25.78 | - | - | - |
| | FT (Warnecke et al., 2023) | 95.45 | 2.78 | 95.91 | 0.89 | 79.74 | 0.40 | 11.56 | 4.78 | 6.34 | 1.77 |
| | RL (Toneva et al., 2018) | 87.11 | 7.00 | 96.27 | 0.44 | 80.00 | 0.17 | 97.95 | 1.06 | 20.75 | 0.61 |
| | GA (Ishida et al., 2020) | 8.06 | 1.28 | 94.98 | 0.15 | 77.09 | 0.13 | 97.34 | 0.45 | 39.18 | 0.50 |
| Model | IU (Izzo et al., 2021) | 39.95 | 5.28 | 97.22 | 0.39 | 79.71 | 0.63 | 83.28 | 3.17 | 27.08 | 2.05 |
| mismatch | BS (Chen et al., 2023) | 18.56 | 0.56 | 95.87 | 0.03 | 74.96 | 2.68 | 94.95 | 0.28 | 36.27 | 0.87 |
| | $L_1$-sparse (Jia et al., 2023) | 91.11 | 5.00 | 94.28 | 0.18 | 77.61 | 0.39 | 15.56 | 4.45 | 5.84 | 1.69 |
| | SalUn (Fan et al., 2023) | 74.78 | 8.45 | 79.98 | 1.14 | 71.55 | 0.77 | 65.61 | 11.39 | 19.71 | 5.44 |
| | SCRUB (Kurmanji et al., 2023) | 92.45 | 2.80 | 99.44 | 0.78 | 78.75 | 1.75 | 20.13 | 4.56 | 4.14 | 1.15 |
| | **TARF** (ours) | 84.78 | 1.90 | 97.19 | 0.14 | 80.02 | 0.15 | 28.89 | 2.89 | **2.37** | 1.15 |

Table 33: Main Results (%). Comparison with the unlearning baselines. All methods are trained on the same backbone, i.e., the basis of unlearning initialization is the same (except for retraining from scratch). Values are percentages. Bold numbers are superior results. ↓ indicates smaller are better.

| CIFAR-100 | Metric | UA | | RA | | TA | | MIA | | Gap↓ | |
|---|---|---|---|---|---|---|---|---|---|---|---|
| | Method | mean | std | mean | std | mean | std | mean | std | mean | std |
| | Retrained | 0.00 | - | 97.85 | - | 73.72 | - | 100.00 | - | - | - |
| | FT (Warnecke et al., 2023) | 58.58 | 0.40 | 96.42 | 0.10 | 72.31 | 0.22 | 45.94 | 0.83 | 28.87 | 0.34 |
| | RL (Toneva et al., 2018) | 57.76 | 1.14 | 96.00 | 0.10 | 72.04 | 0.16 | 50.67 | 3.69 | 27.66 | 1.15 |
| | GA (Ishida et al., 2020) | 22.07 | 0.69 | 96.87 | 0.24 | 70.52 | 0.30 | 90.45 | 0.23 | 8.95 | 0.10 |
| Target | IU (Izzo et al., 2021) | 30.80 | 0.18 | 39.44 | 2.25 | 31.00 | 1.42 | 63.83 | 0.14 | 42.03 | 0.91 |
| mismatch | BS (Chen et al., 2023) | 40.91 | 0.47 | 98.36 | 0.04 | 70.04 | 1.38 | 85.00 | 0.16 | 15.03 | 0.18 |
| | $L_1$-sparse (Jia et al., 2023) | 55.31 | 2.90 | 94.23 | 0.44 | 72.15 | 1.27 | 48.47 | 3.54 | 30.26 | 1.18 |
| | SalUn (Fan et al., 2023) | 43.29 | 1.60 | 77.15 | 1.63 | 63.30 | 0.93 | 64.63 | 1.34 | 27.45 | 0.10 |
| | SCRUB (Kurmanji et al., 2023) | 59.56 | 0.09 | 99.74 | 0.26 | 76.14 | 0.82 | 45.45 | 0.56 | 29.60 | 0.02 |
| | **TARF** (ours) | 0.29 | 0.03 | 97.06 | 0.29 | 73.27 | 0.41 | 100.00 | 0.00 | **0.38** | 0.17 |

Table 34: Main Results (%). Comparison with the unlearning baselines. All methods are trained on the same backbone, i.e., the basis of unlearning initialization is the same (except for retraining from scratch). Values are percentages. Bold numbers are superior results. ↓ indicates smaller are better.

| CIFAR-100 | Metric | UA | | RA | | TA | | MIA | | Gap↓ | |
|---|---|---|---|---|---|---|---|---|---|---|---|
| | Method | mean | std | mean | std | mean | std | mean | std | mean | std |
| | Retrained | 0.00 | - | 98.50 | - | 80.15 | - | 100.00 | - | - | - |
| | FT (Warnecke et al., 2023) | 90.79 | 5.18 | 96.19 | 0.52 | 79.80 | 0.03 | 20.46 | 16.78 | 43.25 | 5.10 |
| | RL (Toneva et al., 2018) | 93.60 | 3.82 | 96.32 | 0.39 | 79.92 | 0.02 | 65.20 | 5.56 | 32.73 | 2.24 |
| | GA (Ishida et al., 2020) | 6.98 | 0.98 | 97.78 | 0.14 | 79.34 | 0.11 | 97.53 | 0.51 | 2.75 | 0.31 |
| Data | IU (Izzo et al., 2021) | 37.22 | 5.71 | 99.17 | 0.21 | 80.01 | 1.81 | 85.41 | 2.41 | 13.54 | 2.08 |
| mismatch | BS (Chen et al., 2023) | 15.71 | 0.33 | 98.47 | 0.04 | 76.02 | 3.74 | 96.05 | 0.18 | 5.86 | 0.18 |
| | $L_1$-sparse (Jia et al., 2023) | 89.02 | 4.67 | 94.18 | 0.05 | 78.89 | 0.20 | 18.67 | 4.36 | 41.64 | 2.20 |
| | SalUn (Fan et al., 2023) | 79.00 | 6.07 | 79.92 | 1.05 | 71.55 | 0.51 | 44.18 | 9.96 | 39.42 | 3.62 |
| | SCRUB (Kurmanji et al., 2023) | 93.28 | 2.10 | 99.25 | 0.98 | 79.18 | 0.48 | 18.45 | 3.55 | 46.13 | 2.37 |
| | **TARF** (ours) | 0.00 | 0.00 | 95.80 | 0.79 | 79.55 | 0.57 | 100.00 | 0.00 | **1.61** | 0.05 |

Table 35: Results (%). Comparison with the unlearning baselines on **varied difficulties on target identification**: to explore the effects of differentiating false retaining data with affected actual retaining data. Specifically, we change the given forgetting data classes to unlearn the target (the superclass "aquatic mammals" including "otter,seal,whale,beaver,dolphin"), it is intuitive that less and biased given forgetting data ("beaver,dolphin") increase the difficulty of representing the whole target concept ("aquatic mammals"), and "lobster" can be mis-identified as potential forgetting data in TARF. With well-represented given forgetting data (left), TARF can perform better; otherwise (right) the benefits upon best baseline decreased.

| Type / $\mathcal{D}$ | (Left) well-represented for the concept (Right) biased and mis-identify "lobster" | (Left) Given "otter,seal,whale" forget "aquatic mammals" | | | | | (Right) Given "beaver,dolphin" forget "aquatic mammals" | | | | |
|---|---|---|---|---|---|---|---|---|---|---|---|
| CIFAR-100 | Method / Metrics | UA | RA | TA | MIA | Gap↓ | UA | RA | TA | MIA | Gap↓ |
| | Retrained (Ref.) | 0.00 | 98.03 | 73.42 | 100.00 | - | 0.00 | 98.03 | 73.42 | 100.00 | - |
| | FT (Warnecke et al., 2023) | 32.98 | 92.98 | 69.78 | 70.98 | 17.67 | 53.42 | 96.58 | 72.26 | 56.58 | 24.86 |
| Target | RL (Toneva et al., 2018) | 38.93 | 96.19 | 72.08 | 64.93 | 19.30 | 57.47 | 95.52 | 71.44 | 47.64 | 28.58 |
| mismatch | GA (Ishida et al., 2020) | 12.76 | 88.59 | 65.27 | 91.91 | 9.61 | 35.07 | 91.81 | 66.39 | 75.91 | 18.10 |
| (varied) | $L_1$-sparse (Jia et al., 2023) | 28.71 | 80.72 | 65.80 | 72.58 | 20.27 | 39.06 | 83.22 | 67.81 | 66.93 | 23.13 |
| | SCRUB (Kurmanji et al., 2023) | 39.46 | 99.22 | 75.81 | 63.24 | 19.95 | 59.11 | 99.46 | 77.14 | 46.09 | 29.54 |
| | **TARF** (ours) | 0.00 | 97.05 | 69.68 | 100.00 | **1.18** | 23.37 | 85.53 | 70.68 | 77.82 | **15.20** |

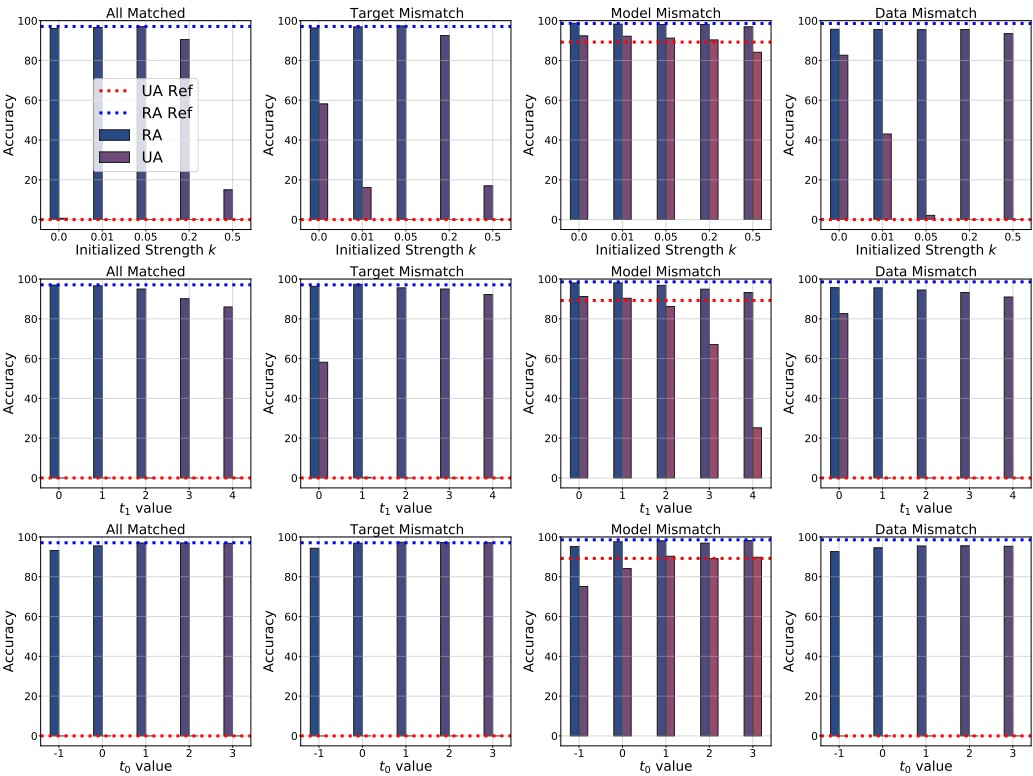

Figure 17: Ablation studies of $k,t_1,t_0$ on four settings (left to right: all matched, target mismatch, model mismatch, and data mismatch) using CIFAR-100 dataset: *top-line:* performance using different initialized $k$ for controlling the forgetting strength, in which $k \geq 0.5$ may induce decreased retaining accuracy; *middle-line:* effect of $t_1$ controlling the length of Phase-I for target identification, generally $t = 1$ is sufficient for differentiate target data like Figure 3 while larger value reduce the retaining epochs; *bottom-line:* effect of $t_0$ controlling the length of Phase-III prevents the excessive forgetting and generally $t_0 = 1$ works well. The above flexibly control the forgetting and retaining part.

