# OpenReview forum: "Decoupling the Class Label and the Target Concept in Machine Unlearning"
_ICLR.cc/2026/Conference — ICLR 2026 Poster_

### Official Review · Reviewer_MLTw · 2025-10-27

**Soundness:** 3
**Presentation:** 2
**Contribution:** 2
**Rating:** 4
**Confidence:** 3

**Summary:**

This paper focuses on a specific case for machine unlearning, where the target concept to be forgotten does not neatly fit into the classes that the model was trained on. Under such a data mismatch, the authors first provide some theoretical analysis of the forgetting. Then, the authors propose a method to handle this scenario, consisting of three steps. Essentially, the forgetting and retaining objectives are used in a dynamically scaled manner.

**Strengths:**

This paper tackles a novel setting of mismatched classes between training time and unlearning time. This may be practical in cases when large models are trained on classification tasks.

Experiments are run on both vision and language settings, which is good.

**Weaknesses:**

A big issue is that the method and theory is not well explained.
For instance, the intuition in Section 3.2 is not well explained, and it is mostly theorems and equations. This is important, because the proposed method relies on these analyses of the forgetting, but the intuition is unclear. For instance, the “gravity effects”, “decomposition lacking”, “representation gravity” terms should be explained intuitively as they seem to be important concepts.

Furthermore, the link between Section 3.2 and 3.3 is not clear. The equations look very different, and how the previous concepts apply here is totally not clear. Thus, the proposed method looks very arbitrary.

There isn’t a related work section that discusses the various related works, and comprehensively covers the similarities and differences to those works.

The words in the figures are of very small size. There is also a lot of text in these figures, and they act like a long caption. Even though I think  adding text in such a manner is technically within the rules, I do not feel comfortable about this.


Many of the compared methods are quite dated, such as  SCRUB, L1-Sparse, SalUn, and  BS which are from 2023. These are already the most recent methods compared. The authors should compare with more recent methods.


Minor typos that do not affect the score:

Line 203: Jacobin -> Jacobian

Line 287: decent -> descent

Line 468: Differernt -> Different

**Questions:**

In Phase 1 Target Identification, how is the dynamic information learnt? How are these information stored?

In Phase 2, how is the dynamic information used to retain the knowledge? I do not observe any weighting or dynamic knowledge directly affecting the loss here.

What does “feature deconstruction” mean? How is this achieved in the method?

I do not understand the results in Fig 5a and 5b. I do not think these experiments and results have been explained clearly anywhere. What is the experiment and what is being visualized here?


Please see more questions in the Weaknesses Section.

---

> ### Author Response · Authors · 2025-11-19
> **Response to Reviewer MLTw [1/5]**
>
> We sincerely thank the reviewer for the time and effort spent reviewing our paper and for acknowledging the novelty of our setting, practicality of the introduced problem, and experimental results. Below, we address the raised questions and concerns in detail:
>
>
> > **W1:** A big issue is that the method and theory is not well explained. For instance, the intuition in Section 3.2 is not well explained, and it is mostly theorems and equations. This is important, because the proposed method relies on these analyses of the forgetting, but the intuition is unclear. For instance, the “gravity effects”, “decomposition lacking”, “representation gravity” terms should be explained intuitively as they seem to be important concepts.
>
> **A1:** We appreciate the reviewer’s constructive comments and would further clarify the conceptual intuition of our theoretical analysis in Section 3.2 and how this forms the underlying mechanism of our method framework.
>
> The intuition of Section 3.2 is mainly to **understand the representation-level relationships through the forgetting dynamics**, which reveal why restrictive unlearning becomes challenging when the label domains are mismatched (as empirically shown in Section 3.1). When we apply a few gradient-ascent steps on the forgetting set $s_1$​, samples in the remaining data that are close in representation space to the forgetting data will react in a similar way (e.g., the loss increase). This is captured by Eq. (2), where the leading term shows that the magnitude of loss change can be proportional to their representation distance. **Intuitively, if two portions of data occupy nearby/far-apart regions in the latent space, pushing the model to forget the one​ will inadvertently/loosely affect the other​.** This co-movement reflects what we termed **representation gravity**, the idea that representation similarity determines how strongly samples move together under forgetting dynamics.
>
> This theoretical analysis naturally gives rise to the terms explaining what challenges mismatched unlearning (“insufficient representation” and “decomposition lacking”) and the core mechanism (“gravity effects”) also related to our method. **Insufficient representation** arises when the forgetting set covers only part of the true target concept; the remaining target samples lie farther away and display only weak following dynamics. **Decomposition lacking** reflects the opposite case, where several subclasses are overly entangled in the learned representation, leading to forgetting updates applied to one subclass to spill over onto others. Those are **empirically demonstrated by our Figure 3** (left shows strong gravity effects among tightly pre-clustered samples, right illustrates weak responses for distant samples under relatively weak semantic similarity). Finally, **gravity effects** themselves, observed when semantically related samples exhibit similar dynamics, provide the essential signal for identifying false-retaining data in target or data mismatch scenarios and also indicate stronger disentanglement is required for successful forgetting in model mismatch scenarios, given the target concept need to be isolated from those affected retaining data (refer to the left of Figure 2).
>
> **This intuition directly motivates the design of TARF in Section 3.3**, where early-phase forgetting is used to identify latent target samples, and later-phase joint or pure retaining disentangles the representation and avoid unintended side effects. We will revise the section to add intuitive explanations more explicit in each remark and better connected to the motivating figures.

---

> > ### Author Response · Authors · 2025-11-19
> > **Response to Reviewer MLTw [2/5]**
> >
> > > **W2:** Furthermore, the link between Section 3.2 and 3.3 is not clear. The equations look very different, and how the previous concepts apply here is totally not clear. Thus, the proposed method looks very arbitrary.
> >
> > **A2:** We thank the reviewer for the comments and would like to clarify that **Section 3.3 is directly motivated by the theoretical insights established in Section 3.2**, while the equations differ due to the different purposes of two sections: Section 3.2 provides an analytical characterization of forgetting dynamics, revealing the behaviors we term representation gravity, insufficient representation, and decomposition lacking. Eq.(2) directly underpins the need for mechanisms that identifies and distinguishes these latent relationships. The theoretical insights in Section 3.2 therefore determine what information is meaningful and actionable, and they highlight gravity effects as the core signal for addressing mismatched unlearning **as clarified in A1**.
> >
> > **Section 3.3 operationalizes exactly these theoretical findings into the general TARF framework**, capable of handling those mismatched unlearning scenarios with target identification and separation. It naturally leads to two parts of objective design, e.g., annealed forgetting and target-aware retaining to organically tackle with the aforementioned challenge of Section 3.2 (which can be understood as 3 Phases with different functionality under a unified structure). The early-stage forgetting dynamic information is used to identify false-retaining data for insufficient representation, and jointly application of forgetting with retaining then addresses the decomposition lacking, supplemented with retaining approximation to prevent over-forgetting phenomenon. Thus, none of the design choices in Section 3.3 are arbitrary and each corresponds directly to a specific representation behavior analyzed earlier. **We will further elaborate the transition paragraph at the later of Section 3.2 and beginning of 3.3** to make this relationship more explicit and easier to follow.
> >
> > > **W3:** There isn’t a related work section that discusses the various related works, and comprehensively covers the similarities and differences to those works.
> >
> > **A3:** Thanks for raising the concern regarding the related work discussion, while we respectfully note **We have provided a related work discussion in Appendix B**, and **also mainly discussed the major differences of our work with the literature** in the end of Section 2 due to the space constraints of the presentation.  The appendix provides a further comparison across related unlearning paradigms and clarifies how our setting fundamentally differs from them, e.g., considering the comprehensive label domain mismatch, and we also include more recent advances on machine unlearning for discussion. **To improve accessibility**, in the revised version we will explicitly highlight cross-references in the main text to guide readers to Appendix B. Furthermore, if space permits in the final version, we will also consider moving a condensed version of this discussion into the main body to more clearly articulate the connections and distinctions between our work and prior literature. We thank the reviewer again for this valuable suggestion.
> >
> >
> > > **W4:** The words in the figures are of very small size. There is also a lot of text in these figures, and they act like a long caption. Even though I think adding text in such a manner is technically within the rules, I do not feel comfortable about this.
> >
> > **A4:** We appreciate the reviewer’s feedback regarding figure content readability. The dense formatting in the current submission largely stems from our newly introduced settings and the need to present comprehensive verification within strict page limits. This led to more information being embedded than we would prefer. Nonetheless, we agree that some text appears small or overly condensed. In the final version, if additional space is available, we will enlarge all figure text and move most explanatory details into the captions to ensure clearer and more comfortable readability. We thank the reviewer again for this valuable and constructive suggestion.

---

> ### Author Response · Authors · 2025-11-19
> **Response to Reviewer MLTw [3/5]**
>
> > **W5:** Many of the compared methods are quite dated, such as SCRUB, L1-Sparse, SalUn, and BS which are from 2023. These are already the most recent methods compared. The authors should compare with more recent methods.
>
> **A5:** Thanks for the reviewer’s suggestion regarding baseline recency and would like to clarify that our **current comparisons already include the representative and widely-adopted baselines used in the recent literature** on class-wise unlearning literature. **Methods** (from 2023 to late 2024) such as SCRUB, SalUn, L1-Sparse, and BS **remain the standard and most frequently adopted baselines in recent publications**. They continue to be the primary points of method consideration and comparison in recent works [1,2,3,4] due to their general applicability and public availability. While several newer studies focus on foundation-model-oriented unlearning, they often rely on architectural or modality-specific mechanisms, which are not directly applicable to a general algorithmic-level comparison.
>
> **To further address the concern** and strengthen the comprehensiveness of our study, **we have additionally incorporated three additional unlearning methods** into our main comparison: LAU [3], SFR-on [1], and SG [4], in the following Table 1. These methods propose some advancements for class-wise unlearning in different aspects while not considering the mismatched challenges. Both have been evaluated under the same mismatched unlearning scenarios introduced in our paper, using identical training budgets and evaluation protocols to ensure a fair comparison. **The additional results also validate the effectiveness and generality of our TARF** framework across all tasks, as the unified framework design enables target identification and separation. TARF maintains robust unlearning performance, due to its flexible capabilities of handling various mismatched unlearning scenarios. We thank the reviewer again for the suggestion.
>
> **Table 1. Unlearning comparison with more recent methods.**
> | All Matched    | UA    | RA    | TA    | MIA    | Gap   | Target Mismatch | UA    | RA    | TA    | MIA    | Gap   |
> |----------------|-------|-------|-------|--------|-------|-----------------|-------|-------|-------|--------|-------|
> | Retrained      | 0.00  | 97.85 | 76.03 | 100.00 | -     || 0.00  | 97.85 | 73.72 | 100.00 | -     |
> | FT | 0.67  | 96.32 | 72.34 | 100.00 | 1.47  || 58.18 | 96.32 | 72.53 | 46.76  | 28.54 |
> | GA  | 1.33  | 94.74 | 68.56 | 99.89  | 3.01  || 21.38 | 96.64 | 70.22 | 90.67  | 8.86  |
> | BS   | 4.60  | 90.18 | 63.66 | 99.55  | 6.27  |       | 40.44 | 98.32 | 68.66 | 85.16  | 15.20 |
> | &L_1&-sparse   | 0.00  | 94.60 | 71.57 | 100.00 | 1.93  |         | 56.09 | 94.63 | 72.00 | 48.04  | 28.25 |
> | SalUn | 0.00  | 75.34 | 62.14 | 100.00 | 9.10  |       | 59.64 | 75.52 | 62.37 | 65.96  | 27.35 |
> | LAU  | 4.11  | 80.44 | 61.64 | 95.78  | 10.03 |       | 46.71 | 88.65 | 68.19 | 66.49  | 23.74 |
> | SFR-on   | 0.00 | 99.21  | 73.16  | 100.00 | 1.08  |      | 59.21| 99.13 | 74.28 | 48.32 | 28.18 |
> | SG   | 0.00 | 96.21  | 72.23  | 100.00 | 1.36  |      | 58.21| 96.26 | 72.18 | 46.24 | 28.78 |
> | SCRUB  | 0.00  | 99.98 | 76.75 | 100.00 | **0.71**  |    | 59.64 | 99.99 | 75.32 | 44.89  | 29.90 |
> | TARF   | 0.00  | 96.90 | 72.53 | 100.00 | 1.11  |       | 0.31  | 97.35 | 73.68 | 100.00 | **0.21**  |
> | **Model Mismatch** | **UA**    | **RA**    | **TA**    | **MIA**    | **Gap**   | **Data Mismatch**   | **UA**    | **RA**    | **TA**    | **MIA**    | **Gap**   |
> | Retrained      | 88.22  | 98.58 | 78.50 | 25.78 | -     || 0.00  | 98.50 | 80.15 | 100.00 | -     |
> | FT             | 92.67 | 95.02 | 79.34 | 16.33  | 4.58  || 82.62 | 95.66 | 79.77 | 37.24  | 37.15 |
> | GA             | 6.78  | 94.83 | 76.96 | 97.78  | 39.68 || 6.00  | 97.65 | 79.23 | 98.04  | 2.43  |
> | BS             | 18.11 | 95.90 | 72.28 | 95.22  | 37.14 || 15.38 | 98.50 | 72.28 | 96.22  | 6.76  |
> | &L_1&-sparse   | 90.22 | 94.78 | 78.81 | 18.88  | 3.25  || 88.31 | 94.91 | 79.02 | 22.49  | 42.64 |
> | SalUn          | 66.33 | 78.83 | 70.78 | 77.00  | 25.15 || 72.93 | 78.87 | 71.04 | 54.13  | 36.89 |
> | LAU            | 80.00 | 96.74 | 79.86 | 45.78  | 7.86  || 85.73 | 96.96 | 80.00 | 40.40  | 36.76 |
> | SFR-on         |  92.12  | 99.21 | 79.21 | 20.65 | 2.59 || 92.68 | 99.21 | 79.23| 18.21 | 44.03 |
> | SG   | 89.27 | 93.52  | 73.45  | 19.31 | 4.41  || 87.52| 93.25 | 73.21 | 23.08 | 44.16 |
> | SCRUB          | 91.44 | 99.74 | 79.23 | 21.11  | 2.45  || 95.50 | 99.79 | 79.68 | 15.11  | 45.54 |
> | TARF           | 86.67 | 97.05 | 80.07 | 26.00  | **1.21**  || 0.00  | 95.01 | 78.98 | 100.00 | **1.17**  |
>
> [1] Unified Gradient-Based Machine Unlearning with Remain Geometry Enhancement. NeurIPS 2024.
> [2] Reminiscence Attack on Residuals: Exploiting Approximate Machine Unlearning for Privacy. ICCV 2025.
> [3] Label-agnostic forgetting: A supervision-free unlearning in deep models. ICLR 2024.
> [4] Adversarial Machine Unlearning. ICLR 2025.

---

> > ### Author Response · Authors · 2025-11-19
> > **Response to Reviewer MLTw [4/5]**
> >
> > > **M1:** Minor typos that do not affect the score: Line 203: Jacobin -> Jacobian; Line 287: decent -> descent; Line 468: Differernt -> Different
> >
> > **A6:** We thank the reviewer for carefully pointing out these typographical issues. We have **corrected all of them in the revised version**, including fixing “Jacobin → Jacobian” (Line 203), “decent → descent” (Line 287), and “Differernt → Different” (Line 468). We appreciate the reviewer’s comments in improving the professionalism of the manuscript and **will thoroughly recheck the draft** to ensure the overall quality of our presentation.
> >
> > > **Q1:** In Phase 1 Target Identification, how is the dynamic information learnt? How are these information stored?
> >
> > **A7:** **The dynamic information refers specifically to the change in loss or accuracy values** after early forgetting iterations. This is computed once at the end of Phase 1 by evaluating the difference between the initial model and the updated model. TARF does not store full update trajectories. Instead, **it keeps only a lightweight ranking result indicating which samples/classes exhibit the larger loss/accuracy changes**, with minimal overhead as it relies on the original forward pass of unlearning. This ranking forms the basis of target identification to select those false-retaining data with closer representation distance with the given forgetting data.
> >
> > > **Q2:** In Phase 2, how is the dynamic information used to retain the knowledge? I do not observe any weighting or dynamic knowledge directly affecting the loss here.
> >
> > **A8:** **The dynamic information obtained in Phase-I is encoded through our introduction gate function $\tau$ (detailed in Eq.(5)), which determines whether each sample in the remaining set participates in the retaining part, as illustrated in Eq.(3)**. In target-aware retaining, samples in the remaining set identified being aligned with the forgetting set are excluded (setting the weight to 0) from gradient descent after Phase I, while the rest (setting the weight to 1) receive the standard gradient descent updates. In this way, **it influences the optimization path through selective participation of the remaining data.**
> >
> >
> > > **Q3:** What does “feature deconstruction” mean? How is this achieved in the method?
> >
> > **A9:** The term “feature deconstruction” refers to **intentionally weakening the latent entanglement between the target and non-target concepts**, which stem from the representative-level understanding of how forgetting unfolds. **Intuitively, as illustrated in the left of Figure 2**, under model mismatch, the forget and a portion of retain data **occupy overlapping regions in the latent space**, so forgetting one unintentionally disturbs the other, which is denoted as affected retaining data. By jointly applying gradient ascent to the target data and gradient descent to the non-target data, TARF gradually reshapes the representation, **leading to disentanglement and mitigation of such unintended effects**. We will revise the manuscript to supplement the term with further clarification using “disentanglement”, to reduce the potential ambiguity.

---

> > > ### Author Response · Authors · 2025-11-19
> > > **Response to Reviewer MLTw [5/5]**
> > >
> > > > **Q4:** I do not understand the results in Fig 5a and 5b. I do not think these experiments and results have been explained clearly anywhere. What is the experiment and what is being visualized here?
> > >
> > > **A10:** We apologize for the lack of detailed explanation. We will adjust them accordingly and supplement with the following explanation:
> > >
> > > - **In Figure 5(a), the two subplots are selected to show behavior under target mismatch (left) and data mismatch (right).** The former demonstrates the accuracy-drop pattern after Phase 1. It shows that classes belonging to the target concept (the blue ones) experience a significantly larger accuracy drop than the remaining classes (the yellow ones), which serves as a representative indicator for target identification. At the same time, the latter presents the robust performance of TARF using different amounts of given forgetting classes. The two subplots present preliminary verification on the effectiveness and robustness of our target identification relying on forgetting dynamics.
> > > - **In Figure 5(b), the two subplots are presented for the model mismatch task**. In the first panel, we compare the accuracy gap on RA and UA, which indicates the success (refer the dashed line of Retrained reference) of disentanglement. It validates the rationality of our method, which jointly applies gradient ascent and descent to deconstruct the entangled representation, achieving the expected accuracy gap (e.g., isolating the target concept with affected retaining data as shown in Figure 2). In contrast, FT and GA result in limited deconstruction due to the partial objective pursuit. Note that in the middle stage, our TARF can induce over-deconstruction (larger Acc Gap than that of the dashed line for Retrained reference), so the right panel also demonstrates the necessity of our Phase-III focusing purely on retraining to approximate the Retrained reference by using different epochs of this stage.
> > >
> > > We will expand both the caption and the corresponding discussion in Section 4 to ensure the purpose of each visualization is unambiguous and clearly explained.
> > >
> > >
> > > ---
> > > We appreciate the reviewer’s comments again to improve our presentation, and **hope our responses have addressed your concerns. Please don’t hesitate to let us know if there are any remaining points** we can clarify or strengthen, and we always welcome further discussion.

---

> > > > ### Author Response · Authors · 2025-11-26
> > > >
> > > > Dear Reviewer,
> > > >
> > > > We sincerely appreciate your comments and the time you have dedicated to reviewing our paper.
> > > >
> > > > We have submitted our rebuttal and hope our responses have clarified the points you raised. We are writing to gently follow up and see if you have had a chance to look at our responses. We remain committed to improving our work based on your valuable suggestions during the discussion, and would like to ask if there is any unclear point so that we should/could further clarify.
> > > >
> > > > We look forward to your feedback.
> > > >
> > > > Best,
> > > >
> > > > Authors of Submission #9776

---

> > > > > ### Author Response · Authors · 2025-11-27
> > > > > **Looking forward to your reply**
> > > > >
> > > > > Dear Reviewer MLTw,
> > > > >
> > > > > Thanks again for your time in reviewing and your comments! We have provided detailed responses and conducted the revision (summarized in General Response) in our updated manuscript.
> > > > >
> > > > > Would you mind confirming if there is any unclear point so that we should/could further clarify? We will do our best to address them. Additionally, if you find our responses helpful, we would sincerely appreciate it if you could acknowledge the efforts we have made.
> > > > >
> > > > > Best regards,
> > > > >
> > > > > Authors of Submission #9776

---

> > > > > > ### Author Response · Authors · 2025-11-27
> > > > > > **Would you mind confirming if there is any unclear point so that we should/could further clarify?**
> > > > > >
> > > > > > Dear Reviewer MLTw,
> > > > > >
> > > > > > We sincerely appreciate your time and effort in reviewing our paper! As the discussion period is drawing to a close, we would like to kindly inquire if there are any remaining unclear parts that we can clarify, given the additional clarification, experiments, and analysis we carefully added following your constructive feedback.
> > > > > >
> > > > > > We remain committed to improving our work based on your suggestions and are happy to provide any further elaboration as needed.
> > > > > >
> > > > > > Sincerely,
> > > > > >
> > > > > > Authors of Submission #9776

---

### Official Review · Reviewer_ES8Q · 2025-10-28

**Soundness:** 3
**Presentation:** 2
**Contribution:** 3
**Rating:** 6
**Confidence:** 3

**Summary:**

This manuscript addresses the machine unlearning problem and proposes a target-aware dynamic forgetting framework (TARF). The framework innovatively formulates the forgetting task as a complex optimization scenario involving a mismatch among the “forgetting data”, “model outputs”, and “target concepts”. It leverages “representation gravity” effect to dynamically identify forgetting targets that are not fully labeled, and employs a collaborative optimization mechanism combining “annealed gradient ascent” with a “target-aware retention” strategy to balance the completeness of forgetting with the preservation of the model’s general performance. Experimental results convincingly demonstrate the effectiveness and superiority of this approach across various mismatch scenarios. Nonetheless, there remain several aspects of the paper that could be improved.

**Strengths:**

1. The manuscript introduces new settings that decouple the class label and the target concept, which investigate the label domain mismatch in class-wise unlearning.
2. The manuscript systematically reveals the challenges of restrictive unlearning with the mismatched label domains, and demonstrates that the "representation gravity" in forgetting dynamics is critical for achieving the forgetting target in the new tasks.

**Weaknesses:**

1. Some of the figure legends in the manuscript obscure content. For example, Fig 5 and Fig 7. Additionally, certain legends are not fully explained, which affects readability. For example, the line legend in Fig 5(b). The readability of the tables is poor. It is recommended to also highlight the second-best values to improve readability in Table 2.
2. Compared with some simpler forgetting methods, the Target Identification stage in TARF requires additional computation to observe the “gravity effect.” The authors do not provide a clearer analysis and comparison of the extra computational overhead introduced by the TARF framework in the main text.

**Questions:**

1. In Section 3.3, the manuscript employs two time functions to control the loss function at each training round. Since the complexity of different datasets can influence the loss dynamics, how are the hyperparameters involved in the time functions(such as β and annealing rate)? How are the temporal checkpoints t_0andt_1  determined among Target Mismatch, Target Mismatch, and Retraining Approximation? Although the ablation studies in the appendix demonstrate the robustness of the method within certain ranges, it is recommended that the authors provide more practical guidance or heuristics on how to select these parameters for new tasks.

2. The manuscript presents four different types of mismatch scenarios. In Phase 1, the method performs Target Identification to locate samples among the remaining data that are similar to the forgetting data. However, in my opinion, this approach may not be applicable when the target scope is broader than that of the forgetting samples (Target Mismatch scenario described in the paper). It is suggested to further clarify how the method addresses this issue.

3. The manuscript primarily focuses on scenarios involving the forgetting of a single or closely related target concept. It would be valuable for the authors to discuss the scalability of the proposed framework when multiple target concepts need to be forgotten simultaneously, particularly when these concepts are entirely unrelated or partially overlapping.

4. How much additional computing overhead in target recognition introduced in phase 1? Will this overhead increase significantly with the number of categories and the proportion of false data?

---

> ### Author Response · Authors · 2025-11-19
> **Response to Reviewer ES8Q [1/3]**
>
> We sincerely thank the reviewer for the time and effort spent reviewing our paper and for acknowledging the new introduced settings, systematically revealed challenges, and exploration of representation gravity. Below, we address the raised questions and concerns in detail:
>
> > **W1:** Some of the figure legends in the manuscript obscure content. For example, Fig 5 and Fig 7. Additionally, certain legends are not fully explained, which affects readability. For example, the line legend in Fig 5(b). The readability of the tables is poor. It is recommended to also highlight the second-best values to improve readability in Table 2.
>
> **A1:** We appreciate the reviewer’s identification of these readability issues and the constructive suggestions. **We have revised the figures and table accordingly** to ensure clearer presentation and reduce visual ambiguity. **For Fig. 5 and Fig. 7**, the original legends partially overlapped with plotted bars, which caused visual occlusion: **we have repositioned all legends** to unobtrusive areas of each panel and adjusted spacing to ensure that no content is obscured. **For Fig. 5(b)**, the legend for the accuracy-gap curves (RA − UA) corresponds to four methods: **we have now added explicit clarification** in the in-figure texts, i.e., Retrained: retrained reference, GA: gradient ascent, FT: finetuning, TARF: our methods. **Regarding Table 2**, the initial submission highlighted only the best values: **following the reviewer’s suggestion, we now additionally highlight the second-best values using underlining** to improve readability in the large table. We thank the reviewer again for these helpful comments, the results will be updated in our revised version.
>
>
> > **W2:** Compared with some simpler forgetting methods, the Target Identification stage in TARF requires additional computation to observe the “gravity effect.” The authors do not provide a clearer analysis and comparison of the extra computational overhead introduced by the TARF framework in the main text. **Q4:** How much additional computing overhead in target recognition introduced in phase 1? Will this overhead increase significantly with the number of categories and the proportion of false data?
>
>
> **A2:** We appreciate the reviewer for the insightful comments and questions. We would like to clarify that **TARF introduces limited computation overhead**, both algorithmically and empirically.  In algorithm realization, we should note that the **target identification (e.g., Phase-I) relies on the gradient ascent, which is inherently aligned with the forgetting objective** instead of a separate operation for computing gravity effects solely. The required information (e.g., loss or accuracy changes) is obtained using the same forward pass that is already necessary for unlearning. Since every method needs to compute the model’s forward outputs on the entire dataset to monitor or update the loss. TARF simply reuses these outputs to compute the change relative to the initial model, which requires only a lightweight subtraction and ranking operation (as described in lines 272) for the remaining classes in our setting. **No additional gradient operations, optimization steps, or repeated evaluations are introduced**.
>
> **Table 1. Computational overhead of target identification compared with unlearning procedure.**
> | Dataset | TIME-Ra | TIME-In | TIME-Un | UA   | RA    | TA    | MIA    | Gap  |
> |---------|-----|----|----|------|-------|-------|--------|------|
> | CIFAR-10 | 0.0000017 | 0.18 | 4.23 | 0.06 | 97.57 | 90.81| 100.00 | 1.23 |
> | CIFAR-100| 0.0000017 | 0.18 | 4.85 | 0.31 | 97.35 | 73.68| 100.00 | 0.21 |
> | Tiny-ImageNet | 0.0000018 | 0.95 | 32.81| 1.08 | 94.78 |69.91 | 100.00 | 1.37 |
> | ImageNet| 0.0000018 | 11.52| 628.87  | 0.00 | 69.93 | 71.79 | 100.00 | 3.97 |
>
> Empirically, Table 1 reports the computation time (mins) of identification (TIME-In) including the forwarding pass and ranking operation, compared with that (TIME-Un) of the whole unlearning process. The resulting overhead is relatively limited compared with the whole unlearning cost across the datasets, especially the ranking time (TIME-Ra) is even negligible than the forwarding pass. **Regarding scalability**, this overhead does not increase with the proportion of false-retain data, as the difference computation is encoded in the same forwarding pass of unlearning on the entire dataset. While we should acknowledged that the complexity of ranking procedure has O(Nlog⁡N) cost over the categories/data size N, **which remains minimal compared to the cost of forward/backward propagation (as previously verified)**. While extremely large-scale settings could further benefit from optional structural priors or sampling-based accelerations, our current implementation can maintain favorable scalability. We will clarify these points and include the above quantitative comparison in our revised version.

---

> > ### Author Response · Authors · 2025-11-19
> > **Response to Reviewer ES8Q [2/3]**
> >
> > > **Q1:** In Section 3.3, the manuscript employs two time functions to control the loss function at each training round. Since the complexity of different datasets can influence the loss dynamics, how are the hyperparameters involved in the time functions(such as $\beta$ and annealing rate)? How are the temporal checkpoints $t_0$ and $t_1$ determined among Target Mismatch, Target Mismatch, and Retraining Approximation? Although the ablation studies in the appendix demonstrate the robustness of the method within certain ranges, it is recommended that the authors provide more practical guidance or heuristics on how to select these parameters for new tasks.
> >
> > **A3:** We appreciate the reviewer’s question and the suggestion to provide more practical guidance. The hyperparameters introduced in TARF are **structurally constrained by the roles they play in the objective in Eq. (3)**, which **align with Theorem 3.2 and Figure 3 indicating that forgetting dynamics** are informative in the early stage for identifying target samples, whereas later updates risk inducing unexpected feature distortion with retaining part. This theoretical insight governs how the time functions and their associated hyperparameters should be chosen. In particular, **$\beta$ is determined automatically by the quantile-based dynamic-response ranking** (as described near line 272). The forgetting scope is assumed to be known as prior information, similar to thresholding approaches in other machine learning literature (e.g., learning with label noise). The remaining hyperparameters are therefore the ones that require structural tuning.
> >
> > Building on above intuition, we can **summarize the practical guidelines** as follows: **First, $k$** should be initialized to a small value and increased cautiously; a modest early ascent is sufficient to reveal representation gravity, while overly large values cause unnecessary distortion. **Second, $t_1$​** can generally be set very early (typically Epoch 1), since the gravity-based ranking relies on the first few steps of forgetting. **Third, $t_0$​** can be tuned with validation set by extending Phase III when additional retaining approximation is needed, ensuring that any feature distortion induced by early forgetting is corrected. These **principles are consistently supported by our ablations:** Figures 7 and 17 show that varying the hyperparameters across reasonable ranges yields robust behaviors except for extreme values that contradict the above intuition. We will add the guidelines in our revised version and add the cross-reference in main text for explicit mention.
> >
> > > **Q2:** The manuscript presents four different types of mismatch scenarios. In Phase 1, the method performs Target Identification to locate samples among the remaining data that are similar to the forgetting data. However, in my opinion, this approach may not be applicable when the target scope is broader than that of the forgetting samples (Target Mismatch scenario described in the paper). It is suggested to further clarify how the method addresses this issue.
> >
> > **A4:**  We thank the reviewer for the thoughtful comments. We would like to clarify that **the core mechanism of Target Identification relies on representation gravity, which does not depend on the exact size of the provided forgetting set but instead on the representation distance between samples**. In our introduced scenario, although the given forgetting samples do not span the full target concept as that in data mismatch scenario, the forgetting data and false-retaining data still share the similar semantic information which can be considered as an interesting group of forgetting. **As shown in the right of Figure 3**, those false-retaining samples (blue ones) exhibit distinct and larger dynamic changes. Although the gravity effect is weaker than what left panels show, it allows our TARF to identify the potential forgetting samples for target-aware retaining, the dynamic information therefore remains reliable even when the target scope is broader and the experimental results across multiple datasets also verified it.

---

> > > ### Author Response · Authors · 2025-11-19
> > > **Response to Reviewer ES8Q [3/3]**
> > >
> > > Although effective, **we should also agree with the reviewer and acknowledged one extremely challenging scenario** that the target scope share weak or limited semantic similarity with the given forgetting data (e.g., unlearning “flowers” when the provided forgetting samples correspond to “ants”), which directly weaken the signal of representation gravity, and can induce more noise in the identified false-retaining data through ranking accuracy-drop. Such a challenging scenario becomes intrinsically difficult but is not well-defined for feasibility. **This limitation is not specific to TARF but reflects a fundamental ambiguity in representation-level signals** when the given forgetting samples provide insufficient semantic cues about the target concept. In these situations, additional information or external priors would be necessary to reliably infer the full target scope. We will make this limitation explicit in the revised manuscript and discuss it as a direction for future research, especially for settings where representation-level relationships are highly ambiguous.
> > >
> > > > **Q3:** The manuscript primarily focuses on scenarios involving the forgetting of a single or closely related target concept. It would be valuable for the authors to discuss the scalability of the proposed framework when multiple target concepts need to be forgotten simultaneously, particularly when these concepts are entirely unrelated or partially overlapping.
> > >
> > >
> > > **A5:** Thanks for the insightful question. We would like to clarify that **the TARF framework is scalable to forgetting multiple concepts or superclasses** simultaneously, as we didn't restrict concept numbers in our algorithm design. **Conceptually, the core challenges** of mismatched unlearning still lie on **insufficient representation** or **decomposition lacking** as revealed in our Section 3.2. For the model mismatch setting, it doesn't introduce extra algorithmic difficulty with multiple target concepts, as our objective simultaneously considers gradient ascent and descent to deconstruct the entangled representation. For the target/data mismatch setting, we can also identify those multiple target concepts by respectively utilizing our Phase-I with the base model, given the forgetting data as a representative support set for each concept. Otherwise, if we have no representative forgetting data, the task becomes infeasible for almost all unlearning methods, as discussed in our A4.
> > >
> > > In our presentation, **we have presented discussion and verification on forgetting multiple classes simultaneously in Appendix G.9.** In Table 25, we show **forgetting multiple target concepts** on Tiny-ImageNet or ImageNet to demonstrate the effectiveness. For **multiple-superclass forgetting**, we also present the results in Table 37, where we conduct unlearning in the target mismatch setting using the CIFAR-100 dataset for two target concepts forgetting, e.g., "people" and "aquatic mammals". As a result, TARF demonstrates consistent performance effectiveness across all scenarios without harming the retention on retaining data, **indicating its scalability to multi-concept forgetting.** We will enhance the cross-reference of this part of discussion in the main text.
> > >
> > > ---
> > > We appreciate the reviewer’s comments again to improve our presentation, and **hope our responses have addressed your concerns. Please don’t hesitate to let us know if there are any remaining points** we can clarify or strengthen, and we always welcome further discussion.

---

> > > > ### Author Response · Authors · 2025-11-27
> > > > **Look forward to your reply**
> > > >
> > > > Dear Reviewer ES8Q,
> > > >
> > > > Thanks again for your time in reviewing and your comments! We have provided detailed responses and conducted the revision (summarized in General Response) in our updated manuscript.
> > > >
> > > > Would you mind confirming if there is any unclear point so that we should/could further clarify? We will do our best to address them. Additionally, if you find our responses helpful, we would sincerely appreciate it if you could acknowledge the efforts we have made.
> > > >
> > > > Best regards,
> > > >
> > > > Authors of Submission #9776

---

### Official Review · Reviewer_gpUj · 2025-10-30

**Soundness:** 3
**Presentation:** 3
**Contribution:** 3
**Rating:** 8
**Confidence:** 3

**Summary:**

This paper addresses a critical and often overlooked limitation in machine unlearning: the implicit assumption that the "target concept" to be forgotten perfectly aligns with a pre-defined "class label". The authors argue that in practical scenarios, a mismatch frequently occurs between the labels of the data slated for forgetting, the model's output taxonomy, and the actual target concept. To address this, the paper formalizes this "label domain mismatch" by introducing three novel and practical unlearning scenarios: target mismatch, model mismatch, and data mismatch. These scenarios significantly expand the scope of unlearning research beyond the conventional "all matched" setting. The authors systematically analyze the challenges arising from these scenarios, revealing crucial representation-level forgetting dynamics, which they term the "gravity effect". Based on these insights, they propose a general framework, TARF (TARget-aware Forgetting), designed to actively forget the true target concept while preserving the utility of the remaining data. TARF achieves this through a unified objective that simultaneously performs annealed gradient ascent on the identified forgetting data and a selective, target-aware gradient descent on a subset of the remaining data that is at risk of being unintentionally affected. Extensive experiments on benchmarks (CIFAR-10/100, ImageNet) and modern applications (LLMs, diffusion models) demonstrate that TARF significantly outperforms existing methods in these challenging new scenarios, showcasing its effectiveness and general applicability.

**Strengths:**

1.Pioneering Problem Formulation: The paper introduces a novel, highly practical, and significant problem by decoupling the class label from the target concept in machine unlearning. This new perspective, formalized into three distinct mismatch scenarios, fundamentally expands the scope of the field and bridges a critical gap between academic research and real-world requirements for privacy, copyright, and AI safety.
2.Principled and General Framework (TARF): The proposed TARF framework is an elegant and well-motivated solution. Rather than being an ad-hoc fix, its design is grounded in a solid theoretical and empirical analysis of representation-level forgetting dynamics. Its ability to handle all four scenarios (matched and mismatched) within a single, unified objective highlights its robust and general design.
3.Comprehensive and Rigorous Empirical Validation: The paper provides extensive and convincing experimental results. TARF's superiority is demonstrated across multiple datasets of varying scales, against a wide array of strong baselines, and in all proposed scenarios. The inclusion of case studies on modern generative models (Stable Diffusion and LLMs) powerfully underscores the method's relevance and practical utility in contemporary AI systems.

**Weaknesses:**

1.While the appendix contains a detailed ablation study for key hyperparameters (Figure 17), the discussion is currently presented as an empirical result. To improve practical adoption, the paper would be strengthened by synthesizing these findings into explicit guidelines or heuristics for practitioners. For example, the authors could provide a recommended strategy for setting k based on dataset characteristics or model capacity.
2.The core mechanism of TARF cleverly leverages "representation gravity" for target identification. The authors commendably acknowledge the limitations of this assumption in Appendix G.2, especially for semantically indistinct concepts. Given its foundational importance, this discussion is critical. Acknowledging this limitation in the main paper (e.g., in the conclusion or future work section) would offer a more balanced perspective on the method's applicability. This also opens up an interesting avenue for future research, such as integrating external knowledge to bolster target identification when representation similarity is ambiguous.

**Questions:**

1.The data-driven approach for setting β by ranking dynamic responses is elegant. My question centers on its robustness in more challenging scenarios: How sensitive are the final unlearning outcomes to the choice of the quantile used to determine β? Does this estimation method remain stable in cases where the "false retain" set is either very small or disproportionately large compared to the initially provided forget set?
2.The appendix provides a good discussion on computational cost, noting its comparability to other methods. Focusing specifically on foundation-model-scale applications, where even a single forward pass on the entire remaining dataset (Dun) for monitoring can be expensive, could the authors elaborate on practical optimization strategies? For example, would a sampling-based approach to estimate the "gravity effects" on a subset of Dun be a viable strategy, and what might be the trade-offs?

---

> ### Author Response · Authors · 2025-11-19
> **Response to Reviewer gpUj [1/3]**
>
> We sincerely thank the reviewer for the time and effort spent reviewing our paper and for acknowledging the pioneering problem formulation, principled and generalized framework, and comprehensive and rigorous validations. Below, we address the raised questions and concerns in detail:
>
> > **W1:** While the appendix contains a detailed ablation study for key hyperparameters (Figure 17), the discussion is currently presented as an empirical result. To improve practical adoption, the paper would be strengthened by synthesizing these findings into explicit guidelines or heuristics for practitioners. For example, the authors could provide a recommended strategy for setting k based on dataset characteristics or model capacity.
>
> **A1:** We appreciate the constructive comments from the reviewer. And we would like to supplement the recommended strategy with conceptual understandings and empirical explanations to strengthen the guidelines.
>
> From a **conceptual perspective**, the hyperparameters introduced in TARF are **structurally constrained by the roles they play in the objective in Eq. (3)**. The initial forgetting strength $k$ determines how aggressively the model separates the target representation of gradient ascent, while $\tau$ determines the scope of samples used for reconstruction and is governed by an automatically ranking threshold $\beta$. The time-related value $t_1$​ and $t_0$​ delimit the phases of target identification, target separation, and retraining approximation. They are not arbitrary tuning knobs but follow the stage transitions illustrated in Fig. 4, which **align with Theorem 3.2 and Figure 3 indicating that forgetting dynamics** are informative in the early stage for identifying target samples, whereas later updates risk inducing unexpected feature distortion. This theoretical connection explains how $k$, $t_1$, $t_0$​ function primarily as structural controls rather than sensitively tunable hyperparameters.
>
> Building on this intuition, we can **summarize the practical and empirically validated guidelines:** **First, $k$** should be initialized to a small value and increased cautiously; a modest early ascent is sufficient to reveal representation gravity, while overly large values cause unnecessary distortion. **Second, $t_1$​** can generally be set very early (typically Epoch 1), since the gravity-based ranking relies on the first few steps of forgetting. **Third, $t_0$​** can be tuned by extending Phase III when additional retaining approximation is needed, ensuring that any feature distortion induced by early forgetting is corrected. These **principles are consistently supported by our ablations:** Figures 7 and 17 show that varying the hyperparameters across wide ranges yields nearly identical behaviors except for extreme values that contradict the above intuition. We will **synthesize these insights into a concise “hyperparameter guidelines” remark** in the revised version to further facilitate adoption in practice.
>
> > **W2:** The core mechanism of TARF cleverly leverages "representation gravity" for target identification. The authors commendably acknowledge the limitations of this assumption in Appendix G.2, especially for semantically indistinct concepts. Given its foundational importance, this discussion is critical. Acknowledging this limitation in the main paper (e.g., in the conclusion or future work section) would offer a more balanced perspective on the method's applicability. This also opens up an interesting avenue for future research, such as integrating external knowledge to bolster target identification when representation similarity is ambiguous.
>
> **A2:**  Thank the reviewer for the acknowledgement and constructive comments, and we would acknowledge the limitation in the main paper with the following discussion in the conclusion section. We hope the **following discussion** can enrich the comprehensive understanding of our introduced mismatch settings, and provide some insight on promising directions for future research.

---

> > ### Author Response · Authors · 2025-11-19
> > **Response to Reviewer gpUj [2/3]**
> >
> > **Opening Challenge and Further Discussion.** Representation gravity, that uses early forgetting dynamics as a proxy for semantic proximity, is central to TARF’s ability to identify latent target concepts. In challenging regimes where concepts are inherently ambiguous, weakly clustered, or attribute-entangled (e.g., certain long-tailed or multi-attribute scenarios), the underlying representation structure itself becomes less separable. This phenomenon affects all existing unlearning methods rather than TARF specifically, as the ambiguity originates from the nature of the data rather than the mechanism. As shown in Appendix G.2, we also observe a few preliminary cases where the gravity signal becomes weaker and the ranking slightly noisier; nevertheless, TARF continues to demonstrate consistent advantages over baselines with the consideration of mismatched scenarios. We therefore view these situations not as method-specific limitations, but as inherent difficulties when the target concept is not well-defined in the representation space. At the same time, these cases also suggest promising avenues for future research, such as incorporating external knowledge (e.g., text embeddings, semantic priors, or multimodal cues) to assist target identification when intrinsic representation similarity is ambiguous.
> >
> > > **Q1:** The data-driven approach for setting $\beta$ by ranking dynamic responses is elegant. My question centers on its robustness in more challenging scenarios: How sensitive are the final unlearning outcomes to the choice of the quantile used to determine $\beta$? Does this estimation method remain stable in cases where the "false retain" set is either very small or disproportionately large compared to the initially provided forget set?
> >
> > **A3:** Thanks for the thoughtful question regarding the quantile-based selection of $\beta$. We would like to clarify that **the robustness** of this choice fundamentally **arises from the structure of the early forgetting dynamics rather than from the quantile itself.** As established in Section 3.2, early gradient-ascent updates induce a distinctive separation in dynamic responses: samples that share the underlying target concept consistently form a sharp “gravity tail” in the loss/accuracy-change ranking, while unrelated samples exhibit relatively weaker responses. Because this **separation is a property of the representation geometry and the forgetting dynamics**, the reliability of $\beta$ reflects the validity and stability of the gravity effect,, as the specific quantile is also assumed to be prior information which is both common in literature and practical as the unlearning request may have predefined interested range.
> >
> > **Table 1. Sensitive check of quantile-based choice on varied false-retain size using CIFAR-100.**
> > | Forgetting Support | Exact Size | UA   | RA    | TA    | MIA    | Gap  |
> > |-----|------|------|-------|-------|--------|-----|
> > Retrained (Ref.)  | 450         | 0.00 | 97.76  | 74.28  | 100.00   | -   |
> > GA (large)   | 450   | 6.35 | 92.32  | 70.12  | 94.53   | 5.36   |
> > TARF (large) | 450   | 0.00   | 96.42  | 72.13 | 100.00  | **0.87**   |
> > Retrained (Ref.)  | 2250  | 0.00 | 98.03  | 73.42  | 100.00  | -  |
> > GA (small)   | 2250  | 35.07 | 91.81  | 66.39  | 75.91  | 18.10   |
> > TARF (small) | 2250   | 23.37 | 85.53  | 70.68  | 77.82   | **15.20**   |
> >
> > For the reviewer’s question **concerning more challenging imbalance conditions**, we provide **empirical verification showing that our TARF remains stable** even though the dynamic-response ranking can include some noise false-retaining data in the above Table 1. **When the false-retain set is very small**, these data remain reliably captured by the top quantiles due to their strong semantic and representation alignment with the forgetting set, and the unlearning performance is good. **In contrast, when the false-retain set is significantly enlarged**, the ranking could include a number of noisy samples as their semantic similarity to the forgetting set is inherently weak. Although we can not achieve accurate forgetting (e.g., achieve 0\% in UA), we can still perform better than plain gradient ascent on the given forgetting data. **This situation also corresponds to the challenging settings discussed in A2**, where representation-gravity cues become ambiguous when the false-retain samples are disproportionately large and given forgetting data can not be representative anymore. Nevertheless, across all feasible mismatch scenarios introduced in the paper, the quantile-based ranking choice remains structurally robust, and quantile-based choice of $\beta$ can be valid if the forgetting dynamics can well capture the semantic representativeness. We will include the expanded results and discussion in the revised manuscript.

---

> > > ### Author Response · Authors · 2025-11-19
> > > **Response to Reviewer gpUj [3/3]**
> > >
> > > > **Q2:** The appendix provides a good discussion on computational cost, noting its comparability to other methods. Focusing specifically on foundation-model-scale applications, where even a single forward pass on the entire remaining dataset ($D_\rm{un}$) for monitoring can be expensive, could the authors elaborate on practical optimization strategies? For example, would a sampling-based approach to estimate the "gravity effects" on a subset of $D_\rm{un}$ be a viable strategy, and what might be the trade-offs?
> > >
> > > **A4:** We appreciate the reviewer’s insightful question regarding optimization strategies to large-scale-application settings. Regarding the “gravity effects” monitoring, we would like to clarify that it relies on the gradient ascent, which **is inherently aligned with the forgetting objective** instead of a separate operation for computing solely. The required dynamic information is obtained **using the same forward pass that is already necessary for unlearning to compute the loss value.** We further supplement the discussion with a computational cost table as follows in our revised version, which shows the **additional ranking** procedure (TIME-Ra) for analyzing the gravity effects is **minimal compared with forward/backward pass** in the whole unlearning process (TIME-Un).
> > >
> > > **Table 1. Computational comparison of target identification compared with unlearning procedure.**
> > > | Dataset       | TIME-Ra   | TIME-In | TIME-Un | UA   | RA    | TA    | MIA    | Gap  |
> > > |---------------|-----------|---------|---------|------|-------|-------|--------|------|
> > > | CIFAR-10      | 0.0000017 | 0.18    | 4.23    | 0.06 | 97.57 | 90.81 | 100.00 | 1.23 |
> > > | CIFAR-100     | 0.0000017 | 0.18    | 4.85    | 0.31 | 97.35 | 73.68 | 100.00 | 0.21 |
> > > | Tiny-ImageNet | 0.0000018 | 0.95    | 32.81   | 1.08 | 94.78 | 69.91 | 100.00 | 1.37 |
> > > | ImageNet      | 0.0000018 | 11.52   | 628.87  | 0.00 | 69.93 | 71.79 | 100.00 | 3.97 |
> > >
> > > Regarding the full forward pass over the entire remaining set, **we also regard it as a potential opportunity for accelerating machine unlearning** as long as the representation gravity holds for strong semantic alignment, where we can use a part of forgetting data to unlearn the entire target or identify the entire target. Below we discuss two potential practical optimization strategies regarding the checking of dynamic responses.
> > > - First, **sampling-based approaches** like uniform or stratified sampling are a viable strategy in large-scale applications, especially when the dataset contains structural information such as category tags or class labels. TARF depends only on relative changes in the gravity shift, rather than exact global estimates, so monitoring the shift using a small, class-balanced validation subset (e.g., a few thousand examples) is feasible. The primary trade-off is representational fidelity: if the sampled subset fails to capture the intra-class structural variation, the estimated gravity shift may introduce mild variance.
> > > - Second, a **lightweight surrogate model** may be used to approximate gravity monitoring. Since TARF relies on representation gravity for target identification, we may also compute the gravity shift using a compressed encoder if available (e.g., a distilled representation model or low-rank projection of the backbone). This reduces monitoring costs by optimizing the forwarding pass, with the trade-off that the surrogate model may introduce slight bias in the absolute magnitude of the gravity shift. We may also consider the time cost of constructing a surrogate model based on the existing literature.
> > >
> > > We will add a discussion in the revised version and point out the promising future direction towards optimizing the efficient unlearning target identification, which is significant to enable the large-scale unlearning applications. Again we appreciate the reviewer’s thoughtful questions.
> > >
> > > ---
> > > We appreciate the reviewer’s comments again to improve our presentation, and **hope our responses have addressed your concerns. Please don’t hesitate to let us know if there are any remaining points** we can clarify or strengthen, and we always welcome further discussion.

---

> > > > ### Author Response · Authors · 2025-11-27
> > > > **Look forward to your reply**
> > > >
> > > > Dear Reviewer gpUj,
> > > >
> > > > Thanks again for your time in reviewing and your comments! We have provided detailed responses and conducted the revision (summarized in General Response) in our updated manuscript.
> > > >
> > > > Would you mind confirming if there is any unclear point so that we should/could further clarify? We will do our best to address them. Additionally, if you find our responses helpful, we would sincerely appreciate it if you could acknowledge the efforts we have made.
> > > >
> > > > Best regards,
> > > >
> > > > Authors of Submission #9776

---

### Official Review · Reviewer_7K9D · 2025-11-03

**Soundness:** 4
**Presentation:** 4
**Contribution:** 3
**Rating:** 10
**Confidence:** 2

**Summary:**

This work addresses the unlearning by exploring label domain mismatch and introduces novel forgetting challenges such as target, model, and data mismatches. The proposed TARF framework addresses these challenges, showcasing promising results in various experiments, thus contributing significantly to advancing the unlearning paradigm.

**Strengths:**

1. The paper presents a novel method that decouples the class label from the target concept, addressing practical scenarios where the forgetting target may not align with pre-training taxonomy.
2. authors provide theoretical insights that introducing the representation gravity. The solid analysis for understanding unlearning dynamics at the representation level is very important.
3. extensive experiments show consisent results with the proposed claims.

**Weaknesses:**

TARF introduces multiple dynamic hyperparameters (e.g., k(t), τ(x,y,t), β, t₀, t₁), but their tuning procedure is only heuristically described. Ablation results help, but further discussion on computational stability and robustness would strengthen the paper.

**Questions:**

How sensitive is TARF to the scheduling of k(t) and τ(x,y,t)? Can the authors provide empirical or theoretical justification for the annealing rate and threshold choices?

---

> ### Author Response · Authors · 2025-11-19
> **Response to Reviewer 7K9D [1/2]**
>
> We sincerely thank the reviewer for the time and effort spent reviewing our paper and for acknowledging the novelty of our setting/method, theoretical insights, and extensive experiments. Below, we address the raised questions and concerns in detail:
>
> > **W1:** TARF introduces multiple dynamic hyperparameters (e.g., $k(t)$, $\tau(x,y,t)$, $\beta$, $t_0$, $t_1$), but their tuning procedure is only heuristically described. Ablation results help, but further discussion on computational stability and robustness would strengthen the paper.
>
> **A1:** We appreciate the reviewer’s acknowledgement of our ablation and constructive comments for discussion. We fully agree with that and would **supplement the discussion** with **conceptual explanations and empirical interpretations** to strengthen the understanding of tuning principles. Conceptually, those **hyperparameters introduced in TARF** are **structurally constrained by their functionality**, although they introduced extra tuning flexibility to enable the capability of handling different mismatched scenarios. We can understand from **an induction view of our unlearning objective** in Eq.(3), where **$k$ and $\tau$ respectively control the strength of forgetting and the scope of retaining**. Generalized from the Phase-II of target separation in Figure 4, $t_1$ and $t_0$ enable the Phase-I for target identification on target/data mismatch and the Phase-III for retaining approximation. Note that as discussed near lines 272 and 1589, $\beta$ is an automatic ranking threshold in realizing the index $\tau$. Thus, we only need to decide the proper value of $k$, $t_1$ and $t_0$, which is guided by specific unlearning scenarios. **Given that intuition, we have several tuning principles:** **1) the initial forgetting strength $k$** can be tuned from a smaller value to avoid extra feature distortion; **2) $t_1$** is generally set to be Epoch 1 as dynamic information can be captured by ranking mechanism; **3) $t_0$** can be also tuned by extending the Phase-III to fix the potential feature distortion induced by forgetting.
>
> Empirically, **the above intuition and tuning principles benefit the computational stability** of our framework, as demonstrated in our ablations which consistently shows that TARF is stable across a wide range of choices. As shown in Figures 7 and 17, sweeping the values of initial $k$, $t_1$, and $t_0$ leads to highly similar outcomes unless: 1) $k$ is set to an unrealistically large value that aggressively destroys the representation; 2) $t_1$ is set to an extremely large value obscure large quota of retaining; 3) $t_0$ is set to near 0 without considering fixing representation destroy. This aligns with Eq. (2), where the early steps dominate the separation of target representations, while the later stage can induce unrelated feature distortion. We will add the discussion in the appendix related to our algorithm design and full ablation for hyperparameters, and also consider a concise version in the main text if there is enough space in our final revision.

---

> > ### Author Response · Authors · 2025-11-19
> > **Response to Reviewer 7K9D [2/2]**
> >
> > >**Q1:** How sensitive is TARF to the scheduling of k(t) and τ(x,y,t)? Can the authors provide empirical or theoretical justification for the annealing rate and threshold choices?
> >
> > **A2:** We thank the reviewer for raising the question. We would further clarify that the scheduling of these two dynamic hyperparameters is not sensitive under the guidance of theoretical insights.  According to Theorem 3.2, we can reveal that the **forgetting dynamics can be constrained** by the gravity effects at the early stage, while the later influence of forgetting can be an arbitrary spill to the retaining part. It can be intuitively understood on with the loss curves of different part of data in Figure 3, which **support the rationality of setting annealing forgetting strength of $k$ to avoid later over-forgetting on remaining data**, and control $\tau(t)$ to only use the early stage of unlearning to conduct target identification.
> >
> > Empirically, we use $t_1$ and $t_0$ to mark when the process shifts from identification to separation and then to retraining approximation. **The annealing factor k(t) is designed to gradually reduce the effect of gradient ascent as the model transitions toward retraining behavior.** We use the initial forgetting strength $k$ and $t_0$ value to control the annealing rate, which aligns the theoretical insight that forgetting should be strong early to induce representation separation, but gradually attenuated to avoid over-forgetting. Under the tuning principle discussed in A1, we can get the robust observation in Figure 17. For $\tau$, it is realized by the threshold $\beta$, which is obtained by an automatic ranking procedure with a pre-assumed forgetting range, **its robustness follows naturally from** Theorem 3.2 that **samples with small representation distance share tightly coupled loss dynamics**, so we can use the proportional value of ranking as $\beta$ for target identification. In summary, although TARF uses dynamic hyperparameters, they are not sensitive knobs but rather **structural mechanisms that follow the principle guide supported by Theorem 3.2**. We will further discuss the robustness observations in the related discussion of our final version.
> >
> > ---
> > We appreciate the reviewer’s comments again to improve our presentation, and **hope our responses have addressed your concerns. Please don’t hesitate to let us know if there are any remaining points** we can clarify or strengthen, and we always welcome further discussion.

---

> > > ### Author Response · Authors · 2025-11-27
> > > **Look forward to your reply**
> > >
> > > Dear Reviewer 7K9D,
> > >
> > > Thanks again for your time in reviewing and your comments! We have provided detailed responses and conducted the revision (summarized in General Response) in our updated manuscript.
> > >
> > > Would you mind confirming if there is any unclear point so that we should/could further clarify? We will do our best to address them. Additionally, if you find our responses helpful, we would sincerely appreciate it if you could acknowledge the efforts we have made.
> > >
> > > Best regards,
> > >
> > > Authors of Submission #9776

---

### Author Response · Authors · 2025-11-23
**General Response**

We appreciate all the reviewers for their thoughtful comments and suggestions on our work.

**We are encouraged to see that the reviewers find** our introducing **novel settings** to decouple the class labels from the target concepts, where the **research formulation is pioneering and highly practical**, **fundamentally expands the scope** of class-wise unlearning, and **bridge a critical gap** between research and real-world requirements (*Reviewer 7K9D, gpUj, ES8Q, MLTw*); The proposed **TARF is a principled and general framework** for machine unlearning in mismatched scenarios, which is **based on solid analysis and theoretical insights** on representation gravity of forgetting dynamics, and **provides a robust and general design** within a unified objective to **handle the systematically revealed challenges** in four scenarios (*Reviewer 7K9D, gpUj, ES8Q*); The experiments are **extensive, comprehensive, and convincing** to demonstrate the effectiveness of the method **over wide range of baselines**, show **promising results in various scenarios**, and also include case studies powerfully **underscores practical utility** (*Reviewer 7K9D, gpUj, ES8Q, MLTw*).

We have tried our best to address the reviewers' comments in **individual responses to each reviewer** with further discussion and justification. **Those valuable comments allowed us to improve our draft** and the contents added in the **revised version** (highlighted in orange) are summarized below:

**From Reviewer 7K9D**
- Add discussion on computational stability with hyperparameters. (in Section 4.3 and Appendix F.1)
- Add discussion on scheduling robustness under theoretical implications, and practical guidance. (in Appendix F.1)

**From Reviewer gpUj**
- Add practical guidance under theoretical implication for key hyperparameters. (in Appendix F.1)
- Add explicit discussion on open challenges and future work in the main text. (in Section 5)
- Add discussion and further exploration of quantile-based thresholding. (in Section 4.3 and Appendix F.1)
- Add empirical results of computation cost and discussion for potential sampling methods. (in Appendix F.2)

**From Reviewer ES8Q**
- Revise the obscure content and unclear legends (in Fig.5, Fig.7 and Table 2)
- Add empirical results and clarification on computation overhead (in Section 4.3 and Appendix F.2)
- Add practical guidance under theoretical implication for key hyperparameters. (in Appendix F.1)
- Clarify the core mechanism of target identification and scalability of multiple forgetting. (in Section 3.2 and Appendix G.9)

**From Reviewer MLTw**
- Add the implication remark and intuitive explanation of analytical results. (in Section 3.2)
- Add intuition and connection from analytical insights with methods. (in Section 3.2 and 3.3)
- Highlight cross-reference for related work discussion. (in Section 2 and Appendix B)
- Enhance the dense presentation with further explanation. (in Section 3.3)
- Add more recent baselines for comparison and discussion. (in Appendix C.1)
- Revise and recheck minor typos.
---
We appreciate all the reviewer’s time and valuable comments again. We would like to kindly ask **would you mind checking it and confirming if there needs further clarification**. We look forward to continuing the valuable discussion and your constructive feedback during the discussion period!

---

### Meta-Review · Area_Chair_JiB1 · 2025-12-29

**Summary:**

The reviewers broadly agree that this paper introduces a novel and practically important formulation of machine unlearning by explicitly decoupling class labels from target concepts, substantially extending the scope of prior class-wise unlearning work. The main concerns motivating discussion focused on clarity and accessibility rather than correctness: several reviewers noted that the theoretical analysis and key concepts such as representation gravity and decomposition were initially difficult to follow, the connection between theory and method design was not sufficiently explicit, and parts of the presentation—including figures, legends, and related work positioning—were dense or underdeveloped. Additional concerns included the perceived complexity of dynamic hyperparameters, the need for clearer computational overhead analysis, and questions about robustness and scalability in challenging mismatch scenarios, especially when semantic signals are weak or multiple targets are involved. Despite these issues, most reviewers viewed the formulation, framework design, and experimental breadth as strong, with reservations centered on exposition and practitioner-facing guidance rather than fundamental flaws.

**Reviewer Concerns:**

The rebuttal and revisions substantively addressed many of the core concerns raised during review. The authors provided clearer intuition for the theoretical analysis, explicitly linked the representation-level insights to the design of TARF, and expanded explanations of key concepts that were previously perceived as abstract or arbitrary. Practical guidance for hyperparameter selection, robustness discussions for annealing and quantile-based thresholds, and additional clarification on computational overhead were added, directly responding to multiple reviewers’ questions. Presentation issues were also improved through revised figures, clarified legends, corrected typos, and strengthened cross-references to related work. Remaining concerns primarily reflect residual skepticism about whether the framework remains intuitive for all readers, especially in extreme cases where representation gravity signals are weak, and whether the complexity of the method may hinder adoption despite the added explanations. However, these outstanding issues are largely about scope limitations and clarity rather than unaddressed technical weaknesses.

**Reviewer Scores:**

Given the discussion and rebuttal, Reviewer 7K9D would likely maintain a very high score, and Reviewer gpUj would likely remain strongly positive, since their main requests for clearer hyperparameter guidance, robustness justification, and practical considerations were addressed with more explicit intuition and supporting evidence. Reviewer ES8Q would likely keep a similar score but with slightly increased confidence, as the revision improves readability, clarifies overhead, and adds more actionable guidance and discussion on scope and scalability. Reviewer MLTw, who was primarily concerned about unclear intuition, weak linkage between analysis and method, presentation density, and baseline coverage, might modestly raise their score after the added explanations, strengthened cross-references, and expanded comparisons, though some reservations about accessibility and perceived complexity would likely persist. Overall, the post-rebuttal scores would still show some dispersion, but the reduced concern about arbitrariness and improved clarity make a weak accept recommendation more defensible.

---

### Decision · Program_Chairs · 2026-01-26

Accept (Poster)